# Gradient Descent as a Perceptron Algorithm:
# Understanding Dynamics and Implicit Acceleration

**Alexander Tyurin** [1][2]

## Abstract

Even for the gradient descent (GD) method applied to neural network training, understanding its *optimization dynamics*, including convergence rate, iterate trajectories, function value oscillations, and especially its *implicit acceleration*, remains a challenging problem. We analyze nonlinear models with the logistic loss and show that the steps of GD reduce to those of *generalized perceptron algorithms* (Rosenblatt, 1958), providing a new perspective on the dynamics. This reduction yields significantly simpler algorithmic steps, which we analyze using classical linear algebra tools. Using these tools, we demonstrate on a minimalistic example that the nonlinearity in a two-layer model can provably yield a faster iteration complexity $\tilde{\mathcal{O}}(\sqrt{d})$ compared to $\Omega(d)$ achieved by linear models, where $d$ is the number of features. This helps explain the *optimization dynamics* and the *implicit acceleration* phenomenon observed in neural networks. The theoretical results are supported by extensive numerical experiments. We believe that this alternative view will further advance research on the optimization of neural networks.

## 1. Introduction

Consider the gradient descent method applied to minimize the classical logistic regression problem $\frac{1}{n} \sum_{i=1}^{n} \log(1 + \exp(-y_i b_i^\top v))$, where $\{(b_i, y_i)\}_{i=1}^{n}$ is a dataset with $b_i \in \mathbb{R}^d$ and $y_i \in \{-1, 1\}$ for all $i \in [n] := \{1, \ldots, n\}$ (Bishop & Nasrabadi, 2006). Although it is one of the most popular approaches in the machine learning literature, logistic regression with GD is still being explored (Wu et al., 2024a; Tyurin, 2025; Zhang et al., 2025b;a), as it serves as a foun-

dation for understanding the nonlinear case with neural networks and large language models.

We take one step further and investigate the training problem with the gradient descent (GD) method

$$w_{t+1} = w_t - \gamma \nabla f(w_t), \tag{GD}$$

together with the logistic loss and *nonlinear* models $m : \mathbb{R}^d \times \mathbb{R}^p \to \mathbb{R}$:

$$\min_{w \in \mathbb{R}^p} \left\{ f(w) := \frac{1}{n} \sum_{i=1}^{n} \log \left(1 + \exp(-y_i m(b_i; w))\right) \right\} \tag{1}$$

where $w_0$ is a starting point and $\gamma > 0$ is a step size. In the case of the logistic regression problem, the model is linear and defined as $\boxed{m_{\text{lin}}(b; v) := b^\top v}$, where $v \in \mathbb{R}^d$ and $b \in \mathbb{R}^d$. However, the main object of our interest is the two-layer model $m_{\text{two}}(b; \mathbf{C}, v) := (\mathbf{C}b)^\top v$, where $\mathbf{C} \in \mathbb{R}^{f \times d}$ and $v \in \mathbb{R}^f$ are the weights, and, in particular, its special case

$$\boxed{m_{\text{cv}}(b; c, v) := (c * b)^\top v, \ c \in \mathbb{R}^k, \ v, b \in \mathbb{R}^d,} \tag{2}$$

where we take the features $b \in \mathbb{R}^d$ and apply the convolution operation[1] with the kernel $c \in \mathbb{R}^k$ of size $k \in \mathbb{N}$ before multiplying by a vector $v \in \mathbb{R}^d$ (throughout the paper, we primarily focus on the case $k = 2$). These nonlinear models can be interpreted as two-layer neural networks without an activation. The importance of $m_{\text{cv}}$ model and our focus on it are explained in Section 1.1.

Instead of considering a fully general nonlinear model, we take a relatively small step by adding a small convolutional kernel to the linear model and investigate how this modification affects the behavior of GD. Only afterwards, we discuss extensions to larger nonlinear models in Section 6.

### 1.1. Observing the dynamics and implicit acceleration experimentally

We begin with a simple experiment, considering (1) with linear model $m_{\text{lin}}$ and samples from classes 0 and 1 of the CIFAR-10 dataset (Krizhevsky et al., 2009). We randomly

[1]AXXX, Moscow, Russia [2]Applied AI Institute, Moscow, Russia. Correspondence to: Alexander Tyurin <alexander-tiurin@gmail.com>.

*Proceedings of the 43ʳᵈ International Conference on Machine Learning*, Seoul, South Korea. PMLR 306, 2026. Copyright 2026 by the author(s).

---

[1]The convolution operation $c, b \mapsto c * b$ returns the vector $y \in \mathbb{R}^d$ such that the $i^{\text{th}}$ coordinate equals $[y]_i = \sum_{j=1}^{k} [b]_{i-j+1} [c]_j$ for all $i \in [d]$, where $[b]_0 \equiv [b]_d$, $[b]_{-1} \equiv [b]_{d-1}$, and so forth.

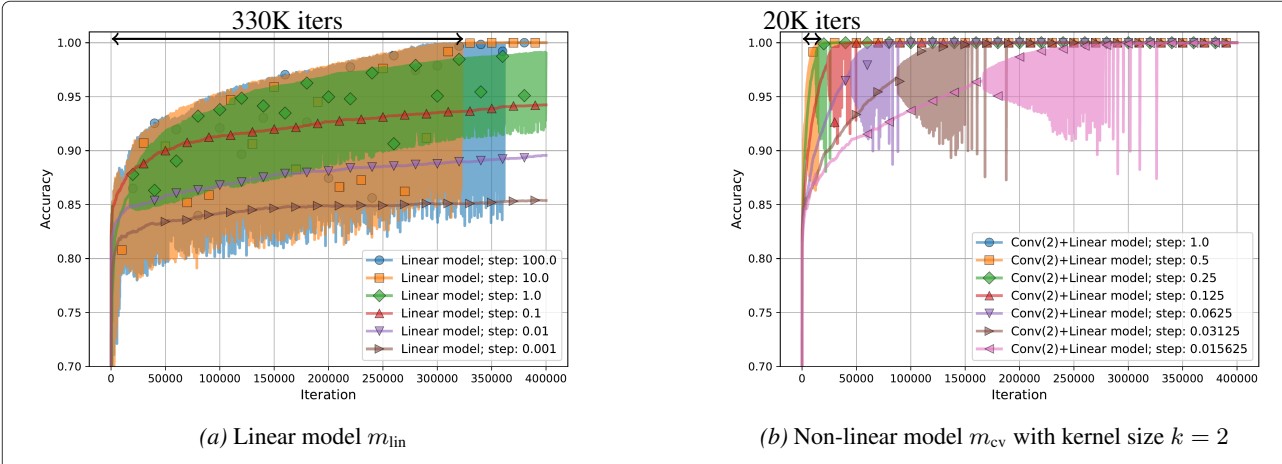

*(a)* Linear model $m_{\text{lin}}$

*(b)* Non-linear model $m_{\text{cv}}$ with kernel size $k = 2$

*Figure 1.* Comparison of accuracies for linear model $m_{\text{lin}}$ and nonlinear model $m_{\text{cv}}$ trained on CIFAR-10 with classes 0 and 1, and # of samples $n = 5000$. The loss values are in Fig. 4 and 5. *The main observation is that, surprisingly, the nonlinear model $m_{\text{cv}}$ converges much faster than the linear model $m_{\text{lin}}$. See discussion in Section 1.1.*

subsample 2500 samples from each class (in Section G.1, we examine other datasets, sample sizes, and classes, and provide additional implementation details). We run GD with step sizes $\gamma \in \{10^{-3}, \ldots, 10^2\}$, as shown in Figure 1a.

In Figure 1a, GD with the *linear model* $m_{\text{lin}}$ and the optimal step size $\gamma = 10.0$ converges non-monotonically, finding a separator between the two classes, that is, a vector $v$ that perfectly classifies the samples, after about **330K iterations**. Next, we repeat exactly the same experiment with the same subset of samples but using *nonlinear model* $m_{\text{cv}}$ from (2) with kernel size $k = 2$ and step sizes $\gamma \in \{2^{-6}, \ldots, 2^0\}$, and observe a surprising result in Figure 1b: the nonlinear model (2) finds the separator after around **20K iterations**, more than **15 times faster than the linear model.**

Notice that $m_{\text{cv}}$ is a reparameterization of $m_{\text{lin}}$. Indeed, for all $c \in \mathbb{R}^k$ and $v \in \mathbb{R}^d$, there exists a vector $\bar{v}$ such that $(c * b)^\top v = b^\top \bar{v}$ for all $b \in \mathbb{R}^d$, since both the convolution and inner product operations are linear operations.

Our first hypothesis was that there might be something special about CIFAR-10 and the chosen classes. Therefore, we repeated this experiment with other popular datasets, classes, and sample sizes, and observed the same effect in almost all cases (see Section G.1). Somehow, although $m_{\text{cv}}$ is just a reparameterization of $m_{\text{lin}}$, the nonlinearity in (2) can significantly accelerate the training speed of GD with the logistic loss.

### 1.2. Previous work: non-stable convergence of the linear model $m_{\text{lin}}$

Before we start exploring the reasons behind the acceleration of the nonlinear model, it is important to recall what we know about the logistic regression problem with linear

model $m_{\text{lin}}$. Notice an interesting and relatively little-known behavior of GD on a separable dataset in Figures 1a and 4. If the step size $\gamma$ is small, the method converges monotonically, in accordance with the classical convex optimization theory (Nesterov, 2018). However, beyond a particular threshold, it enters a *non-stable* regime in which GD converges non-monotonically. *Notice that there is no randomness in the method*: we run deterministic GD and calculate the full gradients without sampling. Oscillations naturally appear with separable data and large step sizes in GD.

Using slightly different techniques, Tyurin (2025); Zhang et al. (2025b;a) explored this phenomenon, and showed that GD converges with arbitrary large step sizes. Tyurin (2025) noticed that GD with linear model $m_{\text{lin}}$ and $\gamma \to \infty$ reduces to a batch version of the celebrated perceptron algorithm (Rosenblatt, 1958; Block, 1962; Novikoff, 1962):

**Theorem 1.1.** *(Tyurin, 2025) For $\gamma \to \infty$ and $w_0 = 0$, logistic regression (1) with GD and linear model $m_{\text{lin}}(b; v) = b^\top v$ reduces to the batch perceptron algorithm defined as*

$$z_{t+1} = z_t + \frac{\phi}{n} \sum_{i \in S_t} y_i b_i, \ \ S_t = \{i \in [n] \, : \, y_i b_i^\top z_t \leq 0\},$$

(Perceptron Algorithm)

*where $z_1 = z_0 + \frac{1}{2n} \sum_{i=1}^n y_i b_i, z_0 = 0$, and step size $\phi = 1$. In particular, $w_t/\gamma \overset{\gamma \to \infty}{\to} z_t$ for all $t \geq 1$ and almost all datasets (does not require the dataset to be separable).*

Under the assumption that $\omega := \max_{\|v\|=1} \min_{i \in [n]} y_i b_i^\top v > 0$, it remains to combine Theorem 1.1 with the classical arguments from (Block, 1962; Novikoff, 1962; Duda et al., 2001) to prove that logistic regression with GD and $\gamma \to \infty$ finds a separator after at most $\mathcal{O}\left(nR^2/\omega^2\right)$ iterations, where

$R := \max_{i \in [n]} \|b_i\|$. *Thus, for large $\gamma$, the oscillations in Figure 1a of the linear model correspond to the steps of Perceptron Algorithm.*

The goal now is to consider the nonlinear model (2) and explain how the nonlinear and chaotic dynamics of GD in this setting can lead to the *accelerated non-monotonic* convergence. Our initial guess is that the dynamics of the nonlinear model are also connected to generalized perceptron algorithms, which we will see is indeed true.

### 1.3. Contributions

♠ **Reduction to Quadratic Perceptron Algorithm.** Our work begins with the observation of the implicit acceleration and the non-monotonic chaotic dynamics discussed in Section 1.1. To understand them, we start with the key finding that GD with the logistic loss (1) and the nonlinear model $m_{\text{cv}}$ reduces to a quadratic extension of the classical perceptron algorithm when the norm of the initial point $w_0$ is large (Section 2). This reduction leads to significantly simpler algorithmic steps (Quadratic Perceptron Algorithm), which we analyze using classical linear algebra tools (Sections 3 and 4).

♣ **Provably faster convergence of the nonlinear model.** Using the discovered reduction and properties, we analyze a quadratic perceptron algorithm and prove that it requires *at most $\tilde{\Theta}(\sqrt{d}/\mu)$* steps to find a separator on a minimalistic dataset, where $d$ is the feature dimension and $\mu$ is the margin between the samples (Section 5). We also show that the classical perceptron algorithm requires *at least $\Omega(d/\mu)$* steps on the same dataset, where the dependence on $d$ is worse. These results highlight and help explain the gap observed in Section 1.1 and Figure 1, as both $m_{\text{lin}}$ and $m_{\text{cv}}$ reduce to perceptron algorithms in the sense of Theorems 1.1 and 2.1.

♦ **Extension to multi-layer models.** We extend our result to multi-layer models and show that they can be also reduced to Generalized Perceptron Algorithm. Our theoretical insights are predictive. For instance, we observe that the maximal allowed step size decreases with larger kernel sizes, which can be easily explained through our reduction (Section 6).

♥ **Numerical experiments.** Our work combines both theoretical and empirical perspectives: each theoretical insight is motivated and corroborated through extensive experiments conducted on different datasets and setups (Sections 1.1 and G).

## 2. From Logistic Loss to Quadratic Perceptron Algorithm

We consider the case where the kernel size is $k = 2$ in (2). Let us take a closer look at one step of GD with $m_{\text{cv}}$. In

Section B.1, we show that it is equivalent to the step

$$w_{t+1} = w_t + \frac{\gamma}{n} \sum_{i=1}^{n} \frac{1}{1 + \exp(\frac{1}{2} w_t^\top \mathbf{A}_i w_t)} \mathbf{A}_i w_t, \quad (3)$$

where $w_t := \begin{bmatrix} [c_t]_1 & [c_t]_2 & v_t^\top \end{bmatrix}^\top \in \mathbb{R}^{d+2}$ for all $t \geq 0$,

$$\mathbf{A}_i := \begin{bmatrix} 0 & 0 & y_i b_i^\top \\ 0 & 0 & y_i b_i^\top \mathbf{P}^\top \\ y_i b_i & y_i \mathbf{P} b_i & \mathbf{0}_d \end{bmatrix} \in \mathbb{R}^{(d+2) \times (d+2)},$$

$$\mathbf{P} := \begin{bmatrix} 0 & 0 & \cdots & 0 & 1 \\ 1 & 0 & \cdots & 0 & 0 \\ 0 & 1 & \cdots & 0 & 0 \\ \vdots & \vdots & \ddots & \vdots & \vdots \\ 0 & 0 & \cdots & 1 & 0 \end{bmatrix} \in \mathbb{R}^{d \times d} \quad (4)$$

Here, $\mathbf{P}$ denotes the standard permutation matrix, which circularly shifts coordinates to the right, and $\mathbf{A}_i$ is the particular matrix induced by the structure of $(c * b)^\top v$.

At first glance, there is no connection between (3) and classical perceptron algorithms. Our first idea is to consider the regime where the starting point $w_0$ has a large norm. Formally, we study (3) under the assumption that $\|w_0\| \to \infty$. Surprisingly, (3) remains (almost) well-defined in this case and reduces to a generalized perceptron algorithm, as shown by the following theorem.

---

**Theorem 2.1** (Reduction to Quadratic Perceptron Algorithm). *Consider the steps (3). Assume that the direction $\theta_0 := \frac{w_0}{\|w_0\|} \in \mathbb{R}^{d+2}$ of the starting point $w_0$ is fixed, $\theta_0^\top \mathbf{A}_i \theta_0 \neq 0$ for all $i \in [n]$, and $\|w_0\| \to \infty$. For almost all choices[2] of $\gamma < 1/\max_{i \in [n]} \|\mathbf{A}_i\|$ and for $\|w_0\| \to \infty$,*

$\frac{w_{t+1}}{\|w_{t+1}\|}$ *is well-defined and equals* $\frac{\theta_{t+1}}{\|\theta_{t+1}\|}$, *where*

$$\theta_{t+1} = \theta_t + \frac{\gamma}{n} \sum_{i \in S_t} \mathbf{A}_i \theta_t,$$

$$S_t = \left\{ i \in [n] : \frac{1}{2} \theta_t^\top \mathbf{A}_i \theta_t \leq 0 \right\}$$

(Quadratic Perceptron Algorithm)

*Moreover, the predictions of $w_t$ and $\theta_t$ are equal, i.e., for $\|w_0\| \to \infty$, $\text{sign}(m_{\text{cv}}(b_i; w_t)) = \text{sign}(m_{\text{cv}}(b_i; \theta_t))$ for all $i \in [n]$ and $t \geq 0$.*

---

Thus, our initial guess was valid: **the dynamics of the nonlinear model in Figure 1b are those of a generalized perceptron algorithm in the regime when $\|w_0\|$ is large.** Indeed, compare Perceptron Algorithm and Quadratic Perceptron Algorithm. These two methods are very similar in that they construct sets $\{S_t\}$ of samples that violate the models' predictions in every iteration, and only use these samples to make an update. We observe that they can be unified into the following generalized perceptron algorithm:

---

[2]There is a set of step sizes of measure zero, which depends on $\theta_0$, where we can not guarantee the statement of the theorem.

$$\theta_{t+1} = \theta_t + \frac{\gamma_t}{n}\sum_{i \in S_t} \nabla h_i(\theta_t),$$

$$S_t = \{i \in [n] \,:\, h_i(\theta_t) \leq 0\},$$

(Generalized Perceptron Algorithm)

where $h_i : \mathbb{R}^p \to \mathbb{R}$ is a transformation associated with sample $i$ under the current model. In the cases of Perceptron Algorithm and Quadratic Perceptron Algorithm, the corresponding transformations for sample $i$ are $z \mapsto y_i b_i^\top z$ and $\theta \mapsto \frac{1}{2}\theta^\top \mathbf{A}_i \theta$, respectively. While the convergence analysis of Perceptron Algorithm with the former transformation is well known, since it is essentially a minor variation of the perceptron algorithm, to the best of our knowledge, the latter case with Quadratic Perceptron Algorithm has not been explored.

**How can the reduction help us?** Quadratic Perceptron Algorithm has arguably much simpler algorithmic steps compared to those of (3). In every iteration, the steps are $\theta_{t+1} = \mathbf{B}_t \theta_t$, where $\mathbf{B}_t := \mathbf{I} + \frac{\gamma}{n}\sum_{i \in S_t} \mathbf{A}_i$ is a *symmetric and positive definite matrix* for all $\gamma < 1/\max_{i \in [n]} \|\mathbf{A}_i\|$. Intuitively, what remains is to understand the properties of $\mathbf{A}_i$ and use classical tools from linear algebra to analyze the optimization dynamics.

**Numerical observation of the norms.** When we run GD with $m_{\mathrm{cv}}$, we observe that even if $\|w_0\| \approx 1$, the norm of subsequent iterates increases, and the method (3) *automatically* tends to the regime where $\|w_{t_0}\|$ is large for some $t_0 \geq 1$ (see Section G.3). Thus, the assumption that $\|w_0\|$ is large is practical up to the first "warm-up" steps $t_0$. Alternatively, it is sufficient to initialize the starting point with a large norm for Theorem 2.1 to hold. In Section G.4, we run the nonlinear model starting from iterates with large norms and observe similar behavior.

## 3. On the Choice of Step Size $\gamma$

The condition in Theorem 2.1 that we must take $\gamma < \gamma_{\max} := 1/\max_{i \in [n]} \|\mathbf{A}_i\|$ is important: the maximal norm of the matrices $\mathbf{A}_i$ determines the largest allowed step size, and we will show that convergence can be guaranteed when $\gamma = \Theta(\gamma_{\max})$. This observation stands in contrast to the classical choice based on the maximal eigenvalue of the Hessian. Let us illustrate the difference:

**Proposition 3.1.** *Consider the logistic loss with* (2). *There exists a dataset* $\{(y_1, b_1)\}$ *of size one such that* $\|b_1\| = 1$, $\|\mathbf{A}_1\| \leq \sqrt{2}$, $\|\nabla^2 f(w)\| \geq \frac{1}{20}\|w\|^2 - \sqrt{2}$, $\frac{\sqrt{2}}{20}\|w\| \leq \|\nabla f(w)\| \leq \sqrt{2}\|w\|$, *and* $\frac{1}{20} \leq f(w) \leq 5$ *for all* $w \in K := \{\alpha v_1 + \beta v_2 \,:\, 0 \leq \beta \leq \alpha \leq \sqrt{\beta^2 + 1}\}$, *where* $v_1, v_2 \in \mathbb{R}^{d+2}$ *are orthonormal vectors.*

According to Proposition 3.1, $\|\mathbf{A}_1\| \leq \sqrt{2}$. Thus, we are allowed to take any $\gamma < \|\mathbf{A}_1\| \leq \sqrt{2}$ in Theorem 2.1, and the choice *does not* depend on the current iterate $w$.

There exists $w \in K$ with arbitrarily large $\|w\|$ such that $\|\nabla^2 f(w)\| \geq \Omega(\|w\|^2)$, and $\nabla^2 f(w)$ is neither $L$–smooth for a finite $L$ nor $(L_0, L_1)$–smooth (Zhang et al., 2019), since $\|\nabla f(w)\| = \Theta(\|w\|)$, nor $(\rho, K_0, K_\rho)$–smooth (Liu et al., 2025), since $f(w) = \Theta(1)$.

In light of this, we arguably need alternative analysis, ones that do not rely on smoothness or bounded Hessians, to explain the non-monotonic and chaotic convergence behavior observed in Figure 1b when using constant, relatively large step sizes $\gamma$. Our reduction in Theorem 2.1 represents one possible direction that helps to explain this convergence.

## 4. Properties of Quadratic Perceptron Algorithm

We now analyze the matrices $\mathbf{A}_i$ from Quadratic Perceptron Algorithm, induced by the nonlinear model $m_{\mathrm{cv}}$ (2), and prove the following properties, which we believe are essential to understand the convergence of Quadratic Perceptron Algorithm and the dynamics in Figure 1b.

---

**Proposition 4.1.** *Consider* $a \in \mathbb{R}^d$ *and*
$$\mathbf{A} = \begin{bmatrix} 0 & 0 & a^\top \\ 0 & 0 & a^\top \mathbf{P}^\top \\ a & \mathbf{P}a & \mathbf{0}_d \end{bmatrix} \in \mathbb{R}^{(d+2)\times(d+2)}.$$
*1. If $a = 0$, then $\mathbf{A}$ is the zero matrix with $d + 2$ zero eigenvalues.*
*2. If $a \neq 0$ and $a = \mathbf{P}a$, then $\mathbf{A}$ has two nonzero eigenvalues $\sqrt{2}\|a\|$ and $-\sqrt{2}\|a\|$ with the corresponding eigenvectors $v_1 = \begin{bmatrix} \|a\| & \|a\| & \sqrt{2}a^\top \end{bmatrix}^\top$ and $v_2 = \begin{bmatrix} -\|a\| & -\|a\| & \sqrt{2}a^\top \end{bmatrix}^\top$.*
*3. If $a \neq 0$ and $a = -\mathbf{P}a$, then $\mathbf{A}$ has two nonzero eigenvalues $\sqrt{2}\|a\|$ and $-\sqrt{2}\|a\|$ with the corresponding eigenvectors $v_1 = \begin{bmatrix} \|a\| & -\|a\| & \sqrt{2}a^\top \end{bmatrix}^\top$ and $v_2 = \begin{bmatrix} -\|a\| & \|a\| & \sqrt{2}a^\top \end{bmatrix}^\top$.*
*4. If $a \neq 0, a \neq \mathbf{P}a$, and $a \neq -\mathbf{P}a$, then $\mathbf{A}$ has four nonzero eigenvalues $\sqrt{\|a\|^2 + a^\top \mathbf{P}a}, -\sqrt{\|a\|^2 + a^\top \mathbf{P}a}, \sqrt{\|a\|^2 - a^\top \mathbf{P}a}, -\sqrt{\|a\|^2 - a^\top \mathbf{P}a}$ with the corresponding eigenvectors*
$$v_1 = \begin{bmatrix} \bar{a}_{+,\mathbf{P}} & \bar{a}_{+,\mathbf{P}} & (a + \mathbf{P}a)^\top \end{bmatrix}^\top,$$
$$v_2 = \begin{bmatrix} -\bar{a}_{+,\mathbf{P}} & -\bar{a}_{+,\mathbf{P}} & (a + \mathbf{P}a)^\top \end{bmatrix}^\top,$$
$$v_3 = \begin{bmatrix} \bar{a}_{-,\mathbf{P}} & -\bar{a}_{-,\mathbf{P}} & (a - \mathbf{P}a)^\top \end{bmatrix}^\top,$$
*and* $v_4 = \begin{bmatrix} -\bar{a}_{-,\mathbf{P}} & \bar{a}_{-,\mathbf{P}} & (a - \mathbf{P}a)^\top \end{bmatrix}^\top$,
*where* $\bar{a}_{+,\mathbf{P}} := \sqrt{\|a\|^2 + a^\top \mathbf{P}a}$ *and* $\bar{a}_{-,\mathbf{P}} := \sqrt{\|a\|^2 - a^\top \mathbf{P}a}$.
*5. $\|a\| \leq \|\mathbf{A}\| \leq \sqrt{2}\|a\|$.*

---

Proposition 4.1 uncovers the low-rank spectrum of $\{\mathbf{A}_i\}$, which we will use later in the analysis.

**Quadratic Perceptron Algorithm does not converge in general.** It is possible to prove that Perceptron Algorithm converges for any separable and any starting point (Duda et al., 2001; Tyurin, 2025). However, using Proposition 4.1, we can show it is not the case for Quadratic Perceptron Algorithm:

**Theorem 4.2.** *For any dataset with the number of samples $n < (d + 2)/4$, there exists a subspace $V$ of at least dimension $(d + 2) - 4n$ from which Quadratic Perceptron Algorithm does not converge.*

The result follows from the fact that the matrices $\mathbf{A}_i$ have low ranks; thus, there exists a subspace $V$ such that $\mathbf{A}_i\theta_0 = 0$ for all $\theta_0 \in V$, and Quadratic Perceptron Algorithm does not move. Nevertheless, the measure of subspace $V$ is zero since $V$ is a strict subspace of $\mathbb{R}^{d+2}$. Moreover, when we tested this result in practice, we observed that Quadratic Perceptron Algorithm converges even if we start from the subspace $V$. The reason is that the numerical computations of convolutions, matrix multiplications, and other operations are not precise and allow the algorithm to leave the subspace $V$. *The subspace $V$ is not stable.* This observation motivates us to consider the following algorithm:

$$\theta_{t+1} = \theta_t + \frac{\gamma}{n} \sum_{i \in S_t} \mathbf{A}_i\theta_t + \xi_t,$$
$$S_t = \left\{ i \in [n] : \frac{1}{2}\theta_t^\top \mathbf{A}_i\theta_t \leq 0 \right\}, \quad \xi_t \sim \mathcal{N}(0, \sigma^2 \mathbf{I}_{d+2}),$$
(Quadratic Perceptron Algorithm with Noise)

where $\xi_t$ are i.i.d. random variables from the normal distribution $\mathcal{N}(0, \sigma^2 \mathbf{I}_{d+2})$ and $\sigma^2$ is a small variance. Note that these random perturbations are introduced for theoretical purposes. In practice, however, random perturbations naturally arise due to numerical errors, making Quadratic Perceptron Algorithm with Noise and Quadratic Perceptron Algorithm effectively equivalent.

## 5. Convergence Guarantees on Minimalistic Example

We now present a comparison between the classical perceptron and quadratic perceptron approaches, and consider the following simple dataset with two samples:

$$a_1 := y_1 b_1, \; b_1 = [1, 1, \ldots, 1]^\top \in \mathbb{R}^d, \text{ and } y_1 = 1; \quad (5)$$
$$a_2 := y_2 b_2, \; b_2 = [1 + \mu, 1, \ldots, 1]^\top \in \mathbb{R}^d, \text{ and } y_2 = -1,$$

where $\mu > 0$ is a margin. The corresponding matrices from (4) are $\mathbf{A}_1$ and $\mathbf{A}_2$. Notice that the features $b_1$ and $b_2$ are close to each other, and their distance is controlled by the parameter $\mu > 0$: the smaller the $\mu$, the more difficult it is for an algorithm to find a separator between $b_1$ and $b_2$. Consider a non-batch version of Quadratic Perceptron Algorithm with Noise with two samples:

$$\theta_{t+1} =$$
$$\begin{cases} \theta_t + \gamma \mathbf{A}_1\theta_t + \xi_t, & \text{if } \theta_t^\top \mathbf{A}_1\theta_t \leq 0, \\ \theta_t + \gamma \mathbf{A}_2\theta_t + \xi_t, & \text{if } \theta_t^\top \mathbf{A}_1\theta_t > 0 \text{ and } \theta_t^\top \mathbf{A}_2\theta_t \leq 0, \end{cases}$$
$$\xi_t \sim \mathcal{N}(0, \sigma^2 \mathbf{I}_{d+2}),$$
(Two-Sample Quadratic Perceptron Algorithm)

which is executed until $\theta_t^\top \mathbf{A}_1\theta_t > 0$ and $\theta_t^\top \mathbf{A}_2\theta_t > 0$. As in the classical perceptron algorithm, if the first sample is correctly classified, we update the point using that sample; otherwise, we use the other sample. To simplify the analysis, compared to Quadratic Perceptron Algorithm with Noise, when both samples are misclassified, we consider only the first sample in the update. Analyzing Two-Sample Quadratic Perceptron Algorithm, we can prove the following convergence rate:

**Theorem 5.1** (Upper Bound for Two-Sample Quadratic Perceptron Algorithm). *Consider Two-Sample Quadratic Perceptron Algorithm on the dataset* (5). *With probability greater than or equal to $1 - \rho$, the required number of steps to find a separator is at most*

$$T = \left\lceil B \times \frac{\sqrt{d}}{\mu} \right\rceil = \tilde{\Theta}\left( \frac{\sqrt{d}}{\mu} \right),$$

*where* $B := A \log\left( \frac{\sqrt{d}}{\mu} A \right) = \tilde{\Theta}(1)$, $A := 2^{27} \log\left( \frac{2^{62}\|\theta_0\|^2}{\rho^3 \langle \theta_0, v_{\mu,+} \rangle^2} \right) \log\left( \frac{2^{15}\sqrt{d}\|\theta_0\|^2}{\mu \langle \theta_0, v_{\mu,+} \rangle^2} \right) = \tilde{\Theta}(1)$, *if* $\gamma = \frac{1}{4\sqrt{d}}$, $\mu \leq \frac{1}{10}$, $\sigma = |\langle \theta_0, v_{\mu,+} \rangle| / (4096T\sqrt{\max\{d, \log(T/\rho)\}})$, *and* $\langle \theta_0, v_{\mu,+} \rangle \neq 0$.

The theorem states that the number of steps required to find a separator is $\tilde{\Theta}(\sqrt{d}/\mu)$ with step size $\gamma = \Theta(1/\max_{i \in \{1,2\}} \|\mathbf{A}_i\|)$ and almost all starting points[3] $\theta_0$. In contrast, the convergence guarantees of Perceptron Algorithm are provably worse (up to logarithmic factors):

**Theorem 5.2** (Lower Bound for Perceptron Algorithm). *The batch perceptron algorithm (Perceptron Algorithm) requires at least*

$$\Omega\left( \frac{d}{\mu} \right)$$

*steps to find a separator of the dataset* (5) *for any $\varepsilon \leq \frac{\mu}{8\max\{\|a_1\|, \|a_2\|\}}$, $\mu \leq 1/2$, step size $\phi > 0$, and for any starting point $z_0 \in B(0, \phi\varepsilon)$.*

**Discussion.** Comparing Theorem 5.1 and Theorem 5.2, we see that the quadratic perceptron approach achieves a $\sqrt{d}$-times better complexity. These theorems help to explain the gap between $m_{\text{lin}}$ and $m_{\text{cv}}$ observed in Section 1.1 and

---

[3]For instance, if $\theta_0 \sim \mathcal{N}(0, \mathbf{I}_{d+2})$, then $\mathbb{P}\left( \langle \theta_0, v_{\mu,+} \rangle = 0 \right) = 0$.

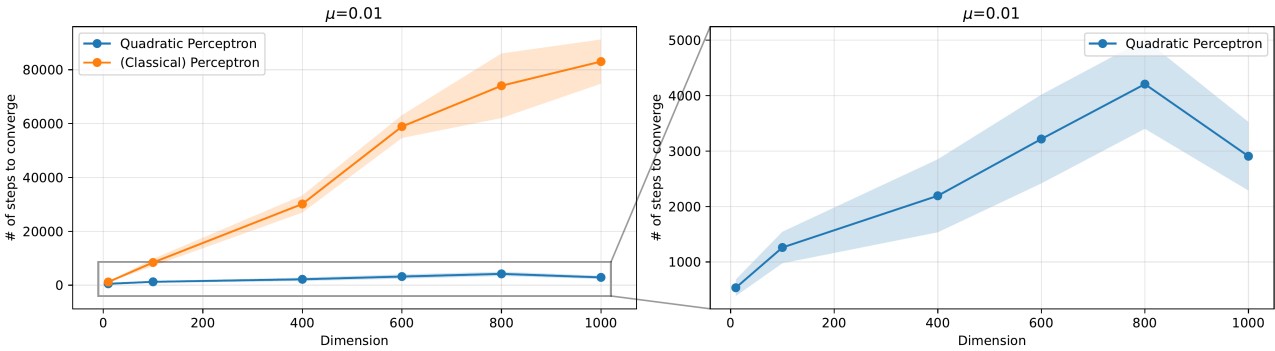

*Figure 2.* Perceptron Algorithm and Two-Sample Quadratic Perceptron Algorithm on dataset (5).

Figure 1. Thus, the main reason for the acceleration is that GD with $m_{\mathrm{cv}}$ is more robust to the dimensionality $d$.

**Numerical test of the theorems.** It is easy to verify that our results hold numerically on the dataset (5). In Figure 2, we fix $\mu = 0.01$, take $d \in \{10, 100, 400, 600, 800, 1000\}$, and compare Perceptron Algorithm and Two-Sample Quadratic Perceptron Algorithm. For each step size from $\{2^{-10}, \dots, 2^{9}\}$, we run each method 30 times from a uniformly random point in $S(0, 1)$ and choose the plot corresponding to the step size with the smallest mean number of iterations required to find a separator. The experiments align with Theorems 5.1 and 5.2: in the case of Perceptron Algorithm, the number of steps increases linearly with $d$, while the dependence in Two-Sample Quadratic Perceptron Algorithm is $\sqrt{d}$.

### 5.1. Proof sketch of Theorem 5.1

The main idea behind the result and proof can be explained using Propositions 4.1 and 5.3.

**Proposition 5.3** (follows from Proposition 4.1). *Consider the matrices $\mathbf{A}_1$ and $\mathbf{A}_2$ of the dataset (5).*

*1. $\mathbf{A}_1$ has two non-zero eigenvalues $\lambda_1$ and $-\lambda_1$ with the corresponding eigenvectors $v_{1,+}$ and $v_{1,-}$, where $\lambda_1 := \sqrt{2d}$.*

*2. $\mathbf{A}_2$ has four non-zero eigenvalues $\lambda_2, -\lambda_2, \mu, -\mu$ with the corresponding eigenvectors $v_{2,+}, v_{2,-}, v_{\mu,+}$, and $v_{\mu,-}$, where $\lambda_2 := \sqrt{(2+\mu)^2 + 2(d-2)}$.*

*3. $v_{\mu,+}, v_{\mu,-} \in \ker(\mathbf{A}_1)$.*

Due to the eigendecomposition of $\mathbf{A}_1$ and $\mathbf{A}_2$,

$$\theta_t^\top \mathbf{A}_1 \theta_t = \lambda_1 \langle v_{1,+}, \theta_t \rangle^2 - \lambda_1 \langle v_{1,-}, \theta_t \rangle^2$$

and

$$\theta_t^\top \mathbf{A}_2 \theta_t = \lambda_2 \langle v_{2,+}, \theta_t \rangle^2 - \lambda_2 \langle v_{2,-}, \theta_t \rangle^2 + \mu \langle v_{\mu,+}, \theta_t \rangle^2 - \mu \langle v_{\mu,-}, \theta_t \rangle^2. \tag{6}$$

(**Number of steps of option 1**): At the beginning, assume that $\theta_0^\top \mathbf{A}_1 \theta_0 \leq 0$ and $\sigma = 0$ in Two-Sample Quadratic Perceptron Algorithm; then option 1 is chosen in Two-Sample Quadratic Perceptron Algorithm, $\theta_1 = (\mathbf{I} + \gamma \mathbf{A}_1) \theta_0$, and $\theta_1^\top \mathbf{A}_1 \theta_1 = \langle (\mathbf{I} + \gamma \mathbf{A}_1) \theta_0, \mathbf{A}_1 (\mathbf{I} + \gamma \mathbf{A}_1) \theta_0 \rangle$. Using Proposition 5.3 and the eigendecomposition, $\theta_1^\top \mathbf{A}_1 \theta_1 = \lambda_1 \langle v_{1,+}, (\mathbf{I} + \gamma \mathbf{A}_1) \theta_0 \rangle^2 - \lambda_1 \langle v_{1,-}, (\mathbf{I} + \gamma \mathbf{A}_1) \theta_0 \rangle^2 = \lambda_1 (1 + \gamma \lambda_1)^2 \langle v_{1,+}, \theta_0 \rangle^2 - \lambda_1 (1 - \gamma \lambda_1)^2 \langle v_{1,-}, \theta_0 \rangle^2$. Repeating $s$ more times,

$$\theta_s^\top \mathbf{A}_1 \theta_s = $$
$$\lambda_1 (1 + \gamma \lambda_1)^{2s} \langle v_{1,+}, \theta_0 \rangle^2 - \lambda_1 (1 - \gamma \lambda_1)^{2s} \langle v_{1,-}, \theta_0 \rangle^2.$$

Thus, there will be the smallest $s_1 = \mathcal{O}\left(\log_{\frac{1+\gamma\lambda_1}{1-\gamma\lambda_1}} \frac{\langle v_{1,-}, \theta_0 \rangle^2}{\langle v_{1,+}, \theta_0 \rangle^2}\right) = \tilde{\mathcal{O}}(1)$ such that $\theta_{s_1}^\top \mathbf{A}_1 \theta_{s_1} > 0$, meaning that the method can only choose option 2 in Two-Sample Quadratic Perceptron Algorithm at step $s_1 + 1$.

(**Number of steps of option 2**): Next, using Proposition 5.3 with matrix $\mathbf{A}_2$, for all $s \geq 1$ steps of option 2 in Two-Sample Quadratic Perceptron Algorithm,

$$\theta_{s+s_1}^\top \mathbf{A}_2 \theta_{s+s_1} = $$
$$\lambda_2 (1 + \gamma \lambda_2)^{2s} \langle v_{2,+}, \theta_{s_1} \rangle^2 - \lambda_2 (1 - \gamma \lambda_2)^{2s} \langle v_{2,-}, \theta_{s_1} \rangle^2$$
$$+ \mu (1 + \gamma \mu)^{2s} \langle v_{\mu,+}, \theta_{s_1} \rangle^2 - \mu (1 - \gamma \mu)^{2s} \langle v_{\mu,-}, \theta_{s_1} \rangle^2, \tag{7}$$

and there exists the smallest $s_2 = \tilde{\mathcal{O}}(1)$ such that $\theta_{s_2+s_1}^\top \mathbf{A}_2 \theta_{s_2+s_1} > 0$ or $\theta_{s_2+s_1}^\top \mathbf{A}_1 \theta_{s_2+s_1} \leq 0$ if $\langle v_{2,+}, \theta_{s_1} \rangle^2 \neq 0$ or $\langle v_{\mu,+}, \theta_{s_1} \rangle^2 \neq 0$.

(**Finite "ping-pong" between options 1 and 2**): Notice that it might be possible that $\theta_{s_1+s_2}^\top \mathbf{A}_1 \theta_{s_1+s_2} \leq 0$, and option 1 repeats again. Therefore, we have a "ping-pong" between options 1 and 2 in Two-Sample Quadratic Perceptron Algorithm, which "almost surely" will stop because, fortunately, if option 1 is chosen, then $\langle v_{\mu,+}, \theta_{t+1} \rangle^2 = \langle v_{\mu,+}, \theta_t \rangle^2$ and $\langle v_{\mu,-}, \theta_{t+1} \rangle^2 = \langle v_{\mu,-}, \theta_t \rangle^2$ due to Proposition 5.3. If option 2 is chosen, then $\langle v_{\mu,+}, \theta_{t+1} \rangle^2 = (1 + \gamma \mu)^2 \langle v_{\mu,+}, \theta_t \rangle^2$ and

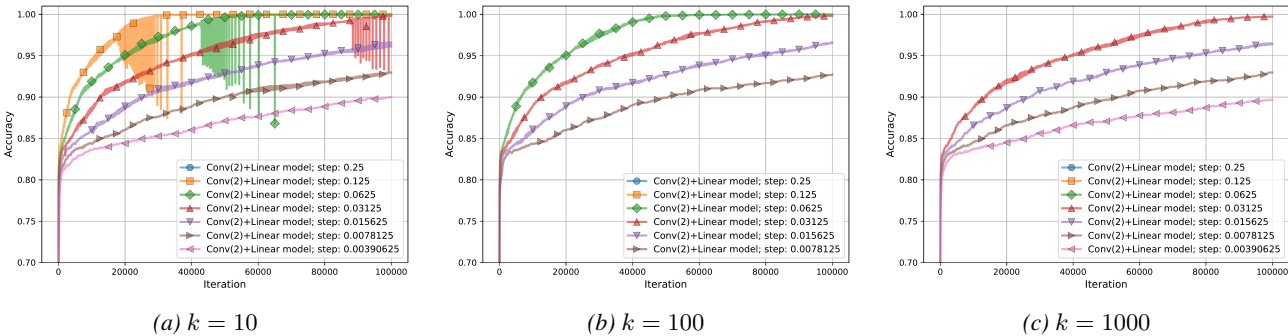

*(a) $k = 10$*             *(b) $k = 100$*             *(c) $k = 1000$*

*Figure 3.* Accuracies for nonlinear model $m_{\mathrm{cv}}$ trained on CIFAR-10 with two classes 0 and 1, # of samples $n = 5000$, and different kernel sizes.

$\langle v_{\mu,-}, \theta_{t+1} \rangle^2 = (1 - \gamma\mu)^2 \langle v_{\mu,-}, \theta_t \rangle^2$. *A subsequence of the sequence* $\{\langle v_{\mu,+}, \theta_t \rangle^2\}_{t \geq 0}$ *increases exponentially. Therefore, option 2 cannot be chosen after several steps $t$, since* $\langle v_{\mu,+}, \theta_t \rangle^2$ *would become too large in* (6); *the algorithm thus necessarily stops and finds a separator.*

> **High-level intuition:** in other words, when *Two-Sample Quadratic Perceptron Algorithm makes the update $\theta_t + \gamma \mathbf{A}_1 \theta_t$, $\theta_t^\top \mathbf{A}_1 \theta_t$ tends to increase and $\theta_t^\top \mathbf{A}_2 \theta_t$ tends to decrease, and vice versa with $\theta_t + \gamma \mathbf{A}_2 \theta_t$. Intuitively, the two matrices play against each other, and we observe a "ping-pong" or "zig-zagging" effect in Figure 1b. However, it will "almost surely" stop since the correlation $\langle v_{\mu,+}, \theta_t \rangle^2$ between $\theta_t$ and the special direction $v_{\mu,+}$ increases, and there will be a moment $t$ when $\theta_t^\top \mathbf{A}_2 \theta_t > 0$ even making the updates $\theta_t + \gamma \mathbf{A}_1 \theta_t$ before.*

The rate $\tilde{\mathcal{O}}(\sqrt{d}/\mu)$ comes from two facts: first, we have to take $\gamma = \mathcal{O}(1/\sqrt{d})$ due to (2.1) and to ensure that $0 < \|\mathbf{I} + \gamma \mathbf{A}_i\| < 2$ for all $i \in [n]$ (see also Section 3), so that the steps do not "explode." This is a natural condition induced by $\max_{i \in [n]} \|\mathbf{A}_i\| = \Theta(\sqrt{d})$. Next, the correlation $|\langle v_{\mu,+}, \theta_t \rangle|$ grows as $\Theta(\exp(t\mu\gamma))$; thus, it will break some necessary threshold after $\tilde{\Theta}(1/\gamma\mu) = \tilde{\Theta}(\sqrt{d}/\mu)$ steps.

**Additional challenges in the proof of Theorem 5.1.** There are a few minor but important details that make the proof technical. The first problem arises in (7). For instance, it might be possible that $\langle v_{2,+}, \theta_{s_1} \rangle^2 = 0$ and $\langle v_{\mu,+}, \theta_{s_1} \rangle^2 = 0$, and $\theta_{s+s_1}^\top \mathbf{A}_2 \theta_{s+s_1} \leq 0$ for all $s \geq 1$. To avoid this, as we explain in Section 4, we have to introduce random noise to escape the "bad" subspace. Another, no less important, problem is to prove that the norm of $\theta_t$ does not increase too fast. This step is crucial to show that the numbers of consecutive steps of options 1 and 2 are $\tilde{\mathcal{O}}(1)$; for instance, in (7), if $\|\theta_{s_1}\|$ is too large, the number of steps $s_2$ might also be huge.

To the best of our knowledge, this is the first attempt to analyze Two-Sample Quadratic Perceptron Algorithm and Quadratic Perceptron Algorithm with Noise. We hope that subsequent research in this area will not only generalize the result but also significantly simplify the proofs using our initial ideas.

## 6. Extending to Larger Kernel Sizes and Multi-Layer Models

Our view of the optimization dynamics through Quadratic Perceptron Algorithm is predictive not only for $m_{\mathrm{cv}}$ with kernel size $k = 2$. In particular, we repeat the experiment from Figure 1b with larger kernel sizes (see Figure 3) and observe that, when GD converges, the largest step size decreases as the kernel size increases. A similar observation holds for the dataset (5) in Table 2. The following result can explain this.

**Larger kernel sizes.** For a kernel of size $k \geq 2$, the results from Section 2 still hold, with the only change that

$$\mathbf{A}_i := \begin{bmatrix} 0 & \dots & 0 & y_i b_i^\top \\ \vdots & \ddots & \vdots & \vdots \\ 0 & \dots & 0 & y_i b_i^\top (\mathbf{P}^{k-1})^\top \\ y_i b_i & \dots & y_i \mathbf{P}^{k-1} b_i & \mathbf{0}_d \end{bmatrix}, \quad (8)$$

$\mathbf{A}_i \in \mathbb{R}^{(d+k) \times (d+k)}$ for all $i \in [n]$ and $w_t := \begin{bmatrix} c_t^\top & v_t^\top \end{bmatrix}^\top \in \mathbb{R}^{k+d}$. For this matrix, we can prove the following result.

**Proposition 6.1.** *Consider matrix $\mathbf{A}_i$ from* (8). *Then, the norm of $\mathbf{A}_i$ can be bounded as* $\frac{1}{\sqrt{k}} \left\| \sum_{j=1}^k \mathbf{P}^{j-1} b_i \right\| \leq \|\mathbf{A}_i\| \leq \sqrt{k} \|b_i\|$.

For instance, assume that $b_i = [1, 1, \dots, 1, 1]^\top \in \mathbb{R}^d$ for some $i \in [n]$. In this case, $\|\mathbf{A}_i\| = \sqrt{kd}$. Therefore, according to the recommendation from Section 3, we should take $\gamma < 1/\sqrt{kd}$, and the maximal allowed step size decreases as the kernel size $k$ increases. This phenomenon we observe

in Figure 3.

**Two-layer models.** For a general two-layer model $\mathbf{C}, v \mapsto (\mathbf{C}b_i)^\top v$ with $\mathbf{C} \in \mathbb{R}^{f \times d}$ and $v \in \mathbb{R}^f$, we can also construct the corresponding matrix $\mathbf{A}_i$ and get

$$\mathbf{A}_i := \begin{bmatrix} \mathbf{0}_{fd} & y_i b_i \otimes \mathbf{I}_f \\ y_i b_i^\top \otimes \mathbf{I}_f & \mathbf{0}_f \end{bmatrix} \in \mathbb{R}^{(fd+f) \times (fd+f)}$$

with $w_t := \begin{bmatrix} \text{vec}(\mathbf{C}_t)^\top & v_t^\top \end{bmatrix}^\top \in \mathbb{R}^{fd+f}$ (see Section B.2), where $\otimes$ is the Kronecker product.

**Multi-layer models.** Finally, we can consider a multilinear model $m(b_i; \mathbf{C}_1, \ldots, \mathbf{C}_\ell, v) := (\mathbf{C}_1 \cdots \mathbf{C}_\ell b_i)^\top v$, where $\mathbf{C}_\ell \in \mathbb{R}^{f_\ell \times d}, \ldots, \mathbf{C}_1 \in \mathbb{R}^{f_1 \times f_2}, v \in \mathbb{R}^{f_1}$, and prove that GD with normalization reduces to Generalized Perceptron Algorithm with $h_i(\theta) := \frac{1}{\ell+1} \mathbf{A}_i[\theta, \ldots, \theta]_{\ell+1}$, where $\mathbf{A}_i$ is an $(\ell+1)$–multilinear map induced by the structure of the multilinear model (see Section F).

Notice that the rank of the matrices $\mathbf{A}_i$ increases significantly with more complex models, which makes developing a generalized analysis of Section 5.1 even more challenging. This is the main reason why we start with the case of a small kernel size $k = 2$. Extending our results for the kernel size $k = 2$ from Section 5 to larger kernels and more general two-linear models remains a challenging direction and may require new mathematical tools. Another important direction is to extend the eigenvalue and eigenvector analysis to multi-layer models and consider activations.

## 7. Related Work

**Classical optimization theory.** Under the $L$-smoothness assumption, the theory of GD in both convex and non-convex settings is well known and well studied. For $L$-smooth functions, the convergence rate of GD is $\mathcal{O}\left(1/\gamma T\right)$ for step sizes $\gamma < 2/L$ (Nesterov, 2003). At the same time, it is possible to find a (quadratic) $L$-smooth function for which GD diverges if $\gamma > 2/L$ (see, e.g., (Cohen et al., 2021)). The value $2/L$ is a fundamental threshold that separates the convergence and divergence regimes of GD.

**Convergence with large step sizes.** In practice, however, GD often converges even when $\gamma \gg 2/L$, seemingly "contradicting" classical theoretical results (Lewkowycz et al., 2020; Cohen et al., 2021). How is this possible? It turns out that the worst-case analysis in classical optimization theory does not fully capture the structure of modern optimization problems. This phenomenon has been studied from various perspectives, including the sharpness of loss functions (characterized by the largest eigenvalue of the Hessian) (Kreisler et al., 2023), non-separable data (Ji & Telgarsky, 2018; Meng et al., 2024), bifurcation theory (Song & Yun, 2023), sharpness dynamics in networks with normalization (Lyu et al., 2022), two-layer linear diagonal networks (Even et al., 2023), self-stabilization effects (Damian et al., 2023;

Wang et al., 2022; Ahn et al., 2022; Ma et al., 2022), and low-dimensional settings (Zhu et al., 2022; Chen & Bruna, 2022; Ahn et al., 2024).

**Logistic loss.** The works by Wu et al. (2024b;a); Tyurin (2025); Zhang et al. (2025b;a) have considered the logistic loss (1) with large step sizes. However, their main focus has been on the linear model $m_{\text{lin}}$. It is worth noting that Zhang et al. (2025a) also studied two-layer networks under the assumption that the last-layer parameters are fixed, which is impractical and does not capture the true dynamics. We take the next step and develop a theory for the realistic setting in which both the first and last layers are optimized.

**Implicit acceleration.** Our observation that nonlinearity in models leads to acceleration is not new. Arora et al. (2018) investigated linear neural networks for regression problems and showed that reparameterization can act as a preconditioner that accelerates training. Our analysis and theoretical framework are orthogonal, as we focus on classification problems with logistic loss and provide a different perspective on explaining this phenomenon through perceptron algorithms. Related analyses of implicit acceleration include studies in sum-product networks (Trapp et al., 2019), infinitely wide neural networks (Littwin et al., 2021), and graph neural networks (Xu et al., 2021).

## 8. Conclusion

We have only scratched the surface. Analyzing the behavior of the algorithm even in the large-norm regime is already a non-trivial task. We hope that, building on our initial results, it will be possible to generalize the theory to the small-norm regime; we leave this for future work. While we established a connection between gradient descent on the logistic loss and perceptron algorithms, and showed that this connection is useful for theoretically explaining implicit acceleration, extending these results remains an open task. We hope that our perspective on training dynamics through Quadratic Perceptron Algorithm and Generalized Perceptron Algorithm, together with its connection to the classical perceptron algorithm (Rosenblatt, 1958), will lead to a better understanding of these dynamics and open new research directions.

## Acknowledgements

The work was supported by the grant for research centers in the field of AI provided by the Ministry of Economic Development of the Russian Federation in accordance with the agreement 000000C313925P4F0002 and the agreement №139-10-2025-033.

## Impact Statement

This paper presents work whose goal is to advance the field of Machine Learning. There are many potential societal

consequences of our work, none which we feel must be specifically highlighted here.

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

# A. Notations

*Table 1.* List of notations used throughout the paper.

| Notation | Meaning |
|---|---|
| $\mathbb{N}$ | The set of natural numbers $\{1, 2, \dots\}$. |
| $[x]_i$ | The $i$-th coordinate of a vector $x$. |
| $[n]$ | Denotes a finite set $\{1, \dots, n\}$. |
| $\mathbf{1}[\cdot]$ | Indicator function, equal to 1 if the condition holds and 0 otherwise. |
| $\|x\|$ | Euclidean norm of a vector $x$. |
| $\|\mathbf{A}\|$ | Spectral norm (largest singular value) of matrix $\mathbf{A}$. |
| $\langle a, b \rangle$ or $a^\top b$ | Standard Euclidean inner product $\sum_i [a]_i [b]_i$. |
| $a * b$ | Standard convolution operation defined in footnote 1. |
| $\otimes$ | Kronecker product. |
| $\mathrm{vec}(\mathbf{A})$ | vectorization of $\mathbf{A} \in \mathbb{R}^{f \times d}$: $\mathrm{vec}(\mathbf{A}) = [[\mathbf{A}]_{1,1} \dots [\mathbf{A}]_{f,1} [\mathbf{A}]_{1,2} \dots [\mathbf{A}]_{f,2} [\mathbf{A}]_{1,d} \dots [\mathbf{A}]_{f,d}]^\top$. |
| $B(a, r)$ | Euclidean ball of radius $r$: $\{x \in \mathbb{R}^d : \|x - a\| \leq r\}$. |
| $S(a, r)$ | Euclidean sphere of radius $r$: $\{x \in \mathbb{R}^d : \|x - a\| = r\}$. |
| $g = \mathcal{O}(f)$ | There exists $C > 0$ such that $g(z) \leq C f(z)$ for all $z \in \mathcal{Z}$. |
| $g = \Omega(f)$ | There exists $C > 0$ such that $g(z) \geq C f(z)$ for all $z \in \mathcal{Z}$. |
| $g = \Theta(f)$ | There exist $C_1, C_2 > 0$ such that $C_1 f(z) \leq g(z) \leq C_2 f(z)$ for all $z \in \mathcal{Z}$. |
| $\tilde{\mathcal{O}}, \tilde{\Omega}$, and $\tilde{\Theta}$ | The same as $\mathcal{O}, \Omega$, and $\Theta$, but up to logarithmic factors. |
| $\mathbf{A}[w_1, \dots, w_\ell]_\ell$ | Multi-lineal map $\mathbf{A}$ with $\ell$ coordinate applied to vectors $(w_1, \dots, w_\ell)$. |
| $\mathbf{0}_d$ | The $d \times d$ zero matrix. |
| $\mathbf{I}_d$ | The $d \times d$ identity matrix. |
| $\mathrm{im}(\mathbf{A})$ | Image of the matrix $\mathbf{A}$. |
| $\mathrm{ker}(\mathbf{A})$ | Kernel of the matrix $\mathbf{A}$. |

# B. Quadratic Perceptron

### B.1. One step of GD

In this section, we derive one GD step for (1) using the model from (2) with $k = 2$. We define $a_i := y_i b_i$ for all $i \in [n]$. Note that

$$y_i v_t^\top (c_t * b_i) = v_t^\top (c_t * a_i) = \frac{1}{2} \begin{bmatrix} [c_t]_1 \\ [c_t]_2 \\ v_t \end{bmatrix}^\top \mathbf{A}_i \begin{bmatrix} [c_t]_1 \\ [c_t]_2 \\ v_t \end{bmatrix} = \frac{1}{2} w_t^\top \mathbf{A}_i w_t, \tag{9}$$

where $\mathbf{A}_i$ is defined in (4). We use that $c_t * a_i = ([c_t]_1 \mathbf{I} + [c_t]_2 \mathbf{P}) a_i$ and $w_t = \begin{bmatrix} [c_t]_1 & [c_t]_2 & v_t^\top \end{bmatrix}^\top$. Thus,

$$
\begin{aligned}
w_{t+1} &= w_t - \gamma \nabla f(w_t) \\
&= w_t + \gamma \frac{1}{n} \sum_{i=1}^n \frac{1}{1 + \exp(\frac{1}{2} w_t^\top \mathbf{A}_i w_t)} \nabla_{w_t} \left( \frac{1}{2} w_t^\top \mathbf{A}_i w_t \right) \\
&= w_t + \gamma \frac{1}{n} \sum_{i=1}^n \frac{1}{1 + \exp(\frac{1}{2} w_t^\top \mathbf{A}_i w_t)} \mathbf{A}_i w_t.
\end{aligned}
\tag{10}
$$

### B.2. One step of GD with two-layer model

In this section, we derive one step of GD for (1) with the nonlinear model $\mathbf{C}, v \mapsto (\mathbf{C}b)^\top v$, where $\mathbf{C} \in \mathbb{R}^{f \times d}$ and $v \in \mathbb{R}^f$ are weights. We define $a_i := y_i b_i$ for all $i \in [n]$. Then,

$$y_i (\mathbf{C}_t b_i)^\top v_t = (\mathbf{C}_t a_i)^\top v_t = v_t^\top \mathbf{C}_t a_i.$$

Since $v_t^\top \mathbf{C}_t a_i \in \mathbb{R}$ is a scalar,

$$y_i \left(\mathbf{C}_t b_i\right)^\top v_t = \text{vec}(v_t^\top \mathbf{C}_t a_i) = (a_i^\top \otimes v_t^\top)\text{vec}(\mathbf{C}_t)$$

due to the property $\text{vec}(\mathbf{AXB}) = (\mathbf{B}^\top \otimes \mathbf{A})\text{vec}(\mathbf{X})$, where $\otimes$ is the Kronecker product. Due to $(a_i^\top \otimes v_t^\top)\text{vec}(\mathbf{C}_t) \in \mathbb{R}$,

$$y_i \left(\mathbf{C}_t b_i\right)^\top v_t = \text{vec}(\mathbf{C}_t)^\top (a_i \otimes v_t) = \text{vec}(\mathbf{C}_t)^\top (a_i \otimes \mathbf{I}_f)v_t.$$

Thus, we can take

$$\mathbf{A}_i := \begin{bmatrix} \mathbf{0}_{fd} & y_i b_i \otimes \mathbf{I}_f \\ y_i b_i^\top \otimes \mathbf{I}_f & \mathbf{0}_f \end{bmatrix}$$

to ensure that

$$\frac{1}{2} \begin{bmatrix} \text{vec}(\mathbf{C}_t) \\ v_t \end{bmatrix}^\top \mathbf{A}_i \begin{bmatrix} \text{vec}(\mathbf{C}_t) \\ v_t \end{bmatrix} = y_i \left(\mathbf{C}_t b_i\right)^\top v_t.$$

All other steps (10) are the same as in Section B.1.

### B.3. The Hessian of $f$

We now calculate the Hessian of $f$. We know that

$$\nabla f(w) = -\frac{1}{n} \sum_{i=1}^n \frac{1}{1 + \exp(\frac{1}{2} w^\top \mathbf{A}_i w)} \mathbf{A}_i w.$$

Therefore,

$$\begin{aligned} \nabla^2 f(w) = &-\frac{1}{n} \sum_{i=1}^n \frac{1}{1 + \exp(\frac{1}{2} w^\top \mathbf{A}_i w)} \mathbf{A}_i \\ &+ \frac{1}{n} \sum_{i=1}^n \frac{1}{1 + \exp(\frac{1}{2} w^\top \mathbf{A}_i w)} \left(1 - \frac{1}{1 + \exp(\frac{1}{2} w^\top \mathbf{A}_i w)}\right) \mathbf{A}_i w w^\top \mathbf{A}_i. \end{aligned} \quad (11)$$

**Proposition 3.1.** *Consider the logistic loss with* (2). *There exists a dataset* $\{(y_1, b_1)\}$ *of size one such that* $\|b_1\| = 1$, $\|\mathbf{A}_1\| \leq \sqrt{2}$, $\|\nabla^2 f(w)\| \geq \frac{1}{20} \|w\|^2 - \sqrt{2}$, $\frac{\sqrt{2}}{20} \|w\| \leq \|\nabla f(w)\| \leq \sqrt{2} \|w\|$, *and* $\frac{1}{20} \leq f(w) \leq 5$ *for all* $w \in K := \{\alpha v_1 + \beta v_2 : 0 \leq \beta \leq \alpha \leq \sqrt{\beta^2 + 1}\}$, *where* $v_1, v_2 \in \mathbb{R}^{d+2}$ *are orthonormal vectors.*

*Proof.* Let us take a dataset of size one with $b_1 = \begin{bmatrix} \frac{1}{\sqrt{d}} & \cdots & \frac{1}{\sqrt{d}} \end{bmatrix}^\top \in \mathbb{R}^d$ and $y_1 = 1$. Let us define $a_1 := y_1 b_1$. Notice that $\|b_1\| = \|a_1\| = 1$ by the construction. Using (11), we have

$$\begin{aligned} &\left\|\nabla^2 f(w)\right\| \\ &\geq \left\|\frac{1}{1 + \exp(\frac{1}{2} w^\top \mathbf{A}_1 w)} \left(1 - \frac{1}{1 + \exp(\frac{1}{2} w^\top \mathbf{A}_1 w)}\right) \mathbf{A}_1 w w^\top \mathbf{A}_1\right\| - \left\|\frac{1}{1 + \exp(\frac{1}{2} w^\top \mathbf{A}_1 w)} \mathbf{A}_1\right\| \\ &= \frac{1}{1 + \exp(\frac{1}{2} w^\top \mathbf{A}_1 w)} \left(1 - \frac{1}{1 + \exp(\frac{1}{2} w^\top \mathbf{A}_1 w)}\right) \left\|\mathbf{A}_1 w w^\top \mathbf{A}_1\right\| - \frac{1}{1 + \exp(\frac{1}{2} w^\top \mathbf{A}_1 w)} \|\mathbf{A}_1\|. \end{aligned}$$

Note that $\mathbf{P} a_1 = a_1$ and the matrix

$$\mathbf{A}_1 = \begin{bmatrix} 0 & 0 & a_1^\top \\ 0 & 0 & a_1^\top \mathbf{P}^\top \\ a_1 & \mathbf{P} a_1 & \mathbf{0}_d \end{bmatrix} = \begin{bmatrix} 0 & 0 & a_1^\top \\ 0 & 0 & a_1^\top \\ a_1 & a_1 & \mathbf{0}_d \end{bmatrix}$$

has two non-zero eigenvalues $\sqrt{2} \|a_1\|$ and $-\sqrt{2} \|a_1\|$ (see Proposition 4.1) with the corresponding orthonormal eigenvectors

$$v_1 \propto \begin{bmatrix} \|a_1\| \\ \|a_1\| \\ \sqrt{2} a_1 \end{bmatrix} \text{ and } v_2 \propto \begin{bmatrix} -\|a_1\| \\ -\|a_1\| \\ \sqrt{2} a_1 \end{bmatrix},$$

where $\propto$ means that we take the normalized vectors collinear the right hand sides. Therefore, $\|\mathbf{A}_1\| = \sqrt{2}\|a_1\|$. Since $\frac{1}{1+\exp(\frac{1}{2}w^\top \mathbf{A}_1 w)} \in [0,1]$, we get

$$\left\|\nabla^2 f(w)\right\|$$
$$\geq \frac{1}{1+\exp(\frac{1}{2}w^\top \mathbf{A}_1 w)}\left(1 - \frac{1}{1+\exp(\frac{1}{2}w^\top \mathbf{A}_1 w)}\right)\left\|\mathbf{A}_1 ww^\top \mathbf{A}_1\right\| - \sqrt{2}\|a_1\|$$
$$= \frac{1}{1+\exp(\frac{1}{2}w^\top \mathbf{A}_1 w)}\left(1 - \frac{1}{1+\exp(\frac{1}{2}w^\top \mathbf{A}_1 w)}\right)w^\top \mathbf{A}_1^2 w - \sqrt{2}\|a_1\|.$$

Let us take $w = \alpha v_1 + \beta v_2$, where $\alpha \geq \beta > 0$. Then

$$w^\top \mathbf{A}_1 w = (\alpha v_1 + \beta v_2)^\top \mathbf{A}_1 (\alpha v_1 + \beta v_2) = \sqrt{2}\|a_1\|(\alpha v_1 + \beta v_2)^\top (\alpha v_1 - \beta v_2)$$
$$= \sqrt{2}\|a_1\|\left(\alpha^2\|v_1\|^2 - \beta^2\|v_2\|^2\right) = \sqrt{2}(\alpha^2 - \beta^2)\|a_1\| \geq 0$$

and

$$w^\top \mathbf{A}_1^2 w = (\alpha v_1 + \beta v_2)^\top \mathbf{A}_1^2 (\alpha v_1 + \beta v_2) = 2\|a_1\|^2 (\alpha v_1 + \beta v_2)^\top (\alpha v_1 + \beta v_2)$$
$$= 2\|a_1\|^2\left(\alpha^2\|v_1\|^2 + \beta^2\|v_2\|^2\right) = 2\left(\alpha^2 + \beta^2\right)\|a_1\|^2$$

because the eigenvectors $v_1$ and $v_2$ are orthonormal. Thus

$$\left\|\nabla^2 f(w)\right\| \geq \frac{\left(\alpha^2 + \beta^2\right)\|a_1\|^2}{1+\exp(\frac{\sqrt{2}}{2}(\alpha^2 - \beta^2)\|a_1\|)} - \sqrt{2}\|a_1\|.$$

Note that $\|w\|^2 = \|\alpha v_1 + \beta v_2\|^2 = \alpha^2 + \beta^2$, meaning that

$$\left\|\nabla^2 f(w)\right\| \geq \frac{\|w\|^2\|a_1\|^2}{1+\exp(\frac{\sqrt{2}}{2}(\alpha^2 - \beta^2)\|a_1\|)} - \sqrt{2}\|a_1\| = \frac{\|w\|^2}{1+\exp(\frac{\sqrt{2}}{2}(\alpha^2 - \beta^2))} - \sqrt{2}.$$

since $\|a_1\| = 1$. Taking $\alpha \leq \sqrt{\beta^2 + 1}$,

$$\left\|\nabla^2 f(w)\right\| \geq \frac{\|w\|^2}{20} - \sqrt{2}.$$

It is left to bound the norm of gradient and the function value:

$$\|\nabla f(w)\| = \frac{1}{1+\exp(\frac{1}{2}w^\top \mathbf{A}_1 w)}\|\mathbf{A}_1 w\| = \frac{1}{1+\exp(\frac{\sqrt{2}}{2}(\alpha^2 - \beta^2))}\|\mathbf{A}_1(\alpha v_1 + \beta v_2)\|$$
$$= \frac{1}{1+\exp(\frac{\sqrt{2}}{2}(\alpha^2 - \beta^2))}\|\mathbf{A}_1(\alpha v_1 + \beta v_2)\| = \frac{\sqrt{2}}{1+\exp(\frac{\sqrt{2}}{2}(\alpha^2 - \beta^2))}\|\alpha v_1 - \beta v_2\|$$
$$= \frac{\sqrt{2}}{1+\exp(\frac{\sqrt{2}}{2}(\alpha^2 - \beta^2))}\sqrt{\alpha^2 + \beta^2} = \frac{\sqrt{2}\|w\|}{1+\exp(\frac{\sqrt{2}}{2}(\alpha^2 - \beta^2))}.$$

Since $\sqrt{\beta^2 + 1} \geq \alpha \geq \beta$,

$$\frac{\sqrt{2}}{20}\|w\| \leq \|\nabla f(w)\| \leq \sqrt{2}\|w\|.$$

Finally,

$$f(w) = \log\left(1 + \exp\left(-\frac{1}{2}w^\top \mathbf{A}_1 w\right)\right) = \log\left(1 + \exp\left(-\frac{\sqrt{2}}{2}(\alpha^2 - \beta^2)\right)\right)$$

and

$$\frac{1}{20} \leq f(w) \leq 5$$

since $\sqrt{\beta^2 + 1} \geq \alpha \geq \beta$. $\qquad\square$

## B.4. Proof of theorems and propositions

**Theorem 2.1** (Reduction to Quadratic Perceptron Algorithm). *Consider the steps* (3). *Assume that the direction* $\theta_0 := \frac{w_0}{\|w_0\|} \in \mathbb{R}^{d+2}$ *of the starting point* $w_0$ *is fixed,* $\theta_0^\top \mathbf{A}_i \theta_0 \neq 0$ *for all* $i \in [n]$, *and* $\|w_0\| \to \infty$. *For almost all choices*[4]*of* $\gamma < 1/\max_{i \in [n]} \|\mathbf{A}_i\|$ *and for* $\|w_0\| \to \infty$, $\frac{w_{t+1}}{\|w_{t+1}\|}$ *is well-defined and equals* $\frac{\theta_{t+1}}{\|\theta_{t+1}\|}$, *where*

$$\theta_{t+1} = \theta_t + \frac{\gamma}{n} \sum_{i \in S_t} \mathbf{A}_i \theta_t,$$

$$S_t = \left\{ i \in [n] : \frac{1}{2} \theta_t^\top \mathbf{A}_i \theta_t \leq 0 \right\} \qquad \text{(Quadratic Perceptron Algorithm)}$$

*Moreover, the predictions of* $w_t$ *and* $\theta_t$ *are equal, i.e., for* $\|w_0\| \to \infty$, $\mathrm{sign}(m_{\mathrm{cv}}(b_i; w_t)) = \mathrm{sign}(m_{\mathrm{cv}}(b_i; \theta_t))$ *for all* $i \in [n]$ *and* $t \geq 0$.

*Proof.* Using mathematical induction, we will prove that $\frac{w_t}{\|w_t\|} \overset{\|w_0\| \to \infty}{=} \frac{\theta_t}{\|\theta_t\|}$, $\|w_t\| \overset{\|w_0\| \to \infty}{\to} \infty$, and $\theta_t^\top \mathbf{A}_i \theta_t \neq 0$ for all $i \in [n]$ and $t \geq 0$. For $t = 0$, it holds by the assumption. Assume that it holds for $t \geq 0$, then

$$\frac{w_{t+1}}{\|w_{t+1}\|} = \frac{w_t + \gamma \frac{1}{n} \sum_{i=1}^n \frac{1}{1+\exp(\frac{1}{2} w_t^\top \mathbf{A}_i w_t)} \mathbf{A}_i w_t}{\left\| w_t + \gamma \frac{1}{n} \sum_{i=1}^n \frac{1}{1+\exp(\frac{1}{2} w_t^\top \mathbf{A}_i w_t)} \mathbf{A}_i w_t \right\|}$$

$$= \frac{\frac{w_t}{\|w_t\|} + \gamma \frac{1}{n} \sum_{i=1}^n \frac{1}{1+\exp(\frac{\|w_t\|^2}{2} \frac{w_t}{\|w_t\|}^\top \mathbf{A}_i \frac{w_t}{\|w_t\|})} \mathbf{A}_i \frac{w_t}{\|w_t\|}}{\left\| \frac{w_t}{\|w_t\|} + \gamma \frac{1}{n} \sum_{i=1}^n \frac{1}{1+\exp(\frac{\|w_t\|^2}{2} \frac{w_t}{\|w_t\|}^\top \mathbf{A}_i \frac{w_t}{\|w_t\|})} \mathbf{A}_i \frac{w_t}{\|w_t\|} \right\|}.$$

Using $\frac{w_t}{\|w_t\|} \overset{\|w_0\| \to \infty}{=} \frac{\theta_t}{\|\theta_t\|}$, $\theta_t^\top \mathbf{A}_i \theta_t \neq 0$ and $\frac{\theta_t}{\|\theta_t\|}^\top \mathbf{A}_i \frac{\theta_t}{\|\theta_t\|} \neq 0$ for all $i \in [n]$, and $\|w_t\| \overset{\|w_0\| \to \infty}{\to} \infty$, we have

$$\frac{1}{1 + \exp(\frac{\|w_t\|^2}{2} \frac{w_t}{\|w_t\|}^\top \mathbf{A}_i \frac{w_t}{\|w_t\|})} \overset{\|w_0\| \to \infty}{=} \mathbb{1} \left[ \frac{\theta_t}{\|\theta_t\|}^\top \mathbf{A}_i \frac{\theta_t}{\|\theta_t\|} < 0 \right] = \mathbb{1} \left[ \theta_t^\top \mathbf{A}_i \theta_t < 0 \right] = \mathbb{1} \left[ \theta_t^\top \mathbf{A}_i \theta_t \leq 0 \right].$$

Thus, we get

$$\frac{w_{t+1}}{\|w_{t+1}\|} \overset{\|w_0\| \to \infty}{=} \frac{\frac{\theta_t}{\|\theta_t\|} + \gamma \frac{1}{n} \sum_{i \in S_t} \mathbf{A}_i \frac{\theta_t}{\|\theta_t\|}}{\left\| \frac{\theta_t}{\|\theta_t\|} + \gamma \frac{1}{n} \sum_{i \in S_t} \mathbf{A}_i \frac{\theta_t}{\|\theta_t\|} \right\|} = \frac{\theta_t + \gamma \frac{1}{n} \sum_{i \in S_t} \mathbf{A}_i \theta_t}{\left\| \theta_t + \gamma \frac{1}{n} \sum_{i \in S_t} \mathbf{A}_i \theta_t \right\|}$$

Since $\gamma < \frac{1}{\max_{i \in [n]} \|\mathbf{A}_i\|}$, we have $\mathbf{I} + \gamma \frac{1}{n} \sum_{i \in S_t} \mathbf{A}_i \succ 0$ and $\left\| \theta_t + \gamma \frac{1}{n} \sum_{i \in S_t} \mathbf{A}_i \theta_t \right\| > 0$. Using the definition of $\theta_{t+1}$,

$$\frac{w_{t+1}}{\|w_{t+1}\|} \overset{\|w_0\| \to \infty}{=} \frac{\theta_{t+1}}{\|\theta_{t+1}\|}.$$

The norm

$$\|w_{t+1}\| = \|w_t\| \left\| \frac{w_t}{\|w_t\|} + \gamma \frac{1}{n} \sum_{i=1}^n \frac{1}{1 + \exp(\frac{\|w_t\|^2}{2} \frac{w_t}{\|w_t\|}^\top \mathbf{A}_i \frac{w_t}{\|w_t\|})} \mathbf{A}_i \frac{w_t}{\|w_t\|} \right\| \overset{\|w_0\| \to \infty}{=} \infty$$

because $\|w_t\| \to \infty$ and the second term converges to $\frac{\|\theta_{t+1}\|}{\|\theta_t\|} > 0$.

It left to consider $\theta_{t+1}^\top \mathbf{A}_i \theta_{t+1}$ Notice that $\theta_{t+1}^\top \mathbf{A}_i \theta_{t+1}$ is the polynomial function of $\gamma$. When $\gamma = 0$, $\theta_{t+1}^\top \mathbf{A}_i \theta_{t+1} = \theta_t^\top \mathbf{A}_i \theta_t \neq 0$ for all $i \in [n]$. Thus, $\theta_{t+1}^\top \mathbf{A}_i \theta_{t+1}$ is a *non-zero* polynomial function of $\gamma$ that equals zero with the particular choices of $\gamma$. The Lebesgue measure of such choices of $\gamma$ is zero. Thus, for almost all choices of $\gamma$, $\theta_{t+1}^\top \mathbf{A}_i \theta_{t+1} \neq 0$ for all $i \in [n]$. We have proved the next step of the mathematical induction.

It left to prove the last statement. For all $t \geq 0$, we know that $\|\theta_t\| \times \frac{w_t}{\|w_t\|} \overset{\|w_0\| \to \infty}{=} \theta_t$. Notice that

$$\mathrm{sign}(m_{\mathrm{cv}}(b_i; w_t)) = \mathrm{sign}((c_t * b_i)^\top v_t) = \mathrm{sign} \left( \frac{\|\theta_t\|^2}{\|w_t\|^2} (c_t * b_i)^\top v_t \right)$$

$$= \text{sign}\left(\left(\frac{\|\theta_t\| \times c_t}{\|w_t\|} * b_i\right)^\top \frac{\|\theta_t\| \times v_t}{\|w_t\|}\right) = \text{sign}\left(m_{\text{cv}}\left(b_i; \|\theta_t\| \times \frac{w_t}{\|w_t\|}\right)\right).$$

Since

$$0 \neq \frac{1}{2}\theta_t^\top \mathbf{A}_i \theta_t \overset{(4)}{=} y_i \left(\bar{c}_t * b_i\right)^\top \bar{v}_t = y_i m_{\text{cv}}(b_i; \theta_t),$$

for all $i \in [n]$ and $t \geq 0$, where $\theta_t \equiv \begin{bmatrix} \bar{c}_t^\top & \bar{v}_t^\top \end{bmatrix}^\top$, we can conclude that $0 \neq m_{\text{cv}}(b_i; \theta_t)$. Therefore, due to $\|\theta_t\| \times \frac{w_t}{\|w_t\|} \overset{\|w_0\| \to \infty}{=} \theta_t$, for large enough $\|w_0\|$, $m_{\text{cv}}\left(b_i; \|\theta_t\| \times \frac{w_t}{\|w_t\|}\right) \neq 0$ and $\text{sign}(m_{\text{cv}}(b_i; w_t)) = \text{sign}\left(m_{\text{cv}}\left(b_i; \|\theta_t\| \times \frac{w_t}{\|w_t\|}\right)\right) = \text{sign}\left(m_{\text{cv}}\left(b_i; \theta_t\right)\right).$ $\qquad \square$

**Proposition 4.1.** *Consider $a \in \mathbb{R}^d$ and*

$$\mathbf{A} = \begin{bmatrix} 0 & 0 & a^\top \\ 0 & 0 & a^\top \mathbf{P}^\top \\ a & \mathbf{P}a & \mathbf{0}_d \end{bmatrix} \in \mathbb{R}^{(d+2)\times(d+2)}.$$

*1. If $a = 0$, then $\mathbf{A}$ is the zero matrix with $d + 2$ zero eigenvalues.*

*2. If $a \neq 0$ and $a = \mathbf{P}a$, then $\mathbf{A}$ has two non-zero eigenvalues $\sqrt{2}\|a\|$ and $-\sqrt{2}\|a\|$ with the corresponding eigenvectors $v_1 = \begin{bmatrix} \|a\| & \|a\| & \sqrt{2}a^\top \end{bmatrix}^\top$ and $v_2 = \begin{bmatrix} -\|a\| & -\|a\| & \sqrt{2}a^\top \end{bmatrix}^\top.$*

*3. If $a \neq 0$ and $a = -\mathbf{P}a$, then $\mathbf{A}$ has two non-zero eigenvalues $\sqrt{2}\|a\|$ and $-\sqrt{2}\|a\|$ with the corresponding eigenvectors $v_1 = \begin{bmatrix} \|a\| & -\|a\| & \sqrt{2}a^\top \end{bmatrix}^\top$ and $v_2 = \begin{bmatrix} -\|a\| & \|a\| & \sqrt{2}a^\top \end{bmatrix}^\top.$*

*4. If $a \neq 0$, $a \neq \mathbf{P}a$, and $a \neq -\mathbf{P}a$, then $\mathbf{A}$ has four non-zero eigenvalues $\sqrt{\|a\|^2 + a^\top \mathbf{P}a}$, $-\sqrt{\|a\|^2 + a^\top \mathbf{P}a}$, $\sqrt{\|a\|^2 - a^\top \mathbf{P}a}$, $-\sqrt{\|a\|^2 - a^\top \mathbf{P}a}$ with the corresponding eigenvectors*
$v_1 = \begin{bmatrix} \bar{a}_{+,\mathbf{P}} & \bar{a}_{+,\mathbf{P}} & (a + \mathbf{P}a)^\top \end{bmatrix}^\top,$
$v_2 = \begin{bmatrix} -\bar{a}_{+,\mathbf{P}} & -\bar{a}_{+,\mathbf{P}} & (a + \mathbf{P}a)^\top \end{bmatrix}^\top,$
$v_3 = \begin{bmatrix} \bar{a}_{-,\mathbf{P}} & -\bar{a}_{-,\mathbf{P}} & (a - \mathbf{P}a)^\top \end{bmatrix}^\top,$
*and* $v_4 = \begin{bmatrix} -\bar{a}_{-,\mathbf{P}} & \bar{a}_{-,\mathbf{P}} & (a - \mathbf{P}a)^\top \end{bmatrix}^\top,$
*where $\bar{a}_{+,\mathbf{P}} := \sqrt{\|a\|^2 + a^\top \mathbf{P}a}$ and $\bar{a}_{-,\mathbf{P}} := \sqrt{\|a\|^2 - a^\top \mathbf{P}a}.$*

*5. $\|a\| \leq \|\mathbf{A}\| \leq \sqrt{2}\|a\|.$*

*Proof.* Note that $\mathbf{A}$ is a matrix with the rank less or equal 4. Thus, it has at least $d - 2$ zero eigenvalues. Next, if $a = \mathbf{P}a$, then $\mathbf{A}$ is a matrix with the rank less or equal 2. Finally, if $a = 0$, then $\mathbf{A}$ is the zero matrix with $d + 2$ zero eigenvalues. Assume that $a \neq 0$ and $a = a\mathbf{P}$, then

$$\mathbf{A} = \begin{bmatrix} 0 & 0 & a^\top \\ 0 & 0 & a^\top \\ a & a & \mathbf{0}_d \end{bmatrix}. \tag{12}$$

By the definition, one can easily show that this matrix has two non-zero eigenvalues $\sqrt{2}\|a\|$ and $-\sqrt{2}\|a\|$ with the corresponding eigenvectors from Proposition 4.1. The case $a = -a\mathbf{P}$ can be analyzed in the same way. Assume that $a \neq \mathbf{P}a$ and $a \neq -\mathbf{P}a$, then the matrix has four non-zero eigenvalues

$$\sqrt{\|a\|^2 + a^\top \mathbf{P}a}, -\sqrt{\|a\|^2 + a^\top \mathbf{P}a}, \sqrt{\|a\|^2 - a^\top \mathbf{P}a}, -\sqrt{\|a\|^2 - a^\top \mathbf{P}a}.$$

with the corresponding eigenvectors from Proposition 4.1. Using the definition of eigenvalues and the property $\|\mathbf{P}x\| = \|x\|$ for all $x \in \mathbb{R}^d$, one can easily check it. Clearly, for all cases we have $\|a\| \leq \|\mathbf{A}\| \leq \sqrt{2}\|a\|$ since $|a^\top \mathbf{P}a| \leq \|a\|^2.$ $\qquad \square$

**Theorem 4.2.** *For any dataset with the number of samples $n < (d+2)/4$, there exists a subspace $V$ of at least dimension $(d+2) - 4n$ from which Quadratic Perceptron Algorithm does not converge.*

*Proof.* Using Proposition 4.1, and the rank–nullity theorem, for each sample $i$, $\dim(\mathrm{im}(\mathbf{A}_i)) = (d+2) - \dim(\ker(\mathbf{A}_i)) \leq 4$, where $\ker(\mathbf{A}_i)$ and $\mathrm{im}(\mathbf{A}_i)$ are the kernel and image of $\mathbf{A}_i$. Thus, $\dim(\sum_{i=1}^n \mathrm{im}(\mathbf{A}_i)) \leq \sum_{i=1}^n \dim(\mathrm{im}(\mathbf{A}_i)) \leq 4n$ and $0 < (d+2) - 4n \leq \dim((\sum_{i=1}^n \mathrm{im}(\mathbf{A}_i))^\perp) = \dim(\cap_{i=1}^n \mathrm{im}(\mathbf{A}_i)^\perp) = \dim(\cap_{i=1}^n \ker(\mathbf{A}_i))$, meaning that we can take $V = \cap_{i=1}^n \ker(\mathbf{A}_i)$. For all $\theta_0 \in \cap_{i=1}^n \ker(\mathbf{A}_i)$, $\mathbf{A}_i \theta_0 = 0$ for all $i \in [n]$, and the method does not move and does not find a separator since $\theta_0 = \theta_1 = \theta_2$, and so forth. $\qquad\square$

**Theorem B.1.** *For all $a \in \mathbb{R}^d$, consider matrix*

$$\mathbf{A} := \begin{bmatrix} 0 & 0 & \ldots & 0 & a^\top \\ \vdots & \vdots & \ddots & \vdots & a^\top \mathbf{P}^\top \\ \vdots & \vdots & \ddots & \vdots & \vdots \\ 0 & 0 & \ldots & 0 & a^\top (\mathbf{P}^{k-1})^\top \\ a & \mathbf{P}a & \ldots & \mathbf{P}^{k-1}a & \mathbf{0}_d \end{bmatrix} \in \mathbb{R}^{(d+k)\times(d+k)}. \tag{13}$$

*Then, $\frac{1}{\sqrt{k}} \left\| \sum_{j=1}^k \mathbf{P}^{j-1}a \right\| \leq \|\mathbf{A}\| \leq \sqrt{k}\,\|a\|$.*

*Proof.* For all $x \equiv [x_1^\top, x_2^\top]^\top \in \mathbb{R}^{d+k}$, $x_1 \in \mathbb{R}^k$, and $x_2 \in \mathbb{R}^d$,

$$\mathbf{A}x = \left[ a^\top x_2, (\mathbf{P}a)^\top x_2, \ldots, (\mathbf{P}^{k-1}a)^\top x_2, \sum_{j=1}^k \mathbf{P}^{j-1}a[x_1]_j \right]^\top.$$

Thus,

$$\|\mathbf{A}x\|^2 = \sum_{j=1}^k \left( (\mathbf{P}^{j-1}a)^\top x_2 \right)^2 + \left\| \sum_{j=1}^k \mathbf{P}^{j-1}a[x_1]_j \right\|^2 \tag{14}$$

$$\leq \sum_{j=1}^k \left\| \mathbf{P}^{j-1}a \right\|^2 \|x_2\|^2 + \left\| \sum_{j=1}^k \mathbf{P}^{j-1}a[x_1]_j \right\|^2,$$

Since $\mathbf{P}^\top \mathbf{P} = \mathbf{I}$,

$$\|\mathbf{A}x\|^2 \leq k\,\|a\|^2 \|x_2\|^2 + \left\| \sum_{j=1}^k \mathbf{P}^{j-1}a[x_1]_j \right\|^2.$$

Using Jensen's inequality,

$$\|\mathbf{A}x\|^2 \leq k\,\|a\|^2 \|x_2\|^2 + \sum_{j=1}^k [x_1]_j^2 \sum_{j=1}^k \left\| \mathbf{P}^{j-1}a \right\|^2 = k\,\|a\|^2 \|x_2\|^2 + k\,\|a\|^2 \|x_1\|^2 = k\,\|a\|^2 \|x\|^2$$

for all $x \in \mathbb{R}^{d+k}$, and we can conclude that $\|\mathbf{A}\| \leq \sqrt{k}\,\|a\|$. On the other hand, using (14),

$$\|\mathbf{A}x\|^2 \geq \left\| \sum_{j=1}^k \mathbf{P}^{j-1}a[x_1]_j \right\|^2.$$

Taking $x_2 = 0$ and $[x_1]_j = \frac{1}{\sqrt{k}}$ for all $j \in [k]$, $\|x\| = 1$ and

$$\|\mathbf{A}x\|^2 \geq \frac{1}{k} \left\| \sum_{j=1}^k \mathbf{P}^{j-1}a \right\|^2.$$

In total,

$$\frac{1}{\sqrt{k}} \left\| \sum_{j=1}^{k} \mathbf{P}^{j-1} a \right\| \leq \|\mathbf{A}\| \leq \sqrt{k} \, \|a\| \, .$$

$\square$

**Proposition 6.1.** *Consider matrix $\mathbf{A}_i$ from* (8). *Then, the norm of $\mathbf{A}_i$ can be bounded as $\frac{1}{\sqrt{k}} \left\| \sum_{j=1}^{k} \mathbf{P}^{j-1} b_i \right\| \leq \|\mathbf{A}_i\| \leq \sqrt{k} \, \|b_i\| \, .*$

*Proof.* This is a corollary of Theorem B.1 $\hfill\square$

**Proposition 5.3** (follows from Proposition 4.1)**.** *Consider the matrices $\mathbf{A}_1$ and $\mathbf{A}_2$ of the dataset* (5)*.*

*1. $\mathbf{A}_1$ has two non-zero eigenvalues $\lambda_1$ and $-\lambda_1$ with the corresponding eigenvectors $v_{1,+}$ and $v_{1,-}$, where $\lambda_1 := \sqrt{2d}$.*

*2. $\mathbf{A}_2$ has four non-zero eigenvalues $\lambda_2, -\lambda_2, \mu, -\mu$ with the corresponding eigenvectors $v_{2,+}, v_{2,-}, v_{\mu,+}$, and $v_{\mu,-}$, where $\lambda_2 := \sqrt{(2+\mu)^2 + 2(d-2)}$.*

*3. $v_{\mu,+}, v_{\mu,-} \in \ker(\mathbf{A}_1)$.*

*Proof.* The first and second result are simple corollaries of Proposition 4.1. The third result follows from that $v_{\mu,+}$ and $v_{\mu,-}$ are propositional to

$$\begin{bmatrix} -\mu & \mu & \mu & -\mu & 0 & \dots & 0 \end{bmatrix}^{\top},$$
$$\begin{bmatrix} \mu & -\mu & \mu & -\mu & 0 & \dots & 0 \end{bmatrix}^{\top}.$$

Direct calculations yield

$$\mathbf{A}_1 v_{\mu,+} = 0$$

and

$$\mathbf{A}_1 v_{\mu,-} = 0.$$

$\square$

# C. Proof of Theorem 5.1

In this section, we present a formal proof of Theorem 5.1. Before doing so, clarifying Section 5.1, we briefly outline the main steps of the proofs.

## C.1. A more detailed proof sketch

At the beginning, we fix the maximal number of steps $T$, step size $\gamma$, and variance $\sigma$ to particular values (which are inferred from the proof).

Next, we fix the event $\Omega_T$ in Lemma C.4 such that

$$\|\xi_t\| \leq \beta := \sigma \sqrt{\max\{6d, 4\log(1/\delta)\}}$$

and

$$\left| \eta_{\lambda,s,t,v} + \sum_{i=1}^{s} (1+\gamma\lambda)^{i-1} \langle \xi_{t+s-i}, v \rangle \right| > \alpha_s := \alpha(1+\gamma\lambda)^{s/2}, \alpha := \frac{\sigma\delta}{20\sqrt{7}\log\left(\frac{1}{\gamma\lambda_1}\right)}, \tag{15}$$

where $\eta_{\lambda,s,t,v} := (1+\gamma\lambda)^s \langle \theta_t, v \rangle$, for all finite choices of $\lambda$ and $v$, and show that the probability of $\Omega_T$ is large. Starting from this point, the proof is deterministic in nature, and we rely only on these inequalities when we work with $\{\xi_t\}$.

**(Number of steps of option 1).** The algorithm either does the update $\theta_t + \gamma \mathbf{A}_1 \theta_t + \xi_t$ (option 1) or $\theta_t + \gamma \mathbf{A}_2 \theta_t + \xi_t$ (option 2), which interleave and divide the steps into groups of sizes $\{s_i\}$. Let $s_k$ be a group with option 1, and $t$ be the iteration where this group starts. Using Proposition 5.3,

$$\langle \theta_{t+s}, \mathbf{A}_1 \theta_{t+s} \rangle = \lambda_1 \langle \theta_{t+s}, v_{1,+} \rangle^2 - \lambda_1 \langle \theta_{t+s}, v_{1,-} \rangle^2. \tag{16}$$

for all $s \geq 0$. Since we apply $\theta_t + \gamma \mathbf{A}_1 \theta_t + \xi_t$ (option 1) $s$ times,

$$\langle \theta_{t+s}, v_{1,+} \rangle^2 = \left( (1 + \gamma \lambda_1)^s \langle \theta_t, v_{1,+} \rangle + \sum_{i=1}^{s} (1 + \gamma \lambda_1)^{i-1} \langle \xi_{t+s-i}, v_{1,+} \rangle \right)^2.$$

Next, we use (15) to ensure that $\langle \theta_{t+s}, v_{1,+} \rangle^2 \geq \alpha^2 (1 + \gamma \lambda_1)^s$. We now know that the first term in (16) grows exponentially.

The next technical step is to prove that $\langle \theta_{t+s}, v_{1,-} \rangle^2$ does not grow faster in order to ensure that (16) $> 0$ after $s = \tilde{\mathcal{O}}(1)$ steps. Clearly, $\langle \theta_{t+s}, v_{1,-} \rangle^2 \leq \|\theta_{t+s}\|^2$. Since $\theta_{t+1} = \theta_t + \gamma \mathbf{A}_1 \theta_t + \xi_t$ and $\|\xi_t\| \leq \beta$,

$$\|\theta_{t+s}\|^2 \lesssim \|(\mathbf{I} + \gamma \mathbf{A}_1) \theta_{t+s-1}\|^2 + \beta^2.$$

In the worst case, $\|\mathbf{I} + \gamma \mathbf{A}_1\| \simeq 1 + \gamma \lambda_1$ and $\|(\mathbf{I} + \gamma \mathbf{A}_1) \theta\| \simeq (1 + \gamma \lambda_1) \|\theta\|$, which is non-tight for our goals. However, we know that $\langle \theta_{t+s}, \mathbf{A}_1 \theta_{t+s} \rangle \leq 0$, due to the condition 1 of the algorithm, which allows to show a much tighter inequality $\|\theta_{t+s}\|^2 \lesssim \|(\mathbf{I} + \gamma \mathbf{A}_1) \theta_{t+s-1}\|^2 + \beta^2 \lesssim (1 + \gamma \mu) \|\theta_{t+s-1}\|^2$ for $\beta$ small enough (we get $(1 + \gamma \mu)$ instead of $(1 + \gamma \lambda_1)$). In general, we can show that $\|\theta_t\|^2 \lesssim \left(1 + \frac{93 \gamma \mu}{50}\right)^{t-s} \max\{\|\theta_s\|^2, \|\theta_0\|^2\}$.

Returning back to (16), we have shown that

$$\langle \theta_{t+s}, \mathbf{A}_1 \theta_{t+s} \rangle \geq \lambda_1 \alpha^2 (1 + \gamma \lambda_1)^s - \lambda_1 256 \left(1 + \frac{93 \gamma \mu}{50}\right)^s \max\{\|\theta_t\|^2, \|\theta_0\|^2\}.$$

Thus, the second term also grows exponentially, but much slower, and we can conclude that the number of consecutive calls of the option with $\mathbf{A}_1$ is less or equal to

$$s_k \leq 1 + \frac{4}{\gamma \lambda_1} \log \left( \frac{512 \max\{\|\theta_t\|^2, \|\theta_0\|^2\}}{\alpha^2} \right) \tag{17}$$

**(Number of steps of option 2).** The case with $\mathbf{A}_2$ can be shown similarly.

**(Finite "ping-pong" between options 1 and 2).** Now we know that $s_k$ satisfies (17), and we observe a "ping-pong" effect between options 1 and 2. Fortunately, matrix $\mathbf{A}_2$ has a special direction such that $\langle v_{\mu,+}, \theta_t \rangle^2$ increases exponentially when option 2 is executed and remains unchanged (up to random noise) when option 1 is executed. By choosing $\sigma$ sufficiently small,

$$\langle \theta_{t_{j+1}}, v_{\mu,+} \rangle^2 > \frac{1}{4} (1 + 2 \gamma \mu)^{\bar{t}_{j+1}} \langle \theta_0, v_{\mu,+} \rangle^2, \tag{18}$$

where $t_j := \sum_{i=1}^{j-1} s_i$ is the total number of steps before group $j$ of size $s_j$, and $\bar{t}_j := \sum_{i \in [j-1] \text{ s.t. } q_i = 2} s_i$ is the number of steps with option 2 before group $j$ of size $s_j$. In other words, (18) formalizes the fact that $\langle \theta_{t_{j+1}}, v_{\mu,+} \rangle^2$ increases exponentially when only option 2 is chosen.

Since

$$\langle \theta_t, \mathbf{A}_2 \theta_t \rangle = \lambda_2 \langle \theta_t, v_{2,+} \rangle^2 - \lambda_2 \langle \theta_t, v_{2,-} \rangle^2 + \mu \langle \theta_t, v_{\mu,+} \rangle^2 - \mu \langle \theta_t, v_{\mu,-} \rangle^2,$$

we can conclude that

$$\langle \theta_{t_j}, \mathbf{A}_2 \theta_{t_j} \rangle > \frac{\mu}{4} (1 + 2 \gamma \mu)^{\bar{t}_j} \langle \theta_0, v_{\mu,+} \rangle^2 - (\lambda_2 + \mu) \|\theta_t\|^2.$$

Using the structure of $\{\mathbf{A}_i\}$ and the conditions that either $\theta_t^\top \mathbf{A}_1 \theta_t \leq 0$ or $\theta_t^\top \mathbf{A}_2 \theta_t \leq 0$, we can prove that

$$\left\| \theta_{t_j} \right\|^2 \leq 1024 \left( 1 + \frac{93\gamma\mu}{50} \right)^{\bar{t}_j} \| \theta_0 \|^2. \tag{19}$$

In other words, the norm of $\theta_t$ increases exponentially proportionally to the number of option 2 calls, and almost does not increase when option 1 is called.

Thus,

$$\langle \theta_{t_j}, \mathbf{A}_2 \theta_{t_j} \rangle > \frac{\mu}{4}(1 + 2\gamma\mu)^{\bar{t}_j} \langle \theta_0, v_{\mu,+} \rangle^2 - 1024(\lambda_2 + \mu) \left( 1 + \frac{93\gamma\mu}{50} \right)^{\bar{t}_j} \| \theta_0 \|^2.$$

The second term may grow exponentially, but not as fast as the first term. Therefore, the maximum number of times option 2 can be called is $\tilde{\mathcal{O}}\left( 1/\gamma\mu \right)$ and $\bar{t}_j = \tilde{\mathcal{O}}\left( 1/\gamma\mu \right)$. Combining (17) and (19),

$$s_j = \tilde{\mathcal{O}} \left( \frac{\mu}{\lambda_1} \bar{t}_j \right) = \tilde{\mathcal{O}} \left( \frac{1}{\gamma\lambda_1} \right) = \tilde{\mathcal{O}}\left( 1 \right)$$

since $\gamma = \Theta(1/\lambda_1)$. Since options 1 and 2 interleave, the size of each group is $\tilde{\mathcal{O}}\left( 1 \right)$, and option 2 can be chosen at most $\tilde{\mathcal{O}}\left( 1/\gamma\mu \right)$ times, we can conclude that the total number of steps is also $\tilde{\mathcal{O}}\left( 1/\gamma\mu \right) = \tilde{\mathcal{O}}\left( \sqrt{d}/\mu \right)$. A formal version of the proof is provided below.

## C.2. Full proof

**Theorem 5.1** (Upper Bound for Two-Sample Quadratic Perceptron Algorithm). *Consider Two-Sample Quadratic Perceptron Algorithm on the dataset* (5). *With probability greater than or equal to* $1 - \rho$, *the required number of steps to find a separator is at most*

$$T = \left\lceil B \times \frac{\sqrt{d}}{\mu} \right\rceil = \tilde{\Theta}\left( \frac{\sqrt{d}}{\mu} \right),$$

*where* $B := A \log \left( \frac{\sqrt{d}}{\mu} A \right) = \tilde{\Theta}\left( 1 \right)$, $A := 2^{27} \log \left( \frac{2^{62} \| \theta_0 \|^2}{\rho^3 \langle \theta_0, v_{\mu,+} \rangle^2} \right) \log \left( \frac{2^{15} \sqrt{d} \| \theta_0 \|^2}{\mu \langle \theta_0, v_{\mu,+} \rangle^2} \right) = \tilde{\Theta}\left( 1 \right)$, *if* $\gamma = \frac{1}{4\sqrt{d}}$, $\mu \leq \frac{1}{10}$, $\sigma = |\langle \theta_0, v_{\mu,+} \rangle| / (4096 T \sqrt{\max\{d, \log(T/\rho)\}})$, *and* $\langle \theta_0, v_{\mu,+} \rangle \neq 0$.

*Proof.* In the theorem, we show it is sufficient to take the number of iterations equal to

$$T = \left\lceil B \times \frac{\sqrt{d}}{\mu} \right\rceil = \tilde{\Theta}\left( \frac{\sqrt{d}}{\mu} \right),$$

where $B := A \log \left( \frac{\sqrt{d}}{\mu} A \right) = \tilde{\Theta}\left( 1 \right)$, $A := 2^{27} \log \left( \frac{2^{62} \| \theta_0 \|^2}{\rho^3 \langle \theta_0, v_{\mu,+} \rangle^2} \right) \log \left( \frac{2^{15} \sqrt{d} \| \theta_0 \|^2}{\mu \langle \theta_0, v_{\mu,+} \rangle^2} \right) = \tilde{\Theta}\left( 1 \right)$. Next, we consider set $\Omega_T$ defined in Lemma C.4. All our following results are true on the set $\Omega_T$ with probability $\mathbb{P}\left( \Omega_T \right) \geq 1 - 5T\delta$. Taking $\delta = \rho/5T$, we ensure the the following steps and the theorem hold with probability greater than or equal to $1 - \rho$.

Next, in the theorem we take $\sigma$ such that

$$
\begin{aligned}
\beta &:= \sigma \sqrt{\max\{6d, 4\log(1/\delta)\}} \\
&\leq \frac{|\langle \theta_0, v_{\mu,+} \rangle|}{1500 T} \\
&\leq \min \left\{ \frac{|\langle \theta_0, v_{\mu,+} \rangle|}{2T}, \frac{1}{64} \min \left\{ \frac{1}{T}, \gamma\mu \right\} \| \theta_0 \|, \frac{1}{128} \gamma\mu \| \theta_0 \|, \| \theta_0 \|, \min \left\{ \frac{\mu \| \theta_0 \|}{1050\lambda_2}, \frac{\gamma\mu \| \theta_0 \|}{1050} \right\}, \gamma\lambda_2 \| \theta_0 \| \right\},
\end{aligned} \tag{20}
$$

where $\beta$ is defined in Lemma C.4. In the inequality, we use that $\gamma \leq \frac{1}{4\sqrt{d}}$, $\lambda_2 \leq 4\sqrt{d}$, $\lambda_2 \geq \mu$, and the choice of $T$. We will use this bound in theorem and lemmas. Once all the parameters are fixed or bounded, we are ready to proceed to the proof.

When we start Two-Sample Quadratic Perceptron Algorithm, it runs through the sequence of steps: $i_{1,1}, \ldots, i_{1,s_1}, i_{2,1}, \ldots, i_{2,s_2} \ldots$. and the corresponding size of groups $s_1, s_2, \ldots$, where $i_{i,j} \in \{1, 2, 3\}$, and 1 corresponds to the option with matrix $\mathbf{A}_1$, 2 corresponds to the option with matrix $\mathbf{A}_2$, and 3 corresponds to the "brake" option

when the algorithm stops. Moreover, we define $q_1, q_2, \ldots$, where $q_i \in \{1, 2, 3\}$, as the type of options of the corresponding group.

If option 3 occurs before iteration $T$, then the result is true. In the proof, we will assume that only options 1 and 2 are possible in the first $T$ iterations and will prove a contradiction.

Our first goal is to show that each $s_k$ is bounded for the groups with $\{1, 2\}$. For all $k \geq 1$, let $i_{k,1} = 1$ and $t$ be the corresponding iteration. Using Proposition 5.3,

$$\langle \theta_{t+s}, \mathbf{A}_1 \theta_{t+s} \rangle = \lambda_1 \langle \theta_{t+s}, v_{1,+} \rangle^2 - \lambda_1 \langle \theta_{t+s}, v_{1,-} \rangle^2. \tag{21}$$

for all $t \geq 0$ and $s \geq 0$. Since $i_{k,1} = 1$,

$$\langle \theta_{t+1}, v_{1,+} \rangle^2 = (\langle (\mathbf{I} + \gamma \mathbf{A}_1) \theta_t + \xi_t, v_{1,+} \rangle)^2 = (\langle \theta_t, (\mathbf{I} + \gamma \mathbf{A}_1) v_{1,+} \rangle + \langle \xi_t, v_{1,+} \rangle)^2$$
$$= ((1 + \gamma \lambda_1) \langle \theta_t, v_{1,+} \rangle + \langle \xi_t, v_{1,+} \rangle)^2.$$

Repeating these steps $s - 1$ more times,

$$\langle \theta_{t+s}, v_{1,+} \rangle^2 = \left( (1 + \gamma \lambda_1)^s \langle \theta_t, v_{1,+} \rangle + \sum_{i=1}^{s} (1 + \gamma \lambda_1)^{i-1} \langle \xi_{t+s-i}, v_{1,+} \rangle \right)^2.$$

Due to Lemma C.4 with $\eta_{\lambda_1, s, t, v_{1,+}} = (1 + \gamma \lambda_1)^s \langle \theta_t, v_{1,+} \rangle$,

$$\langle \theta_{t+s}, v_{1,+} \rangle^2 \geq \alpha^2 (1 + \gamma \lambda_1)^s$$

for all $s \geq 1$ and $t \leq T - 1$. At the same time, due to Lemma C.2,

$$\langle \theta_{t+s}, v_{1,-} \rangle^2 \leq \|\theta_{t+s}\|^2 \leq 256 \left( 1 + \frac{93\gamma\mu}{50} \right)^s \max\{\|\theta_t\|^2, \|\theta_0\|^2\}.$$

Thus, due to (21),

$$\langle \theta_{t+s}, \mathbf{A}_1 \theta_{t+s} \rangle \geq \lambda_1 \alpha^2 (1 + \gamma \lambda_1)^s - \lambda_1 256 \left( 1 + \frac{93\gamma\mu}{50} \right)^s \max\{\|\theta_t\|^2, \|\theta_0\|^2\}$$
$$> \lambda_1 \alpha^2 \exp\left( \frac{\gamma \lambda_1 s}{2} \right) - \lambda_1 256 \exp\left( 2\gamma\mu s \right) \max\{\|\theta_t\|^2, \|\theta_0\|^2\},$$

meaning that $\langle \theta_{t+s}, \mathbf{A}_1 \theta_{t+s} \rangle > 0$ for all $s \geq 1$ such that $\frac{\gamma \lambda_1 s}{2} - 2\gamma\mu s \geq \log\left( \frac{256 \|\theta_t\|^2}{\alpha^2} \right)$, and for all $s \geq 1$ such that

$$s \geq \frac{4}{\gamma \lambda_1} \log\left( \frac{256 \max\{\|\theta_t\|^2, \|\theta_0\|^2\}}{\alpha^2} \right).$$

since $\mu \leq \lambda_1/10$. In this way, the number of consecutive calls of the option with $\mathbf{A}_1$ is less or equal to

$$1 + \frac{4}{\gamma \lambda_1} \log\left( \frac{256 \max\{\|\theta_t\|^2, \|\theta_0\|^2\}}{\alpha^2} \right),$$

for all $t \leq T - 1$, where $t$ is the number of previous iterations. Similarly, let $i_{k,1} = 2$ and $t$ be the corresponding iteration. Using Proposition 5.3,

$$\langle \theta_{t+s}, \mathbf{A}_2 \theta_{t+s} \rangle = \lambda_2 \langle \theta_{t+s}, v_{2,+} \rangle^2 - \lambda_2 \langle \theta_{t+s}, v_{2,-} \rangle^2 + \mu \langle \theta_{t+s}, v_{\mu,+} \rangle^2 - \mu \langle \theta_{t+s}, v_{\mu,-} \rangle^2 \tag{22}$$
$$\geq \lambda_2 \langle \theta_{t+s}, v_{2,+} \rangle^2 - (\lambda_2 + \mu) \|\theta_{t+s}\|^2. \tag{23}$$

Notice that

$$\langle \theta_{t+s}, v_{2,+} \rangle^2 = \left( (1 + \gamma \lambda_2)^s \langle \theta_t, v_{2,+} \rangle + \sum_{i=1}^{s} (1 + \gamma \lambda_2)^{i-1} \langle \xi_{t+s-i}, v_{2,+} \rangle \right)^2.$$

Using Lemmas C.4 and C.2 and $\lambda_2 \geq \mu$,

$$\langle \theta_{t+s}, \mathbf{A}_2\theta_{t+s}\rangle \geq \lambda_2\alpha^2(1+\gamma\lambda_2)^s - 512\lambda_2\left(1 + \frac{93\gamma\mu}{50}\right)^s \max\{\|\theta_t\|^2, \|\theta_0\|^2\}.$$

Thus, $\langle \theta_{t+s}, \mathbf{A}_2\theta_{t+s}\rangle > 0$ for all

$$s \geq \frac{4}{\gamma\lambda_2}\log\left(\frac{512\max\{\|\theta_t\|^2, \|\theta_0\|^2\}}{\alpha^2}\right).$$

In total, since $\lambda_1 \leq \lambda_2$,

$$s_i \leq 1 + \frac{4}{\gamma\lambda_1}\log\left(\frac{512\max\{\|\theta_{t_i}\|^2, \|\theta_0\|^2\}}{\alpha^2}\right) \tag{24}$$

for all $i \geq 1$ such that $t_i \leq T - 1$, where $t_i := \sum_{j=1}^{i-1} s_j$ is the number of iterations before the $i^{\text{th}}$ group.

Next, let $q_i = 1$, then $v_{\mu,+} \in \ker\mathbf{A}_1$ (Proposition 5.3) and

$$\langle \theta_{t+s_i}, v_{\mu,+}\rangle = \langle (\mathbf{I} + \gamma\mathbf{A}_1)\theta_{t+s_i-1}, v_{\mu,+}\rangle + \langle \xi_{t+s_i-1}, v_{\mu,+}\rangle$$

$$= \langle \theta_{t+s_i-1}, v_{\mu,+}\rangle + \langle \xi_{t+s_i-1}, v_{\mu,+}\rangle = \langle \theta_t, v_{\mu,+}\rangle + \sum_{j=1}^{s_i}\langle \xi_{t+s_i-j}, v_{\mu,+}\rangle.$$

for all $t \geq 0$. On the other hand, if $q_i = 2$, using Proposition 5.3,

$$\langle \theta_{t+s_i}, v_{\mu,+}\rangle = \langle (\mathbf{I} + \gamma\mathbf{A}_2)\theta_{t+s_i-1}, v_{\mu,+}\rangle + \langle \xi_{t+s_i-1}, v_{\mu,+}\rangle$$

$$= (1+\gamma\mu)\langle \theta_{t+s_i-1}, v_{\mu,+}\rangle + \langle \xi_{t+s_i-1}, v_{\mu,+}\rangle$$

$$= (1+\gamma\mu)^{s_i}\langle \theta_t, v_{\mu,+}\rangle + \sum_{j=1}^{s_i}(1+\gamma\mu)^{j-1}\langle \xi_{t+s_i-j}, v_{\mu,+}\rangle$$

for all $t \geq 0$. Unrolling the recursion,

$$\langle \theta_{t_{j+1}}, v_{\mu,+}\rangle = \left\langle \theta_{\sum_{i=1}^j s_i}, v_{\mu,+}\right\rangle$$

$$= (1+\gamma\mu)^{\bar{t}_{j+1}}\langle \theta_0, v_{\mu,+}\rangle + \sum_{i=1}^{j}\sum_{p=1}^{s_i}(1+\gamma\mu)^{\bar{t}_{j+1}-\bar{t}_{i+1}+(s_i-p)\cdot\mathbf{1}[q_i=2]}\langle \xi_{t_i+p-1}, v_{\mu,+}\rangle,$$

where $\bar{t}_{j+1} := \sum_{i\in[j]\text{ s.t. } q_i=2} s_i$. Due to Lemma C.4, if $t_{j+1} \leq T - 1$, then

$$\left|\sum_{i=1}^{j}\sum_{p=1}^{s_i}(1+\gamma\mu)^{\bar{t}_{j+1}-\bar{t}_{i+1}+(s_i-p)\cdot\mathbf{1}[q_i=2]}\langle \xi_{t_i+p-1}, v_{\mu,+}\rangle\right|$$

$$\leq \beta\sum_{i=1}^{j}\sum_{p=1}^{s_i}(1+\gamma\mu)^{\bar{t}_{j+1}-\bar{t}_{i+1}+(s_i-p)\cdot\mathbf{1}[q_i=2]}$$

$$\leq \beta(1+\gamma\mu)^{\bar{t}_{j+1}}t_{j+1} \leq \beta(1+\gamma\mu)^{\bar{t}_{j+1}}T$$

since $-\bar{t}_{i+1} + (s_i - p)\cdot\mathbf{1}[q_i=2] = -\sum_{j\in[i]\text{ s.t. } q_j=2} s_j + (s_i - p)\cdot\mathbf{1}[q_i=2] \leq 0$ for all $i \in [j]$ and $p \in [s_i]$, and $t_{j+1} := \sum_{i=1}^{j} s_i$. Thus,

$$\left|\langle \theta_{t_{j+1}}, v_{\mu,+}\rangle\right| \geq (1+\gamma\mu)^{\bar{t}_{j+1}}|\langle \theta_0, v_{\mu,+}\rangle| - \beta(1+\gamma\mu)^{\bar{t}_{j+1}}T.$$

Taking $\beta \leq \frac{|\langle \theta_0, v_{\mu,+}\rangle|}{2T}$, we ensure that

$$\langle \theta_{t_{j+1}}, v_{\mu,+}\rangle^2 \geq \frac{1}{4}(1+\gamma\mu)^{2\bar{t}_{j+1}}\langle \theta_0, v_{\mu,+}\rangle^2 > \frac{1}{4}(1+2\gamma\mu)^{\bar{t}_{j+1}}\langle \theta_0, v_{\mu,+}\rangle^2 \tag{25}$$

for all $j \geq 1$ such that $t_{j+1} \leq T-1$.

Recall that

$$\langle \theta_t, \mathbf{A}_2\theta_t\rangle = \lambda_2\langle \theta_t, v_{2,+}\rangle^2 - \lambda_2\langle \theta_t, v_{2,-}\rangle^2 + \mu\langle \theta_t, v_{\mu,+}\rangle^2 - \mu\langle \theta_t, v_{\mu,-}\rangle^2$$
$$\geq -(\lambda_2+\mu)\|\theta_t\|^2 + \mu\langle \theta_t, v_{\mu,+}\rangle^2$$

for all $t \geq 0$. Using (25) and (50),

$$\langle \theta_{t_j}, \mathbf{A}_2\theta_{t_j}\rangle > \frac{\mu}{4}(1+2\gamma\mu)^{\bar{t}_j}\langle \theta_0, v_{\mu,+}\rangle^2 - 1024(\lambda_2+\mu)\left(1+\frac{93\gamma\mu}{50}\right)^{\bar{t}_j}\|\theta_0\|^2 \tag{26}$$

for all $j \geq 2$ such that $t_j \leq T-1$. Notice that

$$\frac{\mu}{4}(1+2\gamma\mu)^{\bar{t}}\langle \theta_0, v_{\mu,+}\rangle^2 - 1024(\lambda_2+\mu)\left(1+\frac{93\gamma\mu}{50}\right)^{\bar{t}}\|\theta_0\|^2 \geq 0 \tag{27}$$

for

$$\bar{t} := \left\lceil \frac{\log\left(\frac{2^{12}(\lambda_2+\mu)\|\theta_0\|^2}{\mu\langle \theta_0, v_{\mu,+}\rangle^2}\right)}{\log\left(1+\frac{\frac{7}{50}\gamma\mu}{1+\frac{93\gamma\mu}{50}}\right)} \right\rceil \leq \frac{300}{7\gamma\mu}\log\left(\frac{4096(\lambda_2+\mu)\|\theta_0\|^2}{\mu\langle \theta_0, v_{\mu,+}\rangle^2}\right), \tag{28}$$

where we use $\gamma \leq \frac{1}{\mu}$ and $\log(1+x) \geq \frac{x}{2}$ for all $0 \leq x \leq 1$.

For $i = 1$, $t_1 = 0 \leq T-1$, and

$$s_1 \leq 1 + \frac{4}{\gamma\lambda_1}\log\left(\frac{512\|\theta_0\|^2}{\alpha^2}\right).$$

Moreover, combining (24) and (50) with the previous inequality,

$$s_i \leq 1 + \frac{4}{\gamma\lambda_1}\log\left(\frac{2^{19}\|\theta_0\|^2}{\alpha^2}\right) + \frac{8\mu}{\lambda_1}\bar{t}_i \tag{29}$$

for all $i \geq 1$ such that $t_i \leq T-1$.

Notice that $\bar{t}_1 = t_1 = 0 < T-1$. Moreover, due to (27) and (28), we have $\bar{t}_j \leq \bar{t}$ for all $j \geq 1$, since if $\bar{t}_j > \bar{t}$ for some $j \geq 1$, then (27) would be satisfied with $\bar{t}_j - 1$, and option 2 would no longer be chosen by the method. Moreover, if $t_k \leq T-1$, then $t_{k+1} = t_k + s_k \geq t_k + 1 > t_k$, meaning that there exists the smallest $p \geq 2$ such that $t_p \geq T$. Finally, we will show a contradiction to the fact $\bar{t}_j \leq \bar{t}$ for all $j \geq 1$.

Indeed, for all $j < p$, we have $t_j \leq T-1$ and $\bar{t}_j \leq \bar{t}$. Due to (29),

$$s_i \leq 1 + \frac{4}{\gamma\lambda_1}\log\left(\frac{2^{19}\|\theta_0\|^2}{\alpha^2}\right) + \frac{8\mu}{\lambda_1}\bar{t}_i$$
$$\overset{\bar{t}_j \leq \bar{t}}{\leq} 1 + \frac{4}{\gamma\lambda_1}\log\left(\frac{2^{19}\|\theta_0\|^2}{\alpha^2}\right) + \frac{8\mu}{\lambda_1}\bar{t}$$
$$\overset{(28)}{\leq} 1 + \frac{4}{\gamma\lambda_1}\log\left(\frac{2^{19}\|\theta_0\|^2}{\alpha^2}\right) + \frac{350}{\gamma\lambda_1}\log\left(\frac{4096(\lambda_2+\mu)\|\theta_0\|^2}{\mu\langle \theta_0, v_{\mu,+}\rangle^2}\right) \tag{30}$$
$$\leq 1 + \frac{700}{\gamma\lambda_1}\log\left(2^{20}\max\left\{\frac{1}{\alpha^2}, \frac{\lambda_2}{\mu\langle \theta_0, v_{\mu,+}\rangle^2}\right\}\|\theta_0\|^2\right) := M$$

for all $2 \leq i < p$. Thus,

$$t_p := \sum_{j=1}^{p-1} s_j \leq (p-1)M.$$

Since $t_p \geq T$, necessarily,

$$p \geq 1 + \frac{T}{M}.$$

Hence,

$$\bar{t}_p := \sum_{i \in [p-1] \text{ s.t. } q_i=2} s_i \geq \sum_{i \in [p-1] \text{ s.t. } q_i=2} 1 \geq \frac{p-1}{2} - 1 > \frac{T}{2M} - 1,$$

where use that the options 1 and 2 interleave. The inequality contradicts $\bar{t}_p \leq \bar{t}$ for all $p \geq 1$ because we choose $T$ such that

$$\frac{T}{2M} - 1 > \frac{300}{7\gamma\mu} \log \left( \frac{4096(\lambda_2 + \mu) \|\theta_0\|^2}{\mu \langle \theta_0, v_{\mu,+} \rangle^2} \right) \overset{(28)}{\geq} \bar{t},$$

which is shown in Lemma C.1.

$\square$

## C.3. Lemmas

**Lemma C.1.** *For the choice of $T, \gamma$, and $\sigma$ in Theorem 5.1, we have*

$$D := 2M \left( 1 + \frac{300}{7\gamma\mu} \log \left( \frac{4096(\lambda_2 + \mu) \|\theta_0\|^2}{\mu \langle \theta_0, v_{\mu,+} \rangle^2} \right) \right) < T,$$

*where $M := 1 + \frac{700}{\gamma\lambda_1} \log \left( 2^{20} \max \left\{ \frac{1}{\alpha^2}, \frac{\lambda_2}{\mu\langle\theta_0,v_{\mu,+}\rangle^2} \right\} \|\theta_0\|^2 \right)$ and $\delta = \rho/5T$.*

*Proof.* Note that

$$D := 2M \left( 1 + \frac{300}{7\gamma\mu} \log \left( \frac{4096(\lambda_2 + \mu) \|\theta_0\|^2}{\mu \langle \theta_0, v_{\mu,+} \rangle^2} \right) \right)$$

$$\overset{\gamma \leq \frac{1}{4\sqrt{d}}, \lambda_2 \leq 4\sqrt{d}}{\leq} M \left( \frac{86}{\gamma\mu} \log \left( \frac{2^{15}\sqrt{d} \|\theta_0\|^2}{\mu \langle \theta_0, v_{\mu,+} \rangle^2} \right) \right)$$

$$= 2 \left( 1 + \frac{700}{\gamma\lambda_1} \log \left( 2^{20} \max \left\{ \frac{1}{\alpha^2}, \frac{\lambda_2}{\mu \langle \theta_0, v_{\mu,+} \rangle^2} \right\} \|\theta_0\|^2 \right) \right) \left( \frac{86}{\gamma\mu} \log \left( \frac{2^{15}\sqrt{d} \|\theta_0\|^2}{\mu \langle \theta_0, v_{\mu,+} \rangle^2} \right) \right)$$

$$\overset{\lambda_1 = \sqrt{2d}, \gamma \leq \frac{1}{4\sqrt{d}}}{\leq} \frac{2^{19}}{\gamma\mu} \times \log \left( 2^{20} \max \left\{ \frac{1}{\alpha^2}, \frac{\lambda_2}{\mu \langle \theta_0, v_{\mu,+} \rangle^2} \right\} \|\theta_0\|^2 \right) \log \left( \frac{2^{15}\sqrt{d} \|\theta_0\|^2}{\mu \langle \theta_0, v_{\mu,+} \rangle^2} \right).$$

Using the choice of $\gamma$ and $\sigma$, and the fact that $\delta = \rho/5T$,

$$\alpha^2 = \frac{\sigma^2\delta^2}{20^2 7 \log^2 \left( \frac{1}{\gamma\lambda_1} \right)} \geq \frac{\sigma^2\delta^2}{5600} \geq \frac{\sigma^2\rho^2}{140000T^2}$$

$$= \frac{\rho^2}{140000T^2} \times \frac{1}{4096^2 \max\{d, \log(T/\rho)\}} \times \frac{|\langle\theta_0, v_{\mu,+}\rangle|^2}{T^2}.$$

Since $T \geq \frac{1}{\gamma\mu}$ and $T \geq \frac{4\sqrt{d}}{\mu}$, we get

$$\alpha^2 \geq \frac{\rho^3 \langle \theta_0, v_{\mu,+} \rangle^2}{2^{42} T^6}.$$

Thus,

$$\max \left\{ \frac{1}{\alpha^2}, \frac{\lambda_2}{\mu \langle \theta_0, v_{\mu,+} \rangle^2} \right\} \leq \max \left\{ \frac{1}{\alpha^2}, \frac{4\sqrt{d}}{\mu \langle \theta_0, v_{\mu,+} \rangle^2} \right\} \leq \frac{2^{42} T^6}{\rho^3 \langle \theta_0, v_{\mu,+} \rangle^2}$$

and

$$\begin{aligned}
D &\leq \frac{2^{19}}{\gamma\mu} \log \left( \frac{2^{62} T^6 \|\theta_0\|^2}{\rho^3 \langle \theta_0, v_{\mu,+} \rangle^2} \right) \log \left( \frac{2^{15} \sqrt{d} \|\theta_0\|^2}{\mu \langle \theta_0, v_{\mu,+} \rangle^2} \right) \\
&= \frac{2^{19}}{\gamma\mu} \log \left( \frac{2^{62} \|\theta_0\|^2}{\rho^3 \langle \theta_0, v_{\mu,+} \rangle^2} \right) \log \left( \frac{2^{15} \sqrt{d} \|\theta_0\|^2}{\mu \langle \theta_0, v_{\mu,+} \rangle^2} \right) + \frac{6 \times 2^{19}}{\gamma\mu} \log \left( \frac{2^{15} \sqrt{d} \|\theta_0\|^2}{\mu \langle \theta_0, v_{\mu,+} \rangle^2} \right) \times \log (T) \\
&= \frac{2^{21} \sqrt{d}}{\mu} \log \left( \frac{2^{62} \|\theta_0\|^2}{\rho^3 \langle \theta_0, v_{\mu,+} \rangle^2} \right) \log \left( \frac{2^{15} \sqrt{d} \|\theta_0\|^2}{\mu \langle \theta_0, v_{\mu,+} \rangle^2} \right) + \frac{2^{24} \sqrt{d}}{\mu} \log \left( \frac{2^{15} \sqrt{d} \|\theta_0\|^2}{\mu \langle \theta_0, v_{\mu,+} \rangle^2} \right) \times \log (T) \\
&\leq \frac{2^{24} \sqrt{d}}{\mu} \log \left( \frac{2^{62} \|\theta_0\|^2}{\rho^3 \langle \theta_0, v_{\mu,+} \rangle^2} \right) \log \left( \frac{2^{15} \sqrt{d} \|\theta_0\|^2}{\mu \langle \theta_0, v_{\mu,+} \rangle^2} \right) + \frac{2^{24} \sqrt{d}}{\mu} \log \left( \frac{2^{15} \sqrt{d} \|\theta_0\|^2}{\mu \langle \theta_0, v_{\mu,+} \rangle^2} \right) \times \log (T)
\end{aligned}$$

where we arranged terms and use the choice of $\gamma$. Using Lemma D.1, we can conclude that $D < T$ for our initial choice of

$$T = \left\lceil B \times \frac{\sqrt{d}}{\mu} \right\rceil = \tilde{\Theta} \left( \frac{\sqrt{d}}{\mu} \right),$$

where $B := A \log \left( \frac{\sqrt{d}}{\mu} A \right) = \tilde{\Theta} (1)$, $A := 2^{27} \log \left( \frac{2^{62} \|\theta_0\|^2}{\rho^3 \langle \theta_0, v_{\mu,+} \rangle^2} \right) \log \left( \frac{2^{15} \sqrt{d} \|\theta_0\|^2}{\mu \langle \theta_0, v_{\mu,+} \rangle^2} \right) = \tilde{\Theta} (1)$ □

**Lemma C.2.** *In Two-Sample Quadratic Perceptron Algorithm, on the set $\Omega_T$ from Lemma C.4 and for the choice of $\sigma$ in Theorem 5.1,*

$$\|\theta_t\|^2 \leq 256 \left( 1 + \frac{93\gamma\mu}{50} \right)^{t-s} \max\{\|\theta_s\|^2, \|\theta_0\|^2\}. \tag{31}$$

*for all $t \geq s \geq 0$.*

*Moreover, we can provide a more detailed result. For any starting point $\theta_0$, we have have the sequence of options $i_1, \ldots, i_t, \ldots$, where $i_t \in \{1, 2, 3\}$ for all $t \geq 0$, and 1 corresponds to the option with matrix $\mathbf{A}_1$, 2 corresponds to the option with matrix $\mathbf{A}_2$, and 3 corresponds to the "brake" option. For all $t \geq 0$, if $i_t = 1$ and $i_{t+1} = 1$, then*

$$\|\theta_{t+1}\|^2 \leq \left( 1 + \min \left\{ \frac{1}{T}, \gamma\mu \right\} \right) \max\{\|\theta_t\|^2, \|\theta_0\|^2\}. \tag{32}$$

*For all $t \geq 0$, if $i_t = 2$ and $i_{t+1} = 2$, then*

$$\|\theta_{t+1}\|^2 \leq \left( 1 + \frac{11}{8} \gamma\mu \right) \max\{\|\theta_t\|^2, \|\theta_0\|^2\}. \tag{33}$$

*For all $t \geq 0$, if $i_t = 1$ and $i_{t+1} = 2$, then*

$$\|\theta_{t+2}\|^2 \leq \max \left\{ \left( 1 + \frac{93\gamma\mu}{50} \right) \|\theta_t\|^2, 64 \|\theta_0\|^2 \right\}. \tag{34}$$

*For all $t \geq 0$, if $i_t = 2$ and $i_{t+1} = 1$, then*

$$\|\theta_{t+2}\|^2 \leq \max \left\{ \left( 1 + \frac{93\gamma\mu}{50} \right) \|\theta_t\|^2, 64 \|\theta_0\|^2 \right\}. \tag{35}$$

*For all $t \geq 0$,*

$$\|\theta_{t+1}\|^2 \leq 4 \max\{\|\theta_t\|^2, \|\theta_0\|^2\}. \tag{36}$$

*Proof.* For the choice of $\sigma$, we have (20).

For any starting point $\theta_0$, we have have the sequence of options $i_1, \ldots, i_t, \ldots$, where $i_t \in \{1, 2, 3\}$ for all $t \geq 0$, and 1 corresponds to the option with matrix $\mathbf{A}_1$, 2 corresponds to the option with matrix $\mathbf{A}_2$, and 3 corresponds to the "brake" option. Since $\lambda_2 \geq \max_{i \in \{1,2\}} \|\mathbf{A}_i\| = \max\{\lambda_1, \lambda_2\}$, we have

$$\|\theta_{t+1}\| = \|\theta_t + \gamma \mathbf{A}_{j_t} \theta_t + \xi_t\| \leq \|\theta_t\| + \gamma \|\mathbf{A}_{j_t} \theta_t\| + \|\xi_t\| \overset{(53)}{\leq} (1 + \gamma \lambda_2) \|\theta_t\| + \beta.$$

for all $t \geq 0$, where $j_t \in \{1, 2\}$ is the index of the matrix $\mathbf{A}_{j_t}$ that is used at the $t$-th step. Since $\beta \leq \gamma \lambda_2 \|\theta_0\|$, we get

$$\|\theta_{t+1}\|^2 \leq (1 + 2\gamma \lambda_2)^2 \max\{\|\theta_t\|^2, \|\theta_0\|^2\} \leq 4 \max\{\|\theta_t\|^2, \|\theta_0\|^2\} \qquad \forall t \geq 0 \tag{37}$$

for all $\gamma \leq \frac{1}{4\lambda_2}$.

Next, consider any step $i_t$. We will consider four options:
**(Option 1):** If $i_t = 1$ and $i_{t+1} = 1$, then

$$\theta_{t+1} = \theta_t + \gamma \mathbf{A}_1 \theta_t + \xi_t \text{ and } \langle \theta_{t+1}, \mathbf{A}_1 \theta_{t+1} \rangle \leq 0. \tag{38}$$

Using (98), we get

$$\|\theta_{t+1}\|^2 = \|\theta_t + \gamma \mathbf{A}_1 \theta_t + \xi_t\|^2 \leq (1 + s) \|\theta_t + \gamma \mathbf{A}_1 \theta_t\|^2 + (1 + s^{-1}) \|\xi_t\|^2$$

for all $s > 0$. Let us denote $\{v_{1,j}\}_{i=1}^d$ as eigenvectors corresponding to the kernel of $\mathbf{A}_1$. Rewriting the norm $\|\theta_t + \gamma \mathbf{A}_1 \theta_t\|^2$ in the eigenvectors of $\mathbf{A}_1$ (see Proposition 5.3), we get

$$
\begin{aligned}
\|\theta_{t+1}\|^2 &\leq (1 + s) \left( \langle \theta_t + \gamma \mathbf{A}_1 \theta_t, v_{1,+} \rangle^2 + \langle \theta_t + \gamma \mathbf{A}_1 \theta_t, v_{1,-} \rangle^2 + \sum_{j=1}^d \langle \theta_t + \gamma \mathbf{A}_1 \theta_t, v_{1,j} \rangle^2 \right) + (1 + s^{-1}) \|\xi_t\|^2 \\
&= (1 + s) \left( (1 + \gamma \lambda_1)^2 \langle \theta_t, v_{1,+} \rangle^2 + (1 - \gamma \lambda_1)^2 \langle \theta_t, v_{1,-} \rangle^2 + \sum_{j=1}^d \langle \theta_t, v_{1,j} \rangle^2 \right) + (1 + s^{-1}) \|\xi_t\|^2 \\
&= (1 + s) \left( (1 + \gamma^2 \lambda_1^2)(\langle \theta_t, v_{1,+} \rangle^2 + \langle \theta_t, v_{1,-} \rangle^2) + 2\gamma \lambda_1 (\langle \theta_t, v_{1,-} \rangle^2 - \langle \theta_t, v_{1,+} \rangle^2) + \sum_{j=1}^d \langle \theta_t, v_{1,j} \rangle^2 \right) \\
&\quad + (1 + s^{-1}) \|\xi_t\|^2
\end{aligned}
\tag{39}
$$

since $\mathbf{A}_1 v_{1,+} = \lambda_1 v_{1,+}$, $\mathbf{A}_1 v_{1,-} = -\lambda_1 v_{1,-}$, and $\mathbf{A}_1 v_{1,j} = 0$ for all $j \geq 1$. Using (38),

$$
\begin{aligned}
0 \geq \langle \theta_{t+1}, \mathbf{A}_1 \theta_{t+1} \rangle &= \langle \theta_t + \gamma \mathbf{A}_1 \theta_t + \xi_t, \mathbf{A}_1 (\theta_t + \gamma \mathbf{A}_1 \theta_t + \xi_t) \rangle \\
&= \langle \theta_t + \gamma \mathbf{A}_1 \theta_t, \mathbf{A}_1 (\theta_t + \gamma \mathbf{A}_1 \theta_t) \rangle + 2 \langle \xi_t, \mathbf{A}_1 (\theta_t + \gamma \mathbf{A}_1 \theta_t) \rangle + \langle \xi_t, \mathbf{A}_1 \xi_t \rangle \\
&\geq \langle \theta_t + \gamma \mathbf{A}_1 \theta_t, \mathbf{A}_1 (\theta_t + \gamma \mathbf{A}_1 \theta_t) \rangle - 4\lambda_1 \|\xi_t\| \|\theta_t\| - \lambda_1 \|\xi_t\|^2
\end{aligned}
$$

since $\|\mathbf{A}_1\| = \lambda_1$ and $\|\mathbf{I} + \gamma \mathbf{A}_1\| \leq 2$. Rewriting the dot product in the eigenvectors of $\mathbf{A}_1$, we get

$$\lambda_1 \langle \theta_t + \gamma \mathbf{A}_1 \theta_t, v_{1,+} \rangle^2 - \lambda_1 \langle \theta_t + \gamma \mathbf{A}_1 \theta_t, v_{1,-} \rangle^2 - 4\lambda_1 \|\xi_t\| \|\theta_t\| - \lambda_1 \|\xi_t\|^2 \leq 0,$$

$$(1 + \gamma \lambda_1)^2 \langle \theta_t, v_{1,+} \rangle^2 - (1 - \gamma \lambda_1)^2 \langle \theta_t, v_{1,-} \rangle^2 \leq 4 \|\xi_t\| \|\theta_t\| + \|\xi_t\|^2,$$

and

$$
\begin{aligned}
\langle \theta_t, v_{1,+} \rangle^2 - \langle \theta_t, v_{1,-} \rangle^2 &\leq -\frac{2\gamma \lambda_1}{1 + \gamma^2 \lambda_1^2} (\langle \theta_t, v_{1,+} \rangle^2 + \langle \theta_t, v_{1,-} \rangle^2) + \frac{4 \|\xi_t\| \|\theta_t\| + \|\xi_t\|^2}{1 + \gamma^2 \lambda_1^2} \\
&\leq -\gamma \lambda_1 (\langle \theta_t, v_{1,+} \rangle^2 + \langle \theta_t, v_{1,-} \rangle^2) + 4 \|\xi_t\| \|\theta_t\| + \|\xi_t\|^2.
\end{aligned}
$$

since $0 \leq \gamma \leq \frac{1}{\lambda_1}$. Substituting to (39),

$$
\begin{aligned}
\|\theta_{t+1}\|^2 &\leq (1+s)\Bigg[(1+\gamma^2\lambda_1^2)(\langle\theta_t, v_{1,+}\rangle^2 + \langle\theta_t, v_{1,-}\rangle^2) \\
&\quad + 2\gamma\lambda_1\left(-\gamma\lambda_1(\langle\theta_t, v_{1,+}\rangle^2 + \langle\theta_t, v_{1,-}\rangle^2) + 4\|\xi_t\|\|\theta_t\| + \|\xi_t\|^2\right) + \sum_{j=1}^{d}\langle\theta_t, v_{1,j}\rangle^2\Bigg] + (1+s^{-1})\|\xi_t\|^2 \\
&= (1+s)\Bigg[(1-\gamma^2\lambda_1^2)(\langle\theta_t, v_{1,+}\rangle^2 + \langle\theta_t, v_{1,-}\rangle^2) + \sum_{j=1}^{d}\langle\theta_t, v_{1,j}\rangle^2\Bigg] \\
&\quad + (1+s^{-1})\|\xi_t\|^2 + 2(1+s)\gamma\lambda_1\left(4\|\xi_t\|\|\theta_t\| + \|\xi_t\|^2\right) \\
&\leq (1+s)\|\theta_t\|^2 + (1+s^{-1}+2(1+s)\gamma\lambda_1)\|\xi_t\|^2 + 8(1+s)\gamma\lambda_1\|\xi_t\|\|\theta_t\| \\
&\leq (1+2s)\|\theta_t\|^2 + \left(1+s^{-1}+2(1+s)\gamma\lambda_1 + \frac{32(1+s)^2\gamma^2\lambda_1^2}{s}\right)\|\xi_t\|^2
\end{aligned}
$$

due to $\langle\theta_t, v_{1,+}\rangle^2 + \langle\theta_t, v_{1,-}\rangle^2 + \sum_{j=1}^{d}\langle\theta_t, v_{1,j}\rangle^2 = \|\theta_t\|^2$ and Young's inequality (98). Recall that $s$ is a free parameter, and we can choose $s = \frac{1}{4}\min\left\{\frac{1}{T}, \gamma\mu\right\}$, Moreover, $\gamma \leq \frac{1}{\lambda_1}$. Therefore,

$$
\|\theta_{t+1}\|^2 \leq (1+2s)\|\theta_t\|^2 + \frac{256}{s}\|\xi_t\|^2 \overset{(53)}{\leq} (1+2s)\|\theta_t\|^2 + \frac{256}{s}\beta^2,
$$

$$
\|\theta_{t+1}\|^2 \leq \left(1 + \frac{1}{2}\min\left\{\frac{1}{T}, \gamma\mu\right\}\right)\|\theta_t\|^2 + 1024\max\left\{T, \frac{1}{\gamma\mu}\right\}\beta^2
$$

and taking $\beta \leq \frac{1}{64}\min\left\{\frac{1}{T}, \gamma\mu\right\}\|\theta_0\|$,

$$
\|\theta_{t+1}\|^2 \leq \left(1 + \min\left\{\frac{1}{T}, \gamma\mu\right\}\right)\max\{\|\theta_t\|^2, \|\theta_0\|^2\} \leq (1+\gamma\mu)\max\{\|\theta_t\|^2, \|\theta_0\|^2\}. \tag{40}
$$

**(Option 2):** If $i_t = 2$ and $i_{t+1} = 2$, then

$$
\theta_{t+1} = \theta_t + \gamma\mathbf{A}_2\theta_t + \xi_t \text{ and } \langle\theta_{t+1}, \mathbf{A}_2\theta_{t+1}\rangle \leq 0. \tag{41}
$$

Let us denote $\{v_{1,j}\}_{i=1}^{d-2}$ as eigenvectors corresponding to the kernel of $\mathbf{A}_2$. Similarly to the previous steps, using Proposi-

tion 5.3,

$$
\begin{aligned}
\|\theta_{t+1}\|^2 &\leq (1+s)\Bigg( \langle \theta_t + \gamma \mathbf{A}_2 \theta_t, v_{2,+} \rangle^2 + \langle \theta_t + \gamma \mathbf{A}_2 \theta_t, v_{2,-} \rangle^2 \\
&\quad + \langle \theta_t + \gamma \mathbf{A}_2 \theta_t, v_{\mu,+} \rangle^2 + \langle \theta_t + \gamma \mathbf{A}_2 \theta_t, v_{\mu,-} \rangle^2 + \sum_{j=1}^{d-2} \langle \theta_t + \gamma \mathbf{A}_2 \theta_t, v_{2,j} \rangle^2 \Bigg) \\
&\quad + (1 + s^{-1}) \|\xi_t\|^2 \\
&= (1+s)\Bigg( (1+\gamma\lambda_2)^2 \langle \theta_t, v_{2,+} \rangle^2 + (1-\gamma\lambda_2)^2 \langle \theta_t, v_{2,-} \rangle^2 \\
&\quad + (1+\gamma\mu)^2 \langle \theta_t, v_{\mu,+} \rangle^2 + (1-\gamma\mu)^2 \langle \theta_t, v_{\mu,-} \rangle^2 + \sum_{j=1}^{d-2} \langle \theta_t, v_{2,j} \rangle^2 \Bigg) \\
&\quad + (1 + s^{-1}) \|\xi_t\|^2 \\
&= (1+s)\Bigg( (1+\gamma^2\lambda_2^2)(\langle \theta_t, v_{2,+} \rangle^2 + \langle \theta_t, v_{2,-} \rangle^2) + 2\gamma\lambda_2(\langle \theta_t, v_{2,+} \rangle^2 - \langle \theta_t, v_{2,-} \rangle^2) \\
&\quad + (1+\gamma\mu)^2 \langle \theta_t, v_{\mu,+} \rangle^2 + (1-\gamma\mu)^2 \langle \theta_t, v_{\mu,-} \rangle^2 + \sum_{j=1}^{d-2} \langle \theta_t, v_{2,j} \rangle^2 \Bigg) \\
&\quad + (1 + s^{-1}) \|\xi_t\|^2 .
\end{aligned}
\tag{42}
$$

Using (41),

$$
\begin{aligned}
0 \geq \langle \theta_{t+1}, \mathbf{A}_2 \theta_{t+1} \rangle &= \langle \theta_t + \gamma \mathbf{A}_2 \theta_t + \xi_t, \mathbf{A}_2 (\theta_t + \gamma \mathbf{A}_2 \theta_t + \xi_t) \rangle \\
&= \langle \theta_t + \gamma \mathbf{A}_2 \theta_t, \mathbf{A}_2 (\theta_t + \gamma \mathbf{A}_2 \theta_t) \rangle + 2 \langle \xi_t, \mathbf{A}_2 (\theta_t + \gamma \mathbf{A}_2 \theta_t) \rangle + \langle \xi_t, \mathbf{A}_2 \xi_t \rangle \\
&\geq \langle \theta_t + \gamma \mathbf{A}_2 \theta_t, \mathbf{A}_2 (\theta_t + \gamma \mathbf{A}_2 \theta_t) \rangle - 4\lambda_2 \|\xi_t\| \|\theta_t\| - \lambda_2 \|\xi_t\|^2 .
\end{aligned}
$$

Rewriting the dot product in the eigenvectors of $\mathbf{A}_2$, we get

$$
\begin{aligned}
&\lambda_2 \langle \theta_t + \gamma \mathbf{A}_2 \theta_t, v_{2,+} \rangle^2 - \lambda_2 \langle \theta_t + \gamma \mathbf{A}_2 \theta_t, v_{2,-} \rangle^2 + \mu \langle \theta_t + \gamma \mathbf{A}_2 \theta_t, v_{\mu,+} \rangle^2 - \mu \langle \theta_t + \gamma \mathbf{A}_2 \theta_t, v_{\mu,-} \rangle^2 \\
&\leq 4\lambda_2 \|\xi_t\| \|\theta_t\| + \lambda_2 \|\xi_t\|^2 .
\end{aligned}
$$

Due to Proposition 5.3,

$$
\begin{aligned}
&\lambda_2(1+\gamma\lambda_2)^2 \langle \theta_t, v_{2,+} \rangle^2 - \lambda_2(1-\gamma\lambda_2)^2 \langle \theta_t, v_{2,-} \rangle^2 + \mu(1+\gamma\mu)^2 \langle \theta_t, v_{\mu,+} \rangle^2 - \mu(1-\gamma\mu)^2 \langle \theta_t, v_{\mu,-} \rangle^2 \\
&\leq 4\lambda_2 \|\xi_t\| \|\theta_t\| + \lambda_2 \|\xi_t\|^2
\end{aligned}
$$

and

$$
\begin{aligned}
&\langle \theta_t, v_{2,+} \rangle^2 - \langle \theta_t, v_{2,-} \rangle^2 \\
&\leq -\frac{2\gamma\lambda_2}{1+\gamma^2\lambda_2^2}(\langle \theta_t, v_{2,+} \rangle^2 + \langle \theta_t, v_{2,-} \rangle^2) - \frac{\mu(1+\gamma\mu)^2}{\lambda_2(1+\gamma^2\lambda_2^2)} \langle \theta_t, v_{\mu,+} \rangle^2 + \frac{\mu(1-\gamma\mu)^2}{\lambda_2(1+\gamma^2\lambda_2^2)} \langle \theta_t, v_{\mu,-} \rangle^2 \\
&\quad + \frac{4 \|\xi_t\| \|\theta_t\| + \|\xi_t\|^2}{1+\gamma^2\lambda_2^2},
\end{aligned}
$$

where we have arranged terms. Substituting to (42),

$$
\begin{aligned}
\|\theta_{t+1}\|^2 &\leq (1+s)\Bigg[ (1+\gamma^2\lambda_2^2)(\langle \theta_t, v_{2,+} \rangle^2 + \langle \theta_t, v_{2,-} \rangle^2) \\
&\quad + 2\gamma\lambda_2 \left( -\frac{2\gamma\lambda_2}{1+\gamma^2\lambda_2^2}(\langle \theta_t, v_{2,+} \rangle^2 + \langle \theta_t, v_{2,-} \rangle^2) - \frac{\mu(1+\gamma\mu)^2}{\lambda_2(1+\gamma^2\lambda_2^2)} \langle \theta_t, v_{\mu,+} \rangle^2 + \frac{\mu(1-\gamma\mu)^2}{\lambda_2(1+\gamma^2\lambda_2^2)} \langle \theta_t, v_{\mu,-} \rangle^2 \right)
\end{aligned}
$$

$$+ 2\gamma\lambda_2 \left( \frac{4 \left\| \xi_t \right\| \left\| \theta_t \right\| + \left\| \xi_t \right\|^2}{1 + \gamma^2 \lambda_2^2} \right)$$

$$+ (1 + \gamma\mu)^2 \left\langle \theta_t, v_{\mu,+} \right\rangle^2 + (1 - \gamma\mu)^2 \left\langle \theta_t, v_{\mu,-} \right\rangle^2 + \sum_{j=1}^{d-2} \left\langle \theta_t, v_{2,j} \right\rangle^2 \Bigg]$$

$$+ (1 + s^{-1}) \left\| \xi_t \right\|^2 .$$

Since $0 \le \gamma \le \frac{1}{\lambda_2}$ and $0 \le \gamma \le \frac{1}{\mu}$,

$$\left\| \theta_{t+1} \right\|^2 \le (1 + s) \Bigg[ \left\langle \theta_t, v_{2,+} \right\rangle^2 + \left\langle \theta_t, v_{2,-} \right\rangle^2$$

$$+ (1 + \gamma\mu)^2 (1 - \gamma\mu) \left\langle \theta_t, v_{\mu,+} \right\rangle^2 + (1 - \gamma\mu)^2 (1 + 2\gamma\mu) \left\langle \theta_t, v_{\mu,-} \right\rangle^2 + \sum_{j=1}^{d-2} \left\langle \theta_t, v_{2,j} \right\rangle^2 \Bigg]$$

$$+ (1 + s^{-1}) \left\| \xi_t \right\|^2 + (1 + s)2\gamma\lambda_2 \left( 4 \left\| \xi_t \right\| \left\| \theta_t \right\| + \left\| \xi_t \right\|^2 \right)$$

$$\le (1 + s) \Bigg[ \left\langle \theta_t, v_{2,+} \right\rangle^2 + \left\langle \theta_t, v_{2,-} \right\rangle^2$$

$$+ (1 + \gamma\mu) \left\langle \theta_t, v_{\mu,+} \right\rangle^2 + \left\langle \theta_t, v_{\mu,-} \right\rangle^2 + \sum_{j=1}^{d-2} \left\langle \theta_t, v_{2,j} \right\rangle^2 \Bigg]$$

$$+ (1 + s^{-1}) \left\| \xi_t \right\|^2 + (1 + s)2\gamma\lambda_2 \left( 4 \left\| \xi_t \right\| \left\| \theta_t \right\| + \left\| \xi_t \right\|^2 \right)$$

$$\le (1 + s)(1 + \gamma\mu) \left\| \theta_t \right\|^2 + (1 + s^{-1} + 2(1 + s)\gamma\lambda_2) \left\| \xi_t \right\|^2 + 8(1 + s)\gamma\lambda_2 \left\| \xi_t \right\| \left\| \theta_t \right\|$$

because $\left\langle \theta_t, v_{2,+} \right\rangle^2 + \left\langle \theta_t, v_{2,-} \right\rangle^2 + \left\langle \theta_t, v_{\mu,+} \right\rangle^2 + \left\langle \theta_t, v_{\mu,-} \right\rangle^2 + \sum_{j=1}^{d-2} \left\langle \theta_t, v_{1,j} \right\rangle^2 = \left\| \theta_t \right\|^2$. Taking $s = \frac{\gamma\mu}{16} \le 1$ and using Young's inequality,

$$\left\| \theta_{t+1} \right\|^2 \le \left( 1 + \frac{9}{8}\gamma\mu \right) \left\| \theta_t \right\|^2 + (1 + s^{-1} + 2(1 + s)\gamma\lambda_2) \left\| \xi_t \right\|^2 + 8(1 + s)\gamma\lambda_2 \left\| \xi_t \right\| \left\| \theta_t \right\|$$

$$\le \left( 1 + \frac{9}{8}\gamma\mu \right) \left\| \theta_t \right\|^2 + \left( 5 + \frac{16}{\gamma\mu} \right) \left\| \xi_t \right\|^2 + 512 \left\| \xi_t \right\|^2 \frac{1}{\gamma\mu} + \frac{\gamma\mu}{8} \left\| \theta_t \right\|^2$$

$$= \left( 1 + \frac{10}{8}\gamma\mu \right) \left\| \theta_t \right\|^2 + \frac{1024}{\gamma\mu} \left\| \xi_t \right\|^2 \overset{(53)}{\le} \left( 1 + \frac{10}{8}\gamma\mu \right) \left\| \theta_t \right\|^2 + \frac{1024}{\gamma\mu}\beta^2 .$$

where we use $0 \le \gamma \le \frac{1}{\lambda_2}$ and $0 \le \gamma \le \frac{1}{\mu}$. Taking $\beta \le \frac{1}{128}\gamma\mu \left\| \theta_0 \right\|$,

$$\left\| \theta_{t+1} \right\|^2 \le \left( 1 + \frac{11}{8}\gamma\mu \right) \max\{ \left\| \theta_t \right\|^2, \left\| \theta_0 \right\|^2 \}. \tag{43}$$

**(Option 3):** If $i_t = 1$ and $i_{t+1} = 2$, then

$$\theta_{t+2} = (\mathbf{I} + \gamma\mathbf{A}_2) \left( (\mathbf{I} + \gamma\mathbf{A}_1) \theta_t + \xi_t \right) + \xi_{t+1}, \quad \left\langle \theta_t, \mathbf{A}_1\theta_t \right\rangle \le 0,$$

$$\text{and } \left\langle (\mathbf{I} + \gamma\mathbf{A}_1) \theta_t + \xi_t, \mathbf{A}_2 \left( (\mathbf{I} + \gamma\mathbf{A}_1) \theta_t + \xi_t \right) \right\rangle \le 0. \tag{44}$$

Thus,

$$\left\| \theta_{t+2} \right\| = \left\| (\mathbf{I} + \gamma\mathbf{A}_2) \left( (\mathbf{I} + \gamma\mathbf{A}_1) \theta_t + \xi_t \right) + \xi_{t+1} \right\|$$

$$\le \left\| (\mathbf{I} + \gamma\mathbf{A}_2) (\mathbf{I} + \gamma\mathbf{A}_1) \theta_t \right\| + \left\| (\mathbf{I} + \gamma\mathbf{A}_2) \xi_t \right\| + \left\| \xi_{t+1} \right\|$$

$$\le \left\| (\mathbf{I} + \gamma\mathbf{A}_2) (\mathbf{I} + \gamma\mathbf{A}_1) \theta_t \right\| + 2 \left\| \xi_t \right\| + \left\| \xi_{t+1} \right\| \tag{45}$$

because $\|(\mathbf{I} + \gamma \mathbf{A}_2)\| \leq 2$ for all $\gamma \leq \frac{1}{\lambda_2}$. Next,

$$
\begin{aligned}
0 &\geq \langle (\mathbf{I} + \gamma \mathbf{A}_1) \theta_t + \xi_t, \mathbf{A}_2 ((\mathbf{I} + \gamma \mathbf{A}_1) \theta_t + \xi_t) \rangle \\
&= \langle (\mathbf{I} + \gamma \mathbf{A}_1) \theta_t, \mathbf{A}_2 ((\mathbf{I} + \gamma \mathbf{A}_1) \theta_t) \rangle + 2 \langle \xi_t, \mathbf{A}_2 (\mathbf{I} + \gamma \mathbf{A}_1) \theta_t \rangle + \langle \xi_t, \mathbf{A}_2 \xi_t \rangle \\
&\geq \langle (\mathbf{I} + \gamma \mathbf{A}_1) \theta_t, \mathbf{A}_2 ((\mathbf{I} + \gamma \mathbf{A}_1) \theta_t) \rangle - 4 \lambda_2 \|\xi_t\| \|\theta_t\| - \lambda_2 \|\xi_t\|^2 \\
&\geq \langle (\mathbf{I} + \gamma \mathbf{A}_1) \theta_t, \mathbf{A}_2 ((\mathbf{I} + \gamma \mathbf{A}_1) \theta_t) \rangle - s \|\theta_t\|^2 - \left( \lambda_2 + \frac{16 \lambda_2^2}{s} \right) \|\xi_t\|^2 .
\end{aligned}
$$

for all $s > 0$. If $\|\theta_t\| \leq \|\theta_0\|$, then

$$
\|\theta_{t+2}\| \leq 4 \|\theta_0\| + 2 \|\xi_t\| + \|\xi_{t+1}\| \overset{(53)}{\leq} 4 \|\theta_0\| + 3\beta \leq 8 \|\theta_0\| . \tag{46}
$$

Due to (45), $\|\mathbf{I} + \gamma \mathbf{A}_1\| \leq 2$, $\|\mathbf{I} + \gamma \mathbf{A}_2\| \leq 2$, and $\beta \leq \|\theta_0\|$. Otherwise, if $\|\theta_t\| > \|\theta_0\|$, then

$$
\|\theta_{t+2}\| \leq \left\| (\mathbf{I} + \gamma \mathbf{A}_2) (\mathbf{I} + \gamma \mathbf{A}_1) \frac{\theta_t}{\|\theta_t\|} \right\| \|\theta_t\| + 2 \|\xi_t\| + \|\xi_{t+1}\| ,
$$

$$
\left\langle \frac{\theta_t}{\|\theta_t\|}, \mathbf{A}_1 \frac{\theta_t}{\|\theta_t\|} \right\rangle \leq 0,
$$

and

$$
\left\langle (\mathbf{I} + \gamma \mathbf{A}_1) \frac{\theta_t}{\|\theta_t\|}, \mathbf{A}_2 \left( (\mathbf{I} + \gamma \mathbf{A}_1) \frac{\theta_t}{\|\theta_t\|} \right) \right\rangle \leq s + \left( \frac{\lambda_2}{\|\theta_t\|^2} + \frac{16 \lambda_2^2}{s \|\theta_t\|^2} \right) \|\xi_t\|^2 \leq s + \left( \frac{\lambda_2}{\|\theta_0\|^2} + \frac{16 \lambda_2^2}{s \|\theta_0\|^2} \right) \|\xi_t\|^2 .
$$

Using Lemma C.5 with $\delta = s + \left( \frac{\lambda_2}{\|\theta_0\|^2} + \frac{16 \lambda_2^2}{s \|\theta_0\|^2} \right) \|\xi_t\|^2$, we obtain

$$
\|\theta_{t+2}\| \leq \sqrt{1 + \frac{9 \gamma \mu}{5} + \left( s + \left( \frac{\lambda_2}{\|\theta_0\|^2} + \frac{16 \lambda_2^2}{s \|\theta_0\|^2} \right) \|\xi_t\|^2 \right) \frac{\gamma}{2}} \|\theta_t\| + 2 \|\xi_t\| + \|\xi_{t+1}\| ,
$$

Since $\|\xi_t\| \leq \beta$,

$$
\|\theta_{t+2}\| \leq \sqrt{1 + \frac{9 \gamma \mu}{5} + \left( s + \left( \frac{\lambda_2}{\|\theta_0\|^2} + \frac{16 \lambda_2^2}{s \|\theta_0\|^2} \right) \beta^2 \right) \frac{\gamma}{2}} \|\theta_t\| + 3\beta,
$$

Choosing $s = \frac{\mu}{25}$ and $\beta \leq \min \left\{ \frac{\mu \|\theta_0\|}{1050 \lambda_2}, \frac{\gamma \mu \|\theta_0\|}{1050} \right\} \left( \leq \min \left\{ 1, \sqrt{\frac{\mu}{25} \left( \frac{\lambda_2}{\|\theta_0\|^2} + \frac{400 \lambda_2^2}{\mu \|\theta_0\|^2} \right)^{-1}}, \frac{\gamma \mu \|\theta_0\|}{1050} \right\} \right)$,

$$
\|\theta_{t+2}\| \leq \sqrt{1 + \frac{93 \gamma \mu}{50}} \max \{ \|\theta_t\|, \|\theta_0\| \}
$$

for all $\gamma \leq \frac{1}{\mu}$ and $\gamma \leq \frac{1}{\lambda_2}$. Combining the last inequality with (46):

$$
\|\theta_{t+2}\| \leq \max \left\{ \sqrt{1 + \frac{93 \gamma \mu}{50}} \|\theta_t\|, 8 \|\theta_0\| \right\} . \tag{47}
$$

**(Option 4):** If $i_t = 2$ and $i_{t+1} = 1$, then the proof of the inequality

$$
\|\theta_{t+2}\| \leq \max \left\{ \sqrt{1 + \frac{93 \gamma \mu}{50}} \|\theta_t\|, 8 \|\theta_0\| \right\} . \tag{48}
$$

is the same as in the previous option, with the only change that we should use Lemma C.6 instead.

Using mathematical induction, we will show that

$$\min\{\|\theta_{t+1}\|^2, \|\theta_t\|^2\} \le 64 \left(1 + \frac{93\gamma\mu}{50}\right)^{t-s} \max\{\|\theta_s\|^2, \|\theta_0\|^2\} \tag{49}$$

for all $t \ge s$. Clearly, $\|\theta_s\|^2 \le 64 \max\{\|\theta_s\|^2, \|\theta_0\|^2\}$. Thus, $\min\{\|\theta_{s+1}\|^2, \|\theta_s\|^2\} \le 64(1 + \frac{93\gamma\mu}{50})^0 \max\{\|\theta_s\|^2, \|\theta_0\|^2\}$. Let the inequality $\min\{\|\theta_{t+1}\|^2, \|\theta_t\|^2\} \le 64(1 + \frac{93\gamma\mu}{50})^{t-s} \max\{\|\theta_s\|^2, \|\theta_0\|^2\}$ hold. If $\|\theta_t\|^2 > \|\theta_{t+1}\|^2$, then

$$\min\{\|\theta_{t+2}\|^2, \|\theta_{t+1}\|^2\} \le \|\theta_{t+1}\|^2 = \min\{\|\theta_{t+1}\|^2, \|\theta_t\|^2\} \le 64 \left(1 + \frac{93\gamma\mu}{50}\right)^{t-s} \max\{\|\theta_s\|^2, \|\theta_0\|^2\}$$

$$\le 64 \left(1 + \frac{93\gamma\mu}{50}\right)^{t-s+1} \max\{\|\theta_s\|^2, \|\theta_0\|^2\}.$$

Otherwise, if $\|\theta_t\|^2 \le \|\theta_{t+1}\|^2$, then in the case of **(Option 1)** and **(Option 2)**, we have

$$\min\{\|\theta_{t+2}\|^2, \|\theta_{t+1}\|^2\} \le \|\theta_{t+1}\|^2$$

$$\le \left(1 + \frac{11}{8}\gamma\mu\right) \max\{\|\theta_t\|^2, \|\theta_0\|^2\}$$

$$= \left(1 + \frac{11}{8}\gamma\mu\right) \max\left\{\min\left\{\|\theta_{t+1}\|^2, \|\theta_t\|^2\right\}, \|\theta_0\|^2\right\}$$

$$\le \left(1 + \frac{11}{8}\gamma\mu\right) \max\left\{64 \left(1 + \frac{93\gamma\mu}{50}\right)^{t-s} \max\{\|\theta_s\|^2, \|\theta_0\|^2\}, \|\theta_0\|^2\right\}$$

$$\le 64 \left(1 + \frac{93\gamma\mu}{50}\right)^{t-s+1} \max\{\|\theta_s\|^2, \|\theta_0\|^2\}.$$

If $\|\theta_t\|^2 \le \|\theta_{t+1}\|^2$, in the case of **(Option 3)** and **(Option 4)** we have

$$\min\{\|\theta_{t+2}\|^2, \|\theta_{t+1}\|^2\} \le \|\theta_{t+2}\|^2$$

$$\le \max\left\{\left(1 + \frac{93\gamma\mu}{50}\right)\|\theta_t\|^2, 64\|\theta_0\|^2\right\}$$

$$= \max\left\{\left(1 + \frac{93\gamma\mu}{50}\right)\min\left\{\|\theta_{t+1}\|^2, \|\theta_t\|^2\right\}, 64\|\theta_0\|^2\right\}$$

$$\le \max\left\{\left(1 + \frac{93\gamma\mu}{50}\right)64 \left(1 + \frac{93\gamma\mu}{50}\right)^{t-s} \max\{\|\theta_s\|^2, \|\theta_0\|^2\}, 64\|\theta_0\|^2\right\}$$

$$= 64 \left(1 + \frac{93\gamma\mu}{50}\right)^{t-s+1} \max\{\|\theta_s\|^2, \|\theta_0\|^2\}.$$

We have proved the next step of the mathematical induction.

Next, we use mathematical once again to show that $\|\theta_t\|^2 \le 256 \left(1 + \frac{93\gamma\mu}{50}\right)^{t-s} \max\{\|\theta_s\|^2, \|\theta_0\|^2\}$ for all $t \ge s$. Clearly, $\|\theta_s\|^2 \le 256 \max\{\|\theta_s\|^2, \|\theta_0\|^2\}$. For all $t \ge s + 1$, using (49), if $\|\theta_t\|^2 \le \|\theta_{t+1}\|^2$, then $\|\theta_t\|^2 \le 64(1 + \frac{93\gamma\mu}{50})^{t-s} \max\{\|\theta_s\|^2, \|\theta_0\|^2\} \le 256(1 + \frac{93\gamma\mu}{50})^{t-s} \max\{\|\theta_s\|^2, \|\theta_0\|^2\}$. If $\|\theta_t\|^2 > \|\theta_{t+1}\|^2$ and $\|\theta_t\|^2 \le \|\theta_{t-1}\|^2$, then $\|\theta_t\|^2 = \min\{\|\theta_t\|^2, \|\theta_{t-1}\|^2\} \le 64(1 + \frac{93\gamma\mu}{50})^{t-s-1} \max\{\|\theta_s\|^2, \|\theta_0\|^2\} \le 256(1 + \frac{93\gamma\mu}{50})^{t-s} \max\{\|\theta_s\|^2, \|\theta_0\|^2\}$. Finally, if $\|\theta_t\|^2 > \|\theta_{t+1}\|^2$ and $\|\theta_t\|^2 > \|\theta_{t-1}\|^2$, then

$$\|\theta_t\|^2 \overset{(37)}{\le} 4\|\theta_{t-1}\|^2 = 4\min\{\|\theta_t\|^2, \|\theta_{t-1}\|^2\} \le 4 \times 64 \left(1 + \frac{93\gamma\mu}{50}\right)^{t-s-1} \max\{\|\theta_s\|^2, \|\theta_0\|^2\}$$

$$\le 256 \left(1 + \frac{93\gamma\mu}{50}\right)^{t-s} \max\{\|\theta_s\|^2, \|\theta_0\|^2\}.$$

$\square$

**Lemma C.3.** *When we start Two-Sample Quadratic Perceptron Algorithm, it runs through the sequence of steps: $i_{1,1}, \ldots, i_{1,s_1}, i_{2,1}, \ldots, i_{2,s_2}, \ldots$ and the corresponding size of groups $s_1, s_2, \ldots$, where $i_{i,j} \in \{1, 2, 3\}$, and 1 corresponds to the option with matrix $\mathbf{A}_1$, 2 corresponds to the option with matrix $\mathbf{A}_2$, and 3 corresponds to the "brake" option when the algorithm stops. Moreover, we define $q_1, q_2, \ldots$, where $q_i \in \{1, 2, 3\}$, as the type of options of the corresponding group.*

*Let $t_i := \sum\limits_{j=1}^{i-1} s_j$ and $\bar{t}_i := \sum\limits_{j \in [i-1] \text{ s.t. } q_j = 2} s_j$.*

*Then, on the set $\Omega_T$ from Lemma C.4 and for the choice of $\sigma$ in Theorem 5.1,*

$$\left\| \theta_{t_j} \right\|^2 \leq 1024 \left( 1 + \frac{93\gamma\mu}{50} \right)^{\bar{t}_j} \|\theta_0\|^2, \tag{50}$$

*for all $j \geq 2$ such that $t_j \leq T - 1$.*

*Proof.* Using Lemma C.2 and mathematical induction, we will prove that either

$$\left\| \theta_{t_j - 1} \right\|^2 \leq 64 \times \left( 1 + \frac{1}{T} \right)^{t_j - \bar{t}_j} \left( 1 + \frac{93\gamma\mu}{50} \right)^{\bar{t}_j - \mathbf{1}[q_{j-1}=2]} \|\theta_0\|^2 \tag{51}$$

or

$$\left\| \theta_{t_j} \right\|^2 \leq 64 \times \left( 1 + \frac{1}{T} \right)^{t_j - \bar{t}_j} \left( 1 + \frac{93\gamma\mu}{50} \right)^{\bar{t}_j} \|\theta_0\|^2 \tag{52}$$

holds for all $j \geq 2$ such that $t_j \leq T - 1$. For $j = 2$, (51) is true if $t_2 \leq T - 1$. Indeed, if $s_1 = 1$, then

$$\left\| \theta_{t_2 - 1} \right\|^2 = \left\| \theta_{t_1} \right\|^2 = \|\theta_0\|^2.$$

If $s_1 = 2$ and $q_1 = 1$, then

$$\left\| \theta_{t_2 - 1} \right\|^2 = \left\| \theta_{s_1 - 1} \right\|^2 \leq \left( 1 + \frac{1}{T} \right) \max\{ \|\theta_0\|^2, \|\theta_0\|^2 \} = \left( 1 + \frac{1}{T} \right) \|\theta_0\|^2$$

due to (32). If $s_1 = 3$ and $q_1 = 1$, then

$$\left\| \theta_{t_2 - 1} \right\|^2 = \left\| \theta_{s_1 - 1} \right\|^2 \leq \left( 1 + \frac{1}{T} \right) \max\{ \|\theta_{s_2 - 2}\|^2, \|\theta_0\|^2 \}$$

$$\leq \left( 1 + \frac{1}{T} \right) \max \left\{ \left( 1 + \frac{1}{T} \right) \max\{ \|\theta_{s_1 - 3}\|^2, \|\theta_0\|^2 \}, \|\theta_0\|^2 \right\} = \left( 1 + \frac{1}{T} \right)^2 \|\theta_0\|^2,$$

and so on. Thus, if $s_1 \geq 2$ and $q_1 = 1$, then

$$\left\| \theta_{t_2 - 1} \right\|^2 \leq \left( 1 + \frac{1}{T} \right)^{s_1 - 1} \|\theta_0\|^2 \leq \left( 1 + \frac{1}{T} \right)^{t_2} \|\theta_0\|^2$$

$$\leq 64 \times \left( 1 + \frac{1}{T} \right)^{t_2 - \bar{t}_2} \left( 1 + \frac{93\gamma\mu}{50} \right)^{\bar{t}_2 - \mathbf{1}[q_1 = 2]} \|\theta_0\|^2$$

and (51) holds with $j = 2$ since $t_2 = s_1$, $\bar{t}_2 = 0$, and $q_1 = 1$.

If $s_1 \geq 2$ and $q_1 = 2$, then, similarly,

$$\left\| \theta_{t_2 - 1} \right\|^2 \leq \left( 1 + \frac{11}{8}\gamma\mu \right)^{s_1 - 1} \|\theta_0\|^2 \leq 64 \times \left( 1 + \frac{1}{T} \right)^{t_2 - \bar{t}_2} \left( 1 + \frac{93\gamma\mu}{50} \right)^{\bar{t}_2 - \mathbf{1}[q_1 = 2]} \|\theta_0\|^2$$

due to (33), $t_2 = \bar{t}_2 = s_1$, and $q_1 = 2$. We now proceed to the next step of the induction assuming that $t_{j+1} \leq T - 1$ (if $t_{j+1} > T - 1$, then the proof is not needed since we consider the set of $j \geq 1$ such that $t_j \leq T - 1$).

(Case 1): If (51) holds and $q_{j-1} = 2$, then $t_{j+1} = s_j + t_j$, $q_j = 1$, $\bar{t}_{j+1} = \bar{t}_j$, and

$$\left\|\theta_{t_{j+1}}\right\|^2 = \left\|\theta_{t_j-1+2}\right\|^2 \le \max\left\{\left(1 + \frac{93\gamma\mu}{50}\right)\left\|\theta_{t_j-1}\right\|^2, 64\left\|\theta_0\right\|^2\right\}.$$

due to (35) from Lemma C.2. Using (51),

$$\left\|\theta_{t_{j+1}}\right\|^2 \le \max\left\{\left(1 + \frac{93\gamma\mu}{50}\right) \times 64 \times \left(1 + \frac{1}{T}\right)^{t_j - \bar{t}_j}\left(1 + \frac{93\gamma\mu}{50}\right)^{\bar{t}_j - \mathbf{1}[q_{j-1}=2]}\left\|\theta_0\right\|^2, 64\left\|\theta_0\right\|^2\right\}$$

$$= \left(1 + \frac{93\gamma\mu}{50}\right) \times 64 \times \left(1 + \frac{1}{T}\right)^{t_j - \bar{t}_{j+1}}\left(1 + \frac{93\gamma\mu}{50}\right)^{\bar{t}_{j+1} - 1}\left\|\theta_0\right\|^2$$

$$= 64 \times \left(1 + \frac{1}{T}\right)^{t_j - \bar{t}_{j+1}}\left(1 + \frac{93\gamma\mu}{50}\right)^{\bar{t}_{j+1}}\left\|\theta_0\right\|^2.$$

If $s_j = 1$, using $t_{j+1} \ge t_j$, we get

$$\left\|\theta_{t_{j+1}}\right\|^2 = 64 \times \left(1 + \frac{1}{T}\right)^{t_{j+1} - \bar{t}_{j+1}}\left(1 + \frac{93\gamma\mu}{50}\right)^{\bar{t}_{j+1}}\left\|\theta_0\right\|^2.$$

and (52) with $j \to j + 1$. Similarly, if $s_j = 2$, we have

$$\left\|\theta_{t_{j+1}-1}\right\|^2 = \left\|\theta_{t_j+2-1}\right\|^2 = \left\|\theta_{t_j+1}\right\|^2 \le 64 \times \left(1 + \frac{1}{T}\right)^{t_j - \bar{t}_{j+1}}\left(1 + \frac{93\gamma\mu}{50}\right)^{\bar{t}_{j+1}}\left\|\theta_0\right\|^2$$

$$\le 64 \times \left(1 + \frac{1}{T}\right)^{t_{j+1} - \bar{t}_{j+1}}\left(1 + \frac{93\gamma\mu}{50}\right)^{\bar{t}_{j+1}}\left\|\theta_0\right\|^2$$

$$= 64 \times \left(1 + \frac{1}{T}\right)^{t_{j+1} - \bar{t}_{j+1}}\left(1 + \frac{93\gamma\mu}{50}\right)^{\bar{t}_{j+1} - \mathbf{1}[q_j=2]}\left\|\theta_0\right\|^2$$

and we get (51) with $j \to j + 1$. If $s_j > 2$, we have

$$\left\|\theta_{t_{j+1}-1}\right\|^2 = \left\|\theta_{t_j+1+(s_j-2)}\right\|^2 \le \left(1 + \frac{1}{T}\right)\max\{\left\|\theta_{t_j+1+(s_j-3)}\right\|^2, \left\|\theta_0\right\|^2\}$$

due to (32) from Lemma C.2. If $\left\|\theta_0\right\|^2 \ge \left\|\theta_{t_j+1+(s_j-3)}\right\|^2$, we have proved (51) with $j \to j + 1$. Otherwise,

$$\left\|\theta_{t_{j+1}-1}\right\|^2 = \left\|\theta_{t_j+1+(s_j-2)}\right\|^2 \le \left(1 + \frac{1}{T}\right)\left\|\theta_{t_j+1+(s_j-3)}\right\|^2$$

and, repeating the same steps,

$$\left\|\theta_{t_{j+1}-1}\right\|^2 = \left\|\theta_{t_j+1+(s_j-2)}\right\|^2 \le \left(1 + \frac{1}{T}\right)^{s_j-2}\left\|\theta_{t_j+1}\right\|^2$$

$$\le \left(1 + \frac{1}{T}\right)^{s_j-2} \times 64 \times \left(1 + \frac{1}{T}\right)^{t_j - \bar{t}_{j+1}}\left(1 + \frac{93\gamma\mu}{50}\right)^{\bar{t}_{j+1}}\left\|\theta_0\right\|^2$$

$$= 64 \times \left(1 + \frac{1}{T}\right)^{t_{j+1}-2-\bar{t}_{j+1}}\left(1 + \frac{93\gamma\mu}{50}\right)^{\bar{t}_{j+1} - \mathbf{1}[q_j=2]}\left\|\theta_0\right\|^2$$

$$\le 64 \times \left(1 + \frac{1}{T}\right)^{t_{j+1}-\bar{t}_{j+1}}\left(1 + \frac{93\gamma\mu}{50}\right)^{\bar{t}_{j+1} - \mathbf{1}[q_j=2]}\left\|\theta_0\right\|^2,$$

and we obtain (51) with $j \to j + 1$.

(Case 2): If (51) holds and $q_{j-1} = 1$, then $t_{j+1} = s_j + t_j$, $q_j = 2$, $\bar{t}_{j+1} = \bar{t}_j + s_j$, and

$$\left\|\theta_{t_{j+1}}\right\|^2 = \left\|\theta_{t_j-1+2}\right\|^2 \le \max\left\{\left(1 + \frac{93\gamma\mu}{50}\right)\left\|\theta_{t_j-1}\right\|^2, 64\left\|\theta_0\right\|^2\right\}$$

due to (34) from Lemma C.2. Next,

$$\left\|\theta_{t_j+1}\right\|^2 \le \max\left\{\left(1+\frac{93\gamma\mu}{50}\right) \times 64 \times \left(1+\frac{1}{T}\right)^{t_j-\bar{t}_j}\left(1+\frac{93\gamma\mu}{50}\right)^{\bar{t}_j-\mathbf{1}[q_{j-1}=2]}\|\theta_0\|^2, 64\|\theta_0\|^2\right\}$$

$$= 64 \times \left(1+\frac{1}{T}\right)^{t_{j+1}-\bar{t}_{j+1}}\left(1+\frac{93\gamma\mu}{50}\right)^{\bar{t}_j+1}\|\theta_0\|^2.$$

If $s_j = 1$, then

$$\left\|\theta_{t_{j+1}}\right\|^2 \le 64 \times \left(1+\frac{1}{T}\right)^{t_{j+1}-\bar{t}_{j+1}}\left(1+\frac{93\gamma\mu}{50}\right)^{\bar{t}_{j+1}}\|\theta_0\|^2.$$

and we obtain (52) with $j \to j+1$. If $s_{j+1} \ge 2$, then similarly to the previous case and using (33),

$$\left\|\theta_{t_{j+1}-1}\right\|^2 \le \left(1+\frac{11}{8}\gamma\mu\right)^{s_j-2} \times 64 \times \left(1+\frac{1}{T}\right)^{t_{j+1}-\bar{t}_{j+1}}\left(1+\frac{93\gamma\mu}{50}\right)^{\bar{t}_j+1}\|\theta_0\|^2$$

$$\le 64 \times \left(1+\frac{1}{T}\right)^{t_{j+1}-\bar{t}_{j+1}}\left(1+\frac{93\gamma\mu}{50}\right)^{\bar{t}_j+(s_j-2)+1}\|\theta_0\|^2$$

$$= 64 \times \left(1+\frac{1}{T}\right)^{t_{j+1}-\bar{t}_{j+1}}\left(1+\frac{93\gamma\mu}{50}\right)^{\bar{t}_{j+1}-\mathbf{1}[q_j=2]}\|\theta_0\|^2$$

and we obtain (51) with $j \to j+1$.

(Case 3): If (52) holds and $q_j = 1$, then

$$\left\|\theta_{t_{j+1}-1}\right\|^2 = \left\|\theta_{t_j+(s_j-1)}\right\|^2.$$

Notice that

$$\left\|\theta_{t_{j+1}-1}\right\|^2 = \left\|\theta_{t_j+(s_j-1)}\right\|^2 \le \left(1+\frac{1}{T}\right)^{s_j-1} \times 64 \times \left(1+\frac{1}{T}\right)^{t_j-\bar{t}_j}\left(1+\frac{93\gamma\mu}{50}\right)^{\bar{t}_j}\|\theta_0\|^2.$$

due to (32). Thus,

$$\left\|\theta_{t_{j+1}-1}\right\|^2 = \left\|\theta_{t_j+(s_j-1)}\right\|^2 \le 64 \times \left(1+\frac{1}{T}\right)^{t_j+(s_j-1)-\bar{t}_j}\left(1+\frac{93\gamma\mu}{50}\right)^{\bar{t}_j}\|\theta_0\|^2$$

$$= 64 \times \left(1+\frac{1}{T}\right)^{t_{j+1}-1-\bar{t}_j}\left(1+\frac{93\gamma\mu}{50}\right)^{\bar{t}_j}\|\theta_0\|^2$$

$$\le 64 \times \left(1+\frac{1}{T}\right)^{t_{j+1}-\bar{t}_{j+1}}\left(1+\frac{93\gamma\mu}{50}\right)^{\bar{t}_{j+1}-\mathbf{1}[q_j=2]}\|\theta_0\|^2,$$

where we use $\bar{t}_{j+1} = \bar{t}_j$, and we obtain (51) with $j \to j+1$.

(Case 4): Finally, if (52) holds and $q_j = 2$, then

$$\left\|\theta_{t_{j+1}-1}\right\|^2 = \left\|\theta_{t_j+(s_j-1)}\right\|^2 \le \left(1+\frac{11}{8}\gamma\mu\right)^{s_j-1} \times 64 \times \left(1+\frac{1}{T}\right)^{t_j-\bar{t}_j}\left(1+\frac{93\gamma\mu}{50}\right)^{\bar{t}_j}\|\theta_0\|^2.$$

due to (33). Next,

$$\left\|\theta_{t_{j+1}-1}\right\|^2 = \left\|\theta_{t_j+(s_j-1)}\right\|^2 \le 64 \times \left(1+\frac{1}{T}\right)^{t_j-\bar{t}_j}\left(1+\frac{93\gamma\mu}{50}\right)^{\bar{t}_j+s_j-1}\|\theta_0\|^2$$

$$= 64 \times \left(1+\frac{1}{T}\right)^{t_{j+1}-\bar{t}_{j+1}}\left(1+\frac{93\gamma\mu}{50}\right)^{\bar{t}_{j+1}-\mathbf{1}[q_j=2]}\|\theta_0\|^2,$$

where we use $\bar{t}_{j+1} = \bar{t}_{j+1}$, and we obtain (51) with $j \to j+1$.

We have proved the next step of the mathematical induction. If (52) holds, then

$$\left\| \theta_{t_j} \right\|^2 \leq 256 \times \left( 1 + \frac{1}{T} \right)^{t_j} \left( 1 + \frac{93\gamma\mu}{50} \right)^{\bar{t}_j} \left\| \theta_0 \right\|^2 .$$

Otherwise, (51) holds, then

$$\left\| \theta_{t_j} \right\|^2 \overset{(36)}{\leq} 4 \left\| \theta_{t_j - 1} \right\|^2 \leq 256 \times \left( 1 + \frac{1}{T} \right)^{t_j} \left( 1 + \frac{93\gamma\mu}{50} \right)^{\bar{t}_j} \left\| \theta_0 \right\|^2 ,$$

meaning that

$$\left\| \theta_{t_j} \right\|^2 \leq 256 \times \left( 1 + \frac{1}{T} \right)^{t_j} \left( 1 + \frac{93\gamma\mu}{50} \right)^{\bar{t}_j} \left\| \theta_0 \right\|^2 ,$$

holds for all $j \geq 2$ such that $t_j \leq T - 1$. Since $\left( 1 + \frac{1}{T} \right)^{t_j} \leq \left( 1 + \frac{1}{T} \right)^{T-1} \leq 4$ if $t_j \leq T - 1$,

$$\left\| \theta_{t_j} \right\|^2 \leq 1024 \left( 1 + \frac{93\gamma\mu}{50} \right)^{\bar{t}_j} \left\| \theta_0 \right\|^2 ,$$

for all $j \geq 2$ such that $t_j \leq T - 1$. $\qquad\square$

**Lemma C.4.** *In Two-Sample Quadratic Perceptron Algorithm, with probability at least $1 - 5T\delta$, we have*

$$\left\| \xi_t \right\| \leq \beta := \sigma \sqrt{\max\{6d, 4\log(1/\delta)\}} \tag{53}$$

*for all $0 \leq t \leq T - 1$,*

$$\left| \eta_{\lambda,s,t,v} + \sum_{i=1}^{s} (1 + \gamma\lambda)^{i-1} \langle \xi_{t+s-i}, v \rangle \right| > \alpha_s := \alpha(1 + \gamma\lambda)^{s/2}, \alpha := \frac{\sigma\delta}{20\sqrt{7} \log\left( \frac{1}{\gamma\lambda_1} \right)}$$

*for all $0 \leq t \leq T - 1$, $s \geq 1$, $\lambda \in \{\lambda_1, \lambda_2\}$, vectors $v \in \{v_{1,+}, v_{2,+}\}$, where $\eta_{\lambda,s,t,v} := (1 + \gamma\lambda)^s \langle \theta_t, v \rangle$. Equivalently,*

$$\mathbb{P}\left( \Omega_T \right) \geq 1 - 5T\delta,$$

*where*

$$\Omega_T := \bigcap_{t=0}^{T-1} \{ \|\xi_t\| \leq \beta \} \bigcap_{t=0}^{T-1} \bigcap_{v \in \{v_{1,+}, v_{2,+}\}} \bigcap_{\lambda \in \{\lambda_1, \lambda_2\}} \bigcap_{s \geq 1} \left\{ \left| \eta_{\lambda,s,t,v} + \sum_{i=1}^{s} (1 + \gamma\lambda)^{i-1} \langle \xi_{t+s-i}, v \rangle \right| > \alpha_s \right\}.$$

*Proof.* (Part 1): For all $t \geq 0$, we have

$$\mathbb{P}\left( \|\xi_t\| \geq \beta \right) = \mathbb{P}\left( \|\xi_t\|^2 \geq \beta^2 \right).$$

for all $\beta \geq 0$. Without loss of generality assume that $\sigma = 1$. Then, $\|\xi_t\|^2$ is from the standard chi-squared distribution, and using Chernoff bound, we have

$$\mathbb{P}\left( \|\xi_t\| \geq \beta \right) \leq \inf_{0 < p < \frac{1}{2}} \left[ \mathbb{E}\left[ \exp(p \|\xi_t\|^2) \right] \exp(-p\beta^2) \right] = \inf_{0 < p < \frac{1}{2}} \left[ (1 - 2p)^{-d/2} \exp(-p\beta^2) \right]$$

$$= (ze^{1-z})^{d/2} \leq e^{-zd/4}$$

for all $\beta = \sqrt{zd}$ and $z \geq 6$. Taking $z = \max\{6, \frac{4}{d} \log \frac{1}{\delta}\}$, we get $\mathbb{P}\left( \|\xi_t\| > \beta \right) \leq \delta$ for $\beta = \sqrt{\max\{6d, 4\log(1/\delta)\}}$ if $\sigma = 1$. Clearly, we shoould take $\beta = \sigma \sqrt{\max\{6d, 4\log(1/\delta)\}}$ if $\sigma > 0$. (Part 2): Next, for all $t \geq 0$, $\lambda, \alpha > 0$, $v \in B(0, 1)$, and $s \geq 1$,

$$\mathbb{P}\left( \left| \eta_{\lambda,s,t,v} + \sum_{i=1}^{s} (1 + \gamma\lambda)^{i-1} \langle \xi_{t+s-i}, v \rangle \right| \leq \alpha \right) = \mathbb{P}\left( |\eta_{\lambda,s,t,v} + \xi| \leq \alpha \right)$$

$$= \mathbb{E}_{\eta_{\lambda,s,t,v}} \left[ \Phi \left( \frac{\alpha - \eta_{\lambda,s,t,v}}{\bar{\sigma}} \right) - \Phi \left( \frac{-\alpha - \eta_{\lambda,s,t,v}}{\bar{\sigma}} \right) \right]$$

with $\xi \sim \mathcal{N}\left(0, \bar{\sigma}^2\right)$ and $\bar{\sigma}^2 := \sigma^2 \sum_{i=0}^{s-1}(1+\gamma\lambda)^{2i}$ since $\{\xi_i\}$ are i.i.d. and $\{\xi_i\}_{i \geq t}$ are independent of random variable $\eta_{\lambda,s,t,v}$. $\Phi$ is the standard normal cumulative distribution function (CDF). Next,

$$\Phi \left( \frac{\alpha - \eta_{\lambda,s,t,v}}{\bar{\sigma}} \right) - \Phi \left( \frac{-\alpha - \eta_{\lambda,s,t,v}}{\bar{\sigma}} \right) = \int_{\frac{-\alpha-\eta_{\lambda,s,t,v}}{\bar{\sigma}}}^{\frac{\alpha-\eta_{\lambda,s,t,v}}{\bar{\sigma}}} \left( \frac{1}{\sqrt{2\pi}} e^{-t^2/2} \right) dt \leq \frac{\frac{\alpha-\eta_{\lambda,s,t,v}}{\bar{\sigma}} - \frac{-\alpha-\eta_{\lambda,s,t,v}}{\bar{\sigma}}}{\sqrt{2\pi}} = \frac{2\alpha}{\sqrt{2\pi}\bar{\sigma}} \leq \frac{\alpha}{\bar{\sigma}}.$$

Thus,

$$\mathbb{P}\left( \bigcup_{s \geq 1} \left\{ \left| \eta_{\lambda,s,t,v} + \sum_{i=1}^{s}(x+\gamma\lambda)^{i-1} \langle \xi_{t+s-i}, v \rangle \right| \leq \alpha_s \right\} \right) \leq \frac{1}{\sigma} \sum_{s=1}^{\infty} \frac{\alpha_s}{\sqrt{\sum_{i=0}^{s-1}(1+\gamma\lambda)^{2i}}}$$

$$= \frac{\sqrt{(1+\gamma\lambda)^2 - 1}}{\sigma} \sum_{s=1}^{\infty} \frac{\alpha_s}{\sqrt{(1+\gamma\lambda)^{2s} - 1}} \leq \frac{\sqrt{2\gamma\lambda + \gamma^2\lambda^2}}{\sigma} \sum_{s=1}^{\infty} \frac{\alpha_s}{(1+\gamma\lambda)^s - 1}.$$

Since $\gamma = \frac{1}{4\sqrt{d}}$ and $\sqrt{2d+1} \geq \lambda \geq \sqrt{2d}$, we have $2\gamma\lambda + \gamma^2\lambda^2 \leq 7\gamma^2\lambda^2$. We can use Lemma D.2, since $\gamma\lambda \leq \frac{11}{25}$:

$$\mathbb{P}\left( \bigcup_{s \geq 1} \left\{ \left| \eta_{\lambda,s,t,v} + \sum_{i=1}^{s}(x+\gamma\lambda)^{i-1} \langle \xi_{t+s-i}, v \rangle \right| \leq \alpha_s \right\} \right)$$

$$\leq \frac{\sqrt{7}\gamma\lambda\alpha}{\sigma} \sum_{s=1}^{\infty} \frac{(1+\gamma\lambda)^{s/2}}{(1+\gamma\lambda)^s - 1} \leq \frac{\sqrt{7}\gamma\lambda\alpha}{\sigma} \times \frac{20}{\gamma\lambda} \log \left( \frac{1}{\gamma\lambda} \right) = \frac{20\sqrt{7}\alpha}{\sigma} \log \left( \frac{1}{\gamma\lambda} \right)$$

for $\alpha_s := \alpha(1+\gamma\lambda)^{s/2}$ and any $\alpha > 0$. Thus, we should take $\alpha = \frac{\sigma\delta}{20\sqrt{7}\log\left(\frac{1}{\gamma\lambda_1}\right)}$ to ensure that the probability less or equal $\delta$ since $\lambda \in \{\lambda_1, \lambda_2\}$ and $\lambda_1 \leq \lambda_2$.

Finally, using the union bound,

$$\mathbb{P}\left( \bigcap_{t=0}^{T-1} \{ \|\xi_t\| \leq \beta \} \bigcap_{t=0}^{T-1} \bigcap_{v \in \{v_{1,+}, v_{2,+}\}} \bigcap_{\lambda \in \{\lambda_1, \lambda_2\}} \bigcap_{s \geq 1} \left| \eta_{t,v,\lambda,s} + \sum_{i=1}^{s}(1+\gamma\lambda)^{i-1} \langle \xi_{t+s-i}, v \rangle \right| > \alpha_s \right)$$

$$\geq 1 - (T + T \times 2 \times 2)\delta \geq 1 - 5T\delta.$$

$\square$

**Lemma C.5.** *The matrices $\mathbf{A}_1$ and $\mathbf{A}_2$ satisfy*

$$\max_{x \in Q} \|(\mathbf{I} + \gamma\mathbf{A}_2)(\mathbf{I} + \gamma\mathbf{A}_1)x\| \leq \sqrt{\frac{9\gamma\mu}{5} + 1 + \frac{\delta\gamma}{2}},$$

*for all $\gamma \leq \frac{1}{4\sqrt{d}}$, $\mu \leq 0.1$, and $\delta \geq 0$, where*

$$Q := \{ x \in \mathbb{R}^{d+2} \mid \|x\| = 1, x^\top \mathbf{A}_1 x \leq 0, x^\top (\mathbf{I} + \gamma\mathbf{A}_1)\mathbf{A}_2(\mathbf{I} + \gamma\mathbf{A}_1)x \leq \delta \}.$$

*Proof.* Using the method of Lagrange multipliers, we have

$$\max_{x \in Q} x^\top (\mathbf{I} + \gamma\mathbf{A}_1)(\mathbf{I} + \gamma\mathbf{A}_2)^2(\mathbf{I} + \gamma\mathbf{A}_1)x$$

$$= \max_{x \in \mathbb{R}^{d+2}, \|x\|=1} \inf_{\eta \geq 0, \xi \geq 0} \left[ x^\top (\mathbf{I} + \gamma\mathbf{A}_1)(\mathbf{I} + \gamma\mathbf{A}_2)^2(\mathbf{I} + \gamma\mathbf{A}_1)x - \eta x^\top \mathbf{A}_1 x - \xi x^\top (\mathbf{I} + \gamma\mathbf{A}_1)\mathbf{A}_2(\mathbf{I} + \gamma\mathbf{A}_1)x + \xi\delta \right]$$

$$\leq \inf_{\eta \geq 0, \xi \geq 0} \max_{x \in \mathbb{R}^{d+2}, \|x\|=1} \left[ x^\top (\mathbf{I} + \gamma\mathbf{A}_1)(\mathbf{I} + \gamma\mathbf{A}_2)^2(\mathbf{I} + \gamma\mathbf{A}_1)x - \eta x^\top \mathbf{A}_1 x - \xi x^\top (\mathbf{I} + \gamma\mathbf{A}_1)\mathbf{A}_2(\mathbf{I} + \gamma\mathbf{A}_1)x + \xi\delta \right]$$

$$\leq \max_{x\in\mathbb{R}^{d+2},\|x\|=1}\left[x^\top\left(\mathbf{I}+\gamma\mathbf{A}_1\right)\left(\mathbf{I}+\gamma\mathbf{A}_2\right)^2\left(\mathbf{I}+\gamma\mathbf{A}_1\right)x - \eta x^\top\mathbf{A}_1 x - \xi x^\top\left(\mathbf{I}+\gamma\mathbf{A}_1\right)\mathbf{A}_2\left(\mathbf{I}+\gamma\mathbf{A}_1\right)x + \xi\delta\right]$$

for all $\eta\geq 0, \xi\geq 0$. Let us take $\eta=\frac{\gamma}{2}$ and $\xi=\frac{\gamma}{2}$. Then

$$\max_{x\in Q} x^\top\left(\mathbf{I}+\gamma\mathbf{A}_1\right)\left(\mathbf{I}+\gamma\mathbf{A}_2\right)^2\left(\mathbf{I}+\gamma\mathbf{A}_1\right)x$$

$$\leq \max_{x\in\mathbb{R}^{d+2},\|x\|=1}\left[x^\top\left(\mathbf{I}+\gamma\mathbf{A}_1\right)\left(\mathbf{I}+\gamma\mathbf{A}_2\right)^2\left(\mathbf{I}+\gamma\mathbf{A}_1\right)x - \frac{\gamma}{2}x^\top\mathbf{A}_1 x - \frac{\gamma}{2}x^\top\left(\mathbf{I}+\gamma\mathbf{A}_1\right)\mathbf{A}_2\left(\mathbf{I}+\gamma\mathbf{A}_1\right)x\right] + \frac{\delta\gamma}{2} \quad (54)$$

$$= \max_{x\in\mathbb{R}^{d+2},\|x\|=1}\left\{x^\top\left[\left(\mathbf{I}+\gamma\mathbf{A}_1\right)\left(\mathbf{I}+\gamma\mathbf{A}_2\right)^2\left(\mathbf{I}+\gamma\mathbf{A}_1\right)-\frac{\gamma}{2}\mathbf{A}_1-\frac{\gamma}{2}\left(\mathbf{I}+\gamma\mathbf{A}_1\right)\mathbf{A}_2\left(\mathbf{I}+\gamma\mathbf{A}_1\right)\right]x\right\} + \frac{\delta\gamma}{2}.$$

Notice that

$$\mathbf{I}+\gamma\mathbf{A}_1 = \begin{bmatrix} 1 & 0 & \gamma a_1^\top \\ 0 & 1 & \gamma a_1^\top\mathbf{P}^\top \\ \gamma a_1 & \gamma\mathbf{P}a_1 & \mathbf{I}_d \end{bmatrix},$$

$$\mathbf{I}+\gamma\mathbf{A}_2 = \begin{bmatrix} 1 & 0 & \gamma a_2^\top \\ 0 & 1 & \gamma a_2^\top\mathbf{P}^\top \\ \gamma a_2 & \gamma\mathbf{P}a_2 & \mathbf{I}_d \end{bmatrix},$$

and

$$(\mathbf{I}+\gamma\mathbf{A}_2)(\mathbf{I}+\gamma\mathbf{A}_1) = \begin{bmatrix} 1+\gamma^2 a_2^\top a_1 & \gamma^2 a_2^\top\mathbf{P}a_1 & \gamma(a_1^\top+a_2^\top) \\ \gamma^2 a_2^\top\mathbf{P}^\top a_1 & 1+\gamma^2 a_2^\top\mathbf{P}^\top\mathbf{P}a_1 & \gamma(a_1^\top\mathbf{P}^\top+a_2^\top\mathbf{P}^\top) \\ \gamma(a_2+a_1) & \gamma(\mathbf{P}a_2+\mathbf{P}a_1) & \gamma^2(a_2 a_1^\top+\mathbf{P}a_2 a_1^\top\mathbf{P}^\top)+\mathbf{I}_d \end{bmatrix}$$

$$= \begin{bmatrix} 1+\gamma^2 a_2^\top a_1 & \gamma^2 a_2^\top\mathbf{P}a_1 & \gamma(a_1^\top+a_2^\top) \\ \gamma^2 a_2^\top\mathbf{P}^\top a_1 & 1+\gamma^2 a_2^\top a_1 & \gamma(a_1^\top\mathbf{P}^\top+a_2^\top\mathbf{P}^\top) \\ \gamma(a_2+a_1) & \gamma(\mathbf{P}a_2+\mathbf{P}a_1) & \gamma^2(a_2 a_1^\top+\mathbf{P}a_2 a_1^\top\mathbf{P}^\top)+\mathbf{I}_d \end{bmatrix}$$

$$= \mathbf{I} + \begin{bmatrix} \gamma^2 a_2^\top a_1 & \gamma^2 a_2^\top\mathbf{P}a_1 & \gamma(a_1^\top+a_2^\top) \\ \gamma^2 a_2^\top\mathbf{P}^\top a_1 & \gamma^2 a_2^\top a_1 & \gamma(a_1^\top\mathbf{P}^\top+a_2^\top\mathbf{P}^\top) \\ \gamma(a_2+a_1) & \gamma(\mathbf{P}a_2+\mathbf{P}a_1) & \gamma^2(a_2 a_1^\top+\mathbf{P}a_2 a_1^\top\mathbf{P}^\top) \end{bmatrix}$$

since $\mathbf{P}^\top\mathbf{P}=\mathbf{I}$. Moreover,

$$(\mathbf{I}+\gamma\mathbf{A}_1)\mathbf{A}_2(\mathbf{I}+\gamma\mathbf{A}_1)$$

$$= (\mathbf{I}+\gamma\mathbf{A}_1)\begin{bmatrix} 0 & 0 & a_2^\top \\ 0 & 0 & a_2^\top\mathbf{P}^\top \\ a_2 & \mathbf{P}a_2 & \mathbf{0}_d \end{bmatrix}\begin{bmatrix} 1 & 0 & \gamma a_1^\top \\ 0 & 1 & \gamma a_1^\top\mathbf{P}^\top \\ \gamma a_1 & \gamma\mathbf{P}a_1 & \mathbf{I}_d \end{bmatrix}$$

$$= \begin{bmatrix} 1 & 0 & \gamma a_1^\top \\ 0 & 1 & \gamma a_1^\top\mathbf{P}^\top \\ \gamma a_1 & \gamma\mathbf{P}a_1 & \mathbf{I}_d \end{bmatrix}\begin{bmatrix} \gamma a_2^\top a_1 & \gamma a_2^\top\mathbf{P}a_1 & a_2^\top \\ \gamma a_2^\top\mathbf{P}^\top a_1 & \gamma a_2^\top a_1 & a_2^\top\mathbf{P}^\top \\ a_2 & \mathbf{P}a_2 & \gamma(a_2 a_1^\top+\mathbf{P}a_2 a_1^\top\mathbf{P}^\top) \end{bmatrix}$$

$$= \begin{bmatrix} 2\gamma a_2^\top a_1 & \gamma a_2^\top\mathbf{P}a_1+\gamma a_1^\top\mathbf{P}a_2 & a_2^\top+\gamma^2 a_1^\top\left(a_2 a_1^\top+\mathbf{P}a_2 a_1^\top\mathbf{P}^\top\right) \\ \gamma a_2^\top\mathbf{P}^\top a_1+\gamma a_1^\top\mathbf{P}^\top a_2 & 2\gamma a_2^\top a_1 & a_2^\top\mathbf{P}^\top+\gamma^2 a_1^\top\mathbf{P}^\top\left(a_2 a_1^\top+\mathbf{P}a_2 a_1^\top\mathbf{P}^\top\right) \\ a_2+\gamma^2\left(a_1 a_2^\top+\mathbf{P}a_1 a_2^\top\mathbf{P}^\top\right)a_1 & \mathbf{P}a_2+\gamma^2\left(a_1 a_2^\top+\mathbf{P}a_1 a_2^\top\mathbf{P}^\top\right)\mathbf{P}a_1 & \gamma(a_1 a_2^\top+a_2 a_1^\top+\mathbf{P}a_1 a_2^\top\mathbf{P}^\top+\mathbf{P}a_2 a_1^\top\mathbf{P}^\top) \end{bmatrix}.$$

Let us define

$$\mathbf{B} := \left(\mathbf{I}+\gamma\mathbf{A}_1\right)\left(\mathbf{I}+\gamma\mathbf{A}_2\right)^2\left(\mathbf{I}+\gamma\mathbf{A}_1\right)-\frac{\gamma}{2}\mathbf{A}_1-\frac{\gamma}{2}\left(\mathbf{I}+\gamma\mathbf{A}_1\right)\mathbf{A}_2\left(\mathbf{I}+\gamma\mathbf{A}_1\right)-\mathbf{I},$$

then we have to bound

$$1+\max_{x\in\mathbb{R}^{d+2},\|x\|=1}\left\{x^\top\mathbf{B}x\right\}.$$

Substituting the particular choices of $a_1$ and $a_2$ from (5) into our previous calculations, we can use Lemma D.3 with

$$\mathbf{A} = \begin{bmatrix} \gamma^4 (d+\mu)^2 + \gamma^2\mu^2 + \gamma^2 (d+\mu) + (\gamma^2 (d+\mu) - 1)^2 - 1 & \gamma^2 (d+\mu)(2\gamma^2 (d+\mu) - 1) & \gamma (d\gamma^2\mu + d\gamma^2 + 2\gamma^2\mu^2 + 3\gamma^2\mu - \frac{3\mu}{2}) & \gamma^3 (d\mu + d + 2\mu^2 + 3\mu) \\ \gamma^2 (d+\mu)(2\gamma^2 (d+\mu) - 1) & \gamma^4 (d+\mu)^2 + \gamma^2\mu^2 + \gamma^2 (d+\mu) + (\gamma^2 (d+\mu) - 1)^2 - 1 & \gamma^3 (d\mu + d + 2\mu^2 + 3\mu) & \gamma (d\gamma^2\mu + d\gamma^2 + 2\gamma^2\mu^2 + 3\gamma^2\mu - \frac{3\mu}{2}) \\ \gamma (d\gamma^2\mu + d\gamma^2 + 2\gamma^2\mu^2 + 3\gamma^2\mu - \frac{3\mu}{2}) & \gamma^3 (d\mu + d + 2\mu^2 + 3\mu) & \gamma^2 \cdot (2\gamma^2\mu^2 + 8\gamma^2\mu + 12\gamma^2 + \mu^2 - \mu - 2) & \gamma^2 \cdot (4\gamma^2 + \mu + 2(\mu+2)(\gamma^2(\mu+2) - 1) + 2) \\ \gamma^3 (d\mu + d + 2\mu^2 + 3\mu) & \gamma (d\gamma^2\mu + d\gamma^2 + 2\gamma^2\mu^2 + 3\gamma^2\mu - \frac{3\mu}{2}) & \gamma^2 \cdot (4\gamma^2 + \mu + 2(\mu+2)(\gamma^2(\mu+2) - 1) + 2) & \gamma^2 \cdot (2\gamma^2\mu^2 + 8\gamma^2\mu + 12\gamma^2 + \mu^2 - \mu - 2) \end{bmatrix}$$

$$a = \begin{bmatrix} \gamma^3 (d + \mu^2 + 3\mu) & \gamma^3 (d + \mu^2 + 3\mu) & \frac{\gamma^2 \cdot (4\gamma^2\mu^2 + 16\gamma^2\mu + 24\gamma^2 - \mu - 4)}{2} & \frac{\gamma^2 \cdot (4\gamma^2\mu^2 + 16\gamma^2\mu + 24\gamma^2 - \mu - 4)}{2} \end{bmatrix},$$

$$b = 2\gamma^4 (\mu + 2)^2 + 2\gamma^2 + (2\gamma^2 - 1)^2 - 1.$$

Thus,

$$1 + \max_{x \in \mathbb{R}^{d+2}, \|x\|=1} \left\{ x^\top \mathbf{B} x \right\} = 1 + \max_{x \in \mathbb{R}^5, \|x\|=1} \left\{ x^\top \mathbf{C} x \right\} = \max_{x \in \mathbb{R}^5, \|x\|=1} \left\{ x^\top (\mathbf{I} + \mathbf{C}) x \right\} \tag{55}$$

with $\mathbf{C}$ defined in Lemma D.3. We consider the characteristic polynomial of $(\mathbf{I} + \mathbf{C})$ :

$$p_5(\lambda) := \det\left(\lambda \mathbf{I} - (\mathbf{I} + \mathbf{C})\right). \tag{56}$$

Using Taylor's theorem, we have

$$p_5(\lambda) = \sum_{i=0}^{5} \frac{p_5^{(i)}(x)}{i!} (\lambda - x)^i \quad \forall x \in \mathbb{R}.$$

If we want to show that all solutions are bounded by $x$, it is sufficient to show that all derivatives $p_5^{(i)}(x)$ are positive at this point. Let us consider $x = \frac{9\gamma\mu}{5} + 1$. Clearly, we have $p_5^{(5)}(x) = 120 > 0$ for all $x \in \mathbb{R}$. We use the symbolic computation library Sympy (Meurer et al., 2017) to obtain

$$p_5^{(4)}(x)/\gamma = \gamma^3 \left(-192d^2 - 48d\mu^2 - 384d\mu - 96\mu^2\right) + \gamma \left(96d - 96\mu^2 + 96\mu\right) + 216\mu. \tag{57}$$

Using $\gamma \leq \frac{1}{2\sqrt{d}}$, we have

$$p_5^{(4)}(x)/\gamma \geq \gamma \left(48d - 108\mu^2 - \frac{24\mu^2}{d}\right) + 216\mu. \tag{58}$$

Using $d \geq \mu$, we get

$$p_5^{(4)}(x)/\gamma \geq \gamma \left(48d - 108\mu^2 - 24\mu\right) + 216\mu. \tag{59}$$

Using $d \geq 1$ and $\mu \leq 0.1$, we get

$$p_5^{(4)}(x)/\gamma \geq \frac{1113\gamma}{25} + 216\mu. \tag{60}$$

Thus, we have $p_5^{(4)}(x) > 0$. Next, we get

$$\begin{aligned} p_5^{(3)}(x)/\gamma^2 &= \gamma^6 \cdot \left(96d^4 + 48d^3\mu^2 + 384d^3\mu + 96d^2\mu^3 + 480d^2\mu^2 + 48d\mu^4 + 192d\mu^3\right) \\ &+ \gamma^4 \left(-108d^3 + 96d^2\mu^2 - 408d^2\mu + 36d\mu^4 + 144d\mu^3 - 588d\mu^2 - 24\mu^4 - 288\mu^3\right) \\ &+ \gamma^3 \left(-\frac{1728d^2\mu}{5} - \frac{432d\mu^3}{5} - \frac{3456d\mu^2}{5} - \frac{864\mu^3}{5}\right) + \gamma^2 \\ &\cdot \left(24d^2 - 51d\mu^2 + 84d\mu + 36\mu^4 + 162\mu^2\right) + \gamma \left(\frac{864d\mu}{5} - \frac{864\mu^3}{5} + \frac{864\mu^2}{5}\right) + \frac{837\mu^2}{5}. \end{aligned} \tag{61}$$

We ignore all positive terms w.r.t. $\gamma^4$ and $\gamma^6$ and obtain

$$p_5^{(3)}(x)/\gamma^2 \geq \gamma^4 \left(-108d^3 - 408d^2\mu - 588d\mu^2 - 24\mu^4 - 288\mu^3\right)$$

$$+ \gamma^3 \left(-\frac{1728d^2\mu}{5} - \frac{432d\mu^3}{5} - \frac{3456d\mu^2}{5} - \frac{864\mu^3}{5}\right) + \gamma^2 \tag{62}$$

$$\cdot \left(24d^2 - 51d\mu^2 + 84d\mu + 36\mu^4 + 162\mu^2\right) + \gamma \left(\frac{864d\mu}{5} - \frac{864\mu^3}{5} + \frac{864\mu^2}{5}\right) + \frac{837\mu^2}{5}.$$

Using $\gamma \leq \frac{1}{2\sqrt{2}\sqrt{d}}$, we have

$$p_5^{(3)}(x)/\gamma^2 \geq \gamma^2 \cdot \left(\frac{21d^2}{2} - 51d\mu^2 + 33d\mu + 36\mu^4 + \frac{177\mu^2}{2} - \frac{3\mu^4}{d} - \frac{36\mu^3}{d}\right) \tag{63}$$

$$+ \gamma \left(\frac{648d\mu}{5} - \frac{918\mu^3}{5} + \frac{432\mu^2}{5} - \frac{108\mu^3}{5d}\right) + \frac{837\mu^2}{5}.$$

Using $d \geq \mu$, we get

$$p_5^{(3)}(x)/\gamma^2 \geq \gamma^2 \cdot \left(\frac{21d^2}{2} - 51d\mu^2 + 33d\mu + 36\mu^4 - 3\mu^3 + \frac{105\mu^2}{2}\right) + \gamma \left(\frac{648d\mu}{5} - \frac{918\mu^3}{5} + \frac{324\mu^2}{5}\right) + \frac{837\mu^2}{5}. \tag{64}$$

Using $d \geq 1$ and $\mu \leq 0.1$, we get

$$p_5^{(3)}(x)/\gamma^2 \geq \gamma^2 \cdot \left(\frac{21d}{2} + 36\mu^4 + \frac{261\mu^2}{5} + \frac{279\mu}{10}\right) + \gamma \left(\frac{1161\mu^2}{25} + \frac{648\mu}{5}\right) + \frac{837\mu^2}{5}. \tag{65}$$

Thus, $p_5^{(3)}(x) > 0$. Next, we consider

$$p_5^{(2)}(x)/\gamma^3 = \gamma^7 \left(-64d^4\mu^2 - 32d^3\mu^4 - 256d^3\mu^3 - 64d^2\mu^5 - 320d^2\mu^4 - 32d\mu^6 - 128d\mu^5\right)$$

$$+ \gamma^6 \cdot \left(\frac{864d^4\mu}{5} + \frac{432d^3\mu^3}{5} + \frac{3456d^3\mu^2}{5} + \frac{864d^2\mu^4}{5} + 864d^2\mu^3 + \frac{432d\mu^5}{5} + \frac{1728d\mu^4}{5}\right) + \gamma^5$$

$$\cdot \left(56d^3\mu^2 - 24d^2\mu^4 + 240d^2\mu^3 + 32d^2\mu^2 - 12d\mu^6 + 408d\mu^4 + 64d\mu^3 + 40\mu^6 + 192\mu^5\right)$$

$$+ \gamma^4 \left(-\frac{972d^3\mu}{5} + \frac{864d^2\mu^3}{5} - \frac{3672d^2\mu^2}{5} + \frac{324d\mu^5}{5} + \frac{1296d\mu^4}{5} - \frac{5292d\mu^3}{5} - \frac{216\mu^5}{5} - \frac{2592\mu^4}{5}\right) \tag{66}$$

$$+ \gamma^3 \left(-\frac{6626d^2\mu^2}{25} - \frac{1269d\mu^4}{25} - \frac{14352d\mu^3}{25} - 52d\mu^2 - 8\mu^6 - 24\mu^5 - \frac{6538\mu^4}{25} - 64\mu^3\right)$$

$$+ \gamma^2 \cdot \left(\frac{216d^2\mu}{5} - \frac{459d\mu^3}{5} + \frac{756d\mu^2}{5} + \frac{324\mu^5}{5} + \frac{1458\mu^3}{5}\right)$$

$$+ \gamma \left(\frac{3213d\mu^2}{25} - \frac{3438\mu^4}{25} + \frac{3438\mu^3}{25} + 18\mu^2\right) + \frac{1701\mu^3}{25}.$$

We ignore all positive terms w.r.t. $\gamma^6$ and obtain

$$
\begin{aligned}
p_5^{(2)}(x)/\gamma^3 \geq{} & \gamma^7 \left(-64d^4\mu^2 - 32d^3\mu^4 - 256d^3\mu^3 - 64d^2\mu^5 - 320d^2\mu^4 - 32d\mu^6 - 128d\mu^5\right) \\
& + \gamma^5 \cdot \left(56d^3\mu^2 - 24d^2\mu^4 + 240d^2\mu^3 + 32d^2\mu^2 - 12d\mu^6 + 408d\mu^4 + 64d\mu^3 + 40\mu^6 + 192\mu^5\right) \\
& + \gamma^4 \left(-\frac{972d^3\mu}{5} + \frac{864d^2\mu^3}{5} - \frac{3672d^2\mu^2}{5} + \frac{324d\mu^5}{5} + \frac{1296d\mu^4}{5} - \frac{5292d\mu^3}{5} - \frac{216\mu^5}{5} - \frac{2592\mu^4}{5}\right) \\
& + \gamma^3 \left(-\frac{6626d^2\mu^2}{25} - \frac{1269d\mu^4}{25} - \frac{14352d\mu^3}{25} - 52d\mu^2 - 8\mu^6 - 24\mu^5 - \frac{6538\mu^4}{25} - 64\mu^3\right) \\
& + \gamma^2 \cdot \left(\frac{216d^2\mu}{5} - \frac{459d\mu^3}{5} + \frac{756d\mu^2}{5} + \frac{324\mu^5}{5} + \frac{1458\mu^3}{5}\right) \\
& + \gamma \left(\frac{3213d\mu^2}{25} - \frac{3438\mu^4}{25} + \frac{3438\mu^3}{25} + 18\mu^2\right) + \frac{1701\mu^3}{25}.
\end{aligned}
\tag{67}
$$

Using $\gamma \leq \frac{1}{2\sqrt{2}\sqrt{d}}$, we have

$$
\begin{aligned}
p_5^{(2)}(x)/\gamma^3 \geq{} & \gamma^5 \\
& \cdot \left(48d^3\mu^2 - 28d^2\mu^4 + 208d^2\mu^3 + 32d^2\mu^2 - 12d\mu^6 - 8d\mu^5 + 368d\mu^4 + 64d\mu^3 + 36\mu^6 + 176\mu^5\right) \\
& + \gamma^4 \cdot \left(\frac{864d^2\mu^3}{5} + \frac{324d\mu^5}{5} + \frac{1296d\mu^4}{5}\right) + \gamma^2 \\
& \cdot \left(\frac{189d^2\mu}{10} - \frac{459d\mu^3}{5} + \frac{297d\mu^2}{5} + \frac{324\mu^5}{5} + \frac{1593\mu^3}{10} - \frac{27\mu^5}{5d} - \frac{324\mu^4}{5d}\right) \\
& + \gamma \left(\frac{9539d\mu^2}{100} - \frac{28773\mu^4}{200} + \frac{1644\mu^3}{25} + \frac{23\mu^2}{2} - \frac{\mu^6}{d} - \frac{3\mu^5}{d} - \frac{3269\mu^4}{100d} - \frac{8\mu^3}{d}\right) + \frac{1701\mu^3}{25}.
\end{aligned}
\tag{68}
$$

Using $d \geq \mu$, we get

$$
\begin{aligned}
p_5^{(2)}(x)/\gamma^3 \geq{} & \gamma^5 \\
& \cdot \left(48d^3\mu^2 - 28d^2\mu^4 + 208d^2\mu^3 + 32d^2\mu^2 - 12d\mu^6 - 8d\mu^5 + 368d\mu^4 + 64d\mu^3 + 36\mu^6 + 176\mu^5\right) + \gamma^4 \\
& \cdot \left(\frac{864d^2\mu^3}{5} + \frac{324d\mu^5}{5} + \frac{1296d\mu^4}{5}\right) + \gamma^2 \cdot \left(\frac{189d^2\mu}{10} - \frac{459d\mu^3}{5} + \frac{297d\mu^2}{5} + \frac{324\mu^5}{5} - \frac{27\mu^4}{5} + \frac{189\mu^3}{2}\right) \\
& + \gamma \left(\frac{9539d\mu^2}{100} - \mu^5 - \frac{29373\mu^4}{200} + \frac{3307\mu^3}{100} + \frac{7\mu^2}{2}\right) + \frac{1701\mu^3}{25}.
\end{aligned}
\tag{69}
$$

Using $d \geq 1$ and $\mu \leq 0.1$, we get

$$
\begin{aligned}
p_5^{(2)}(x)/\gamma^3 \geq{} & \gamma^5 \cdot \left(48d^2\mu^2 + \frac{51297d\mu^3}{250} + 32d\mu^2 + 36\mu^6 + 176\mu^5 + \frac{1836\mu^4}{5} + 64\mu^3\right) \\
& + \gamma^4 \cdot \left(\frac{864d\mu^3}{5} + \frac{324\mu^5}{5} + \frac{1296\mu^4}{5}\right) + \gamma^2 \\
& \cdot \left(\frac{189d\mu}{10} + \frac{324\mu^5}{5} + \frac{2349\mu^3}{25} + \frac{2511\mu^2}{50}\right) + \gamma \left(\frac{36747\mu^3}{2000} + \frac{9889\mu^2}{100}\right) + \frac{1701\mu^3}{25}.
\end{aligned}
\tag{70}
$$

Thus, we have $p_5^{(2)}(x) > 0$. We continue, and consider

$$
\begin{aligned}
p_5^{(1)}(x)/\gamma^4 = {}& \gamma^8 \cdot \left(16d^4\mu^4 + 8d^3\mu^6 + 64d^3\mu^5 + 16d^2\mu^7 + 80d^2\mu^6 + 8d\mu^8 + 32d\mu^7\right) \\
& + \gamma^7 \left(-\frac{576d^4\mu^3}{5} - \frac{288d^3\mu^5}{5} - \frac{2304d^3\mu^4}{5} - \frac{576d^2\mu^6}{5} - 576d^2\mu^5 - \frac{288d\mu^7}{5} - \frac{1152d\mu^6}{5}\right) \\
& + \gamma^6 \cdot \left(\frac{2988d^4\mu^2}{25} + \frac{1444d^3\mu^4}{25} + \frac{11952d^3\mu^3}{25} + 8d^2\mu^6 + \frac{2088d^2\mu^5}{25}\right. \\
& \left. + \frac{2828d^2\mu^4}{5} + 2d\mu^8 - 8d\mu^7 - \frac{1356d\mu^6}{25} + \frac{4376d\mu^5}{25} - 12\mu^8 - 48\mu^7\right) + \gamma^5 \\
& \cdot \left(\frac{504d^3\mu^3}{5} - \frac{216d^2\mu^5}{5} + 432d^2\mu^4 + \frac{288d^2\mu^3}{5} - \frac{108d\mu^7}{5} + \frac{3672d\mu^5}{5} + \frac{576d\mu^4}{5} + 72\mu^7 + \frac{1728\mu^6}{5}\right) \\
& + \gamma^4 \left(-\frac{6723d^3\mu^2}{50} + \frac{3338d^2\mu^4}{25} - \frac{12699d^2\mu^3}{25} + \frac{1133d\mu^6}{25}\right. \\
& \left. + \frac{5832d\mu^5}{25} - \frac{34003d\mu^4}{50} + \mu^8 + 8\mu^7 + \frac{778\mu^6}{25} - \frac{7364\mu^5}{25}\right) \\
& + \gamma^3 \left(-\frac{12978d^2\mu^3}{125} + \frac{243d\mu^5}{125} - \frac{35856d\mu^4}{125} - \frac{468d\mu^3}{5} - \frac{72\mu^7}{5} - \frac{216\mu^6}{5} - \frac{35514\mu^5}{125} - \frac{576\mu^4}{5}\right) \\
& + \gamma^2 \cdot \left(\frac{747d^2\mu^2}{25} - \frac{14499d\mu^4}{200} + \frac{5229d\mu^3}{50} + \frac{2691\mu^6}{50} - 18\mu^5 + \frac{18369\mu^4}{100}\right) \\
& + \gamma \left(\frac{5589d\mu^3}{125} - \frac{7614\mu^5}{125} + \frac{7614\mu^4}{125} + \frac{162\mu^3}{5}\right) + \frac{27621\mu^4}{2000}.
\end{aligned}
\tag{71}
$$

We ignore all positive terms w.r.t. $\gamma^8$, use $\gamma \le \frac{1}{2\sqrt{2}\sqrt{d}}$ and $d \ge \mu$ to obtain

$$
\begin{aligned}
p_5^{(1)}(x)/\gamma^4 \ge {}& \frac{243d\gamma^3\mu^5}{125} + \gamma^6 \cdot \left(\frac{2988d^4\mu^2}{25} + \frac{1444d^3\mu^4}{25} + \frac{11952d^3\mu^3}{25} + 8d^2\mu^6\right. \\
& \left. + \frac{2088d^2\mu^5}{25} + \frac{2828d^2\mu^4}{5} + 2d\mu^8 - 8d\mu^7 - \frac{1356d\mu^6}{25} + \frac{4376d\mu^5}{25} - 12\mu^8 - 48\mu^7\right) + \gamma^5 \\
& \cdot \left(\frac{432d^3\mu^3}{5} - \frac{252d^2\mu^5}{5} + \frac{1872d^2\mu^4}{5} + \frac{288d^2\mu^3}{5} - \frac{108d\mu^7}{5} - \frac{72d\mu^6}{5} + \frac{3312d\mu^5}{5} + \frac{576d\mu^4}{5}\right. \\
& \left. + \frac{324\mu^7}{5} + \frac{1584\mu^6}{5}\right) + \gamma^4 \cdot \left(\frac{3338d^2\mu^4}{25} + \frac{1133d\mu^6}{25} + \frac{5832d\mu^5}{25} + \mu^8 + 8\mu^7 + \frac{778\mu^6}{25}\right) \\
& + \gamma^2 \cdot \left(\frac{5229d^2\mu^2}{400} - \frac{14499d\mu^4}{200} + \frac{8217d\mu^3}{200} + \frac{2691\mu^6}{50} - 18\mu^5 + \frac{4949\mu^4}{80}\right) \\
& + \gamma \left(\frac{15867d\mu^3}{500} - \frac{9\mu^6}{5} - \frac{8289\mu^5}{125} - \frac{5229\mu^4}{500} + \frac{63\mu^3}{10}\right) + \frac{27621\mu^4}{2000}.
\end{aligned}
\tag{72}
$$

Using $d \ge 1$ and $\mu \le 0.1$, we get

$$
\begin{aligned}
p_5^{(1)}(x)/\gamma^4 \ge {}& \gamma^6 \cdot \left(\frac{2988d^3\mu^2}{25} + \frac{1444d^2\mu^4}{25} + \frac{11952d^2\mu^3}{25} + \frac{9852d\mu^5}{125} + \frac{2828d\mu^4}{5} + \frac{3491\mu^5}{20}\right) + \gamma^5 \\
& \cdot \left(\frac{10737d^2\mu^3}{125} + \frac{467793d\mu^4}{1250} + \frac{288d\mu^3}{5} + \frac{324\mu^7}{5} + \frac{1584\mu^6}{5} + \frac{3312\mu^5}{5} + \frac{576\mu^4}{5}\right) \\
& + \gamma^4 \cdot \left(\frac{3338d\mu^4}{25} + \mu^8 + 8\mu^7 + \frac{1911\mu^6}{25} + \frac{5832\mu^5}{25}\right) + \frac{243\gamma^3\mu^5}{125} + \gamma^2 \\
& \cdot \left(\frac{246951d\mu^2}{20000} + \frac{2691\mu^6}{50} + \frac{961\mu^4}{16} + \frac{8217\mu^3}{200}\right) + \frac{454041\gamma\mu^3}{12500} + \frac{27621\mu^4}{2000}.
\end{aligned}
\tag{73}
$$

Thus, we have $p_5^{(1)}(x) > 0$. Next, we consider

$$
\begin{aligned}
p_5(x)/\gamma^5 ={}& \gamma^8 \cdot \left( \frac{144d^4\mu^5}{5} + \frac{72d^3\mu^7}{5} + \frac{576d^3\mu^6}{5} + \frac{144d^2\mu^8}{5} + 144d^2\mu^7 + \frac{72d\mu^9}{5} + \frac{288d\mu^8}{5} \right) \\
&+ \gamma^7 \left( -\frac{2592d^4\mu^4}{25} - \frac{1496d^3\mu^6}{25} - \frac{10368d^3\mu^5}{25} - 4d^2\mu^8 - \frac{2992d^2\mu^7}{25} - \frac{2512d^2\mu^6}{5} - \frac{1096d\mu^8}{25} \right. \\
&\left. - \frac{4384d\mu^7}{25} \right) + \gamma^6 \cdot \left( \frac{3564d^4\mu^3}{125} + \frac{1332d^3\mu^5}{125} + \frac{14256d^3\mu^4}{125} + \frac{72d^2\mu^7}{5} - \frac{4536d^2\mu^6}{125} \right. \\
&\left. + \frac{2124d^2\mu^5}{25} + \frac{18d\mu^9}{5} - \frac{72d\mu^8}{5} - \frac{23868d\mu^7}{125} - \frac{7272d\mu^6}{125} - \frac{108\mu^9}{5} - \frac{432\mu^8}{5} \right) + \gamma^5 \\
&\cdot \left( \frac{2718d^3\mu^4}{25} - \frac{422d^2\mu^6}{25} + \frac{2124d^2\mu^5}{5} + \frac{396d^2\mu^4}{25} - \frac{386d\mu^8}{25} + 16d\mu^7 + \frac{15424d\mu^6}{25} \right. \\
&\left. + \frac{792d\mu^5}{25} + \frac{284\mu^8}{5} + \frac{6976\mu^7}{25} \right) + \gamma^4 \left( -\frac{8019d^3\mu^3}{250} + \frac{6714d^2\mu^5}{125} - \frac{15147d^2\mu^4}{125} \right. \\
&\left. + \frac{1449d\mu^7}{125} + \frac{17496d\mu^6}{125} - \frac{20259d\mu^5}{250} + \frac{9\mu^9}{5} + \frac{72\mu^8}{5} + \frac{12834\mu^7}{125} + \frac{3708\mu^6}{125} \right) \\
&+ \gamma^3 \left( -\frac{96777d^2\mu^4}{2500} + \frac{5778d\mu^6}{625} - \frac{78876d\mu^5}{625} - \frac{1287d\mu^4}{50} - \frac{324\mu^8}{25} - \frac{972\mu^7}{25} - \frac{116694\mu^6}{625} \right. \\
&\left. - \frac{792\mu^5}{25} \right) + \gamma^2 \cdot \left( \frac{891d^2\mu^3}{125} - \frac{31347d\mu^5}{1000} + \frac{6237d\mu^4}{250} + \frac{6723\mu^7}{250} - \frac{162\mu^6}{5} + \frac{7857\mu^5}{500} \right) \\
&+ \gamma \left( \frac{41877d\mu^4}{5000} - \frac{8019\mu^6}{625} + \frac{8019\mu^5}{625} + \frac{891\mu^4}{100} \right) + \frac{88209\mu^5}{50000}.
\end{aligned} \tag{74}
$$

We ignore all positive terms w.r.t. $\gamma^8$ and use $\gamma \leq \frac{1}{4\sqrt{d}}$ and $d \geq \mu$ to obtain

$$
\begin{aligned}
p_5(x)/\gamma^5 \geq{}& \frac{5778d\gamma^3\mu^6}{625} + \gamma^6 \cdot \left( \frac{3564d^4\mu^3}{125} + \frac{1332d^3\mu^5}{125} + \frac{14256d^3\mu^4}{125} + \frac{72d^2\mu^7}{5} \right. \\
&\left. - \frac{4536d^2\mu^6}{125} + \frac{2124d^2\mu^5}{25} + \frac{18d\mu^9}{5} - \frac{72d\mu^8}{5} - \frac{23868d\mu^7}{125} - \frac{7272d\mu^6}{125} - \frac{108\mu^9}{5} - \frac{432\mu^8}{5} \right) \\
&+ \gamma^5 \cdot \left( \frac{2556d^3\mu^4}{25} - \frac{1031d^2\mu^6}{50} + \frac{9972d^2\mu^5}{25} + \frac{396d^2\mu^4}{25} - \frac{1569d\mu^8}{100} \right. \\
&\left. + \frac{213d\mu^7}{25} + \frac{14639d\mu^6}{25} + \frac{792d\mu^5}{25} + \frac{2703\mu^8}{50} + \frac{6702\mu^7}{25} \right) + \gamma^4 \\
&\cdot \left( \frac{6714d^2\mu^5}{125} + \frac{1449d\mu^7}{125} + \frac{17496d\mu^6}{125} + \frac{9\mu^9}{5} + \frac{72\mu^8}{5} + \frac{12834\mu^7}{125} + \frac{3708\mu^6}{125} \right) \\
&+ \gamma^2 \cdot \left( \frac{20493d^2\mu^3}{4000} - \frac{31347d\mu^5}{1000} + \frac{34749d\mu^4}{2000} + \frac{6723\mu^7}{250} - \frac{162\mu^6}{5} + \frac{42597\mu^5}{4000} \right) \\
&+ \gamma \left( \frac{238239d\mu^4}{40000} - \frac{81\mu^7}{100} - \frac{38151\mu^6}{2500} - \frac{33633\mu^5}{5000} + \frac{4257\mu^4}{800} \right) + \frac{88209\mu^5}{50000}.
\end{aligned} \tag{75}
$$

Using $d \geq 1$ and $\mu \leq 0.1$, we get

$$
\begin{aligned}
p_5(x)/\gamma^5 \geq {}& \gamma^6 \cdot \left( \frac{2107539\mu^5}{25000} + \frac{14256\mu^4}{125} + \frac{3564\mu^3}{125} \right) + \gamma^5 \\
& \cdot \left( \frac{2703\mu^8}{50} + \frac{1383\mu^7}{5} + \frac{14639\mu^6}{25} + \frac{10764\mu^5}{25} + \frac{117872231\mu^4}{1000000} \right) + \gamma^4 \\
& \cdot \left( \frac{9\mu^9}{5} + \frac{72\mu^8}{5} + \frac{14283\mu^7}{125} + \frac{21204\mu^6}{125} + \frac{6714\mu^5}{125} \right) + \frac{5778\gamma^3\mu^6}{625} + \gamma^2 \\
& \cdot \left( \frac{6723\mu^7}{250} + \frac{29637\mu^5}{4000} + \frac{34749\mu^4}{2000} + \frac{240489\mu^3}{50000} \right) + \frac{10451151\gamma\mu^4}{1000000} + \frac{88209\mu^5}{50000}.
\end{aligned}
\tag{76}
$$

Thus, we have $p_5(x) > 0$. It means that $x = \frac{9\gamma\mu}{5} + 1$ is an upper bound to all solutions of (56). Thus, we get $\max_{x \in \mathbb{R}^5, \|x\|=1} \left\{ x^\top (\mathbf{I} + \mathbf{C}) x \right\} \leq \frac{9\gamma\mu}{5} + 1$. It is left to substitute this bound to (55) and (54). $\qquad \square$

**Lemma C.6.** *The matrices* $\mathbf{A}_1$ *and* $\mathbf{A}_2$ *satisfy*

$$
\max_{x \in Q} \|(\mathbf{I} + \gamma\mathbf{A}_1)(\mathbf{I} + \gamma\mathbf{A}_2)x\| \leq \sqrt{\frac{9\gamma\mu}{5} + 1 + \frac{\delta\gamma}{2}},
$$

*for all* $\gamma \leq \frac{1}{4\sqrt{d}}$, $\mu \leq 0.1$, *and* $\delta \geq 0$, *where*

$$
Q := \{ x \in \mathbb{R}^{d+2} \,|\, \|x\| = 1, x^\top \mathbf{A}_2 x \leq 0, x^\top (\mathbf{I} + \gamma\mathbf{A}_2)\mathbf{A}_1(\mathbf{I} + \gamma\mathbf{A}_2)x \leq \delta \}.
$$

*Proof.* The proof is similar to Lemma C.6 with only change that we have to swap $\mathbf{A}_1$ and $\mathbf{A}_2$.

Note that

$$
(\mathbf{I} + \gamma\mathbf{A}_2)(\mathbf{I} + \gamma\mathbf{A}_1) = \mathbf{I} + \begin{bmatrix} \gamma^2 a_1^\top a_2 & \gamma^2 a_1^\top \mathbf{P} a_2 & \gamma(a_2^\top + a_1^\top) \\ \gamma^2 a_1^\top \mathbf{P}^\top a_2 & \gamma^2 a_1^\top a_2 & \gamma(a_2^\top \mathbf{P}^\top + a_1^\top \mathbf{P}^\top) \\ \gamma(a_1 + a_2) & \gamma(\mathbf{P}a_1 + \mathbf{P}a_2) & \gamma^2(a_1 a_2^\top + \mathbf{P}a_1 a_2^\top \mathbf{P}^\top) \end{bmatrix}
$$

since $\mathbf{P}^\top \mathbf{P} = \mathbf{I}$. Moreover,

$$
\begin{aligned}
& (\mathbf{I} + \gamma\mathbf{A}_1)\mathbf{A}_2(\mathbf{I} + \gamma\mathbf{A}_1) \\
={}& \begin{bmatrix} 2\gamma a_1^\top a_2 & \gamma a_1^\top \mathbf{P}a_2 + \gamma a_2^\top \mathbf{P}a_1 & a_1^\top + \gamma^2 a_2^\top \left(a_1 a_2^\top + \mathbf{P}a_1 a_2^\top \mathbf{P}^\top\right) \\ \gamma a_1^\top \mathbf{P}^\top a_2 + \gamma a_2^\top \mathbf{P}^\top a_1 & 2\gamma a_1^\top a_2 & a_1^\top \mathbf{P}^\top + \gamma^2 a_2^\top \mathbf{P}^\top \left(a_1 a_2^\top + \mathbf{P}a_1 a_2^\top \mathbf{P}^\top\right) \\ a_1 + \gamma^2 \left(a_2 a_1^\top + \mathbf{P}a_2 a_1^\top \mathbf{P}^\top\right) a_2 & \mathbf{P}a_1 + \gamma^2 \left(a_2 a_1^\top + \mathbf{P}a_2 a_1^\top \mathbf{P}^\top\right)\mathbf{P}a_2 & \gamma(a_2 a_1^\top + a_1 a_2^\top + \mathbf{P}a_2 a_1^\top \mathbf{P}^\top + \mathbf{P}a_1 a_2^\top \mathbf{P}^\top). \end{bmatrix}
\end{aligned}
$$

Similarly to the proof of Lemma C.5,

$$
\begin{aligned}
& \max_{x \in Q} x^\top (\mathbf{I} + \gamma\mathbf{A}_2)(\mathbf{I} + \gamma\mathbf{A}_1)^2(\mathbf{I} + \gamma\mathbf{A}_2) x \\
& \leq \max_{x \in \mathbb{R}^{d+2}, \|x\|=1} \left\{ x^\top \left[ (\mathbf{I} + \gamma\mathbf{A}_2)(\mathbf{I} + \gamma\mathbf{A}_1)^2(\mathbf{I} + \gamma\mathbf{A}_2) - \frac{\gamma}{2}\mathbf{A}_2 - \frac{\gamma}{2}(\mathbf{I} + \gamma\mathbf{A}_2)\mathbf{A}_1(\mathbf{I} + \gamma\mathbf{A}_2) \right] x \right\} + \frac{\delta\gamma}{2} \\
& = 1 + \max_{x \in \mathbb{R}^{d+2}, \|x\|=1} \left\{ x^\top \mathbf{B} x \right\} + \frac{\delta\gamma}{2},
\end{aligned}
\tag{77}
$$

where

$$
\mathbf{B} := (\mathbf{I} + \gamma\mathbf{A}_2)(\mathbf{I} + \gamma\mathbf{A}_1)^2(\mathbf{I} + \gamma\mathbf{A}_2) - \frac{\gamma}{2}\mathbf{A}_2 - \frac{\gamma}{2}(\mathbf{I} + \gamma\mathbf{A}_2)\mathbf{A}_1(\mathbf{I} + \gamma\mathbf{A}_2) - \mathbf{I},
$$

and we apply Lemma D.3 to $\mathbf{B}$ to get an equivalent problem

$$
\max_{x \in \mathbb{R}^5, \|x\|=1} \left\{ x^\top (\mathbf{I} + \mathbf{C})x \right\} + \frac{\delta\gamma}{2}
$$

with $\mathbf{C}$ defined in Lemma D.3.

We consider the characteristic polynomial of $(\mathbf{I} + \mathbf{C})$ :

$$p_5(\lambda) := \det\left(\lambda\mathbf{I} - (\mathbf{I} + \mathbf{C})\right).$$

If we want to show that all solutions are bounded by $x$, it is sufficient to show that all derivatives $p_5^{(i)}(x)$ are positive at this point. Let us consider $x = \frac{9\gamma\mu}{5} + 1$. Clearly, we have $p_5^{(5)}(x) = 120 > 0$ for all $x \in \mathbb{R}$. We use the symbolic computation library Sympy (Meurer et al., 2017) to obtain

$$p_5^{(4)}(x)/\gamma = \gamma^3\left(-192d^2 - 48d\mu^2 - 384d\mu - 96\mu^2\right) + \gamma\left(96d - 96\mu^2 + 96\mu\right) + 216\mu. \tag{78}$$

Using $\gamma \leq \frac{1}{2\sqrt{d}}$, we have

$$p_5^{(4)}(x)/\gamma \geq \gamma\left(48d - 108\mu^2 - \frac{24\mu^2}{d}\right) + 216\mu. \tag{79}$$

Using $d \geq \mu$, we get

$$p_5^{(4)}(x)/\gamma \geq \gamma\left(48d - 108\mu^2 - 24\mu\right) + 216\mu. \tag{80}$$

Using $d \geq 1$ and $\mu \leq 0.1$, we get

$$p_5^{(4)}(x)/\gamma \geq \frac{1113\gamma}{25} + 216\mu. \tag{81}$$

Thus, we have $p_5^{(4)}(x) > 0$. Next, we get

$$
\begin{aligned}
p_5^{(3)}(x)/\gamma^2 = {} & \gamma^6 \cdot \left(96d^4 + 48d^3\mu^2 + 384d^3\mu + 96d^2\mu^3 + 480d^2\mu^2 + 48d\mu^4 + 192d\mu^3\right) \\
& + \gamma^4\left(-108d^3 + 138d^2\mu^2 - 240d^2\mu + 36d\mu^4 + 228d\mu^3 - 252d\mu^2 + 18\mu^4 - 120\mu^3\right) \\
& + \gamma^3\left(-\frac{1728d^2\mu}{5} - \frac{432d\mu^3}{5} - \frac{3456d\mu^2}{5} - \frac{864\mu^3}{5}\right) + \gamma^2 \\
& \cdot \left(24d^2 - 69d\mu^2 + 12d\mu + 36\mu^4 - 18\mu^3 + 90\mu^2\right) + \gamma\left(\frac{864d\mu}{5} - \frac{864\mu^3}{5} + \frac{864\mu^2}{5}\right) + \frac{837\mu^2}{5}.
\end{aligned}
\tag{82}
$$

We ignore all positive terms w.r.t. $\gamma^4$ and $\gamma^6$ and obtain

$$
\begin{aligned}
p_5^{(3)}(x)/\gamma^2 \geq {} & \gamma^4\left(-108d^3 - 240d^2\mu - 252d\mu^2 - 120\mu^3\right) + \gamma^3\left(-\frac{1728d^2\mu}{5} - \frac{432d\mu^3}{5} - \frac{3456d\mu^2}{5} - \frac{864\mu^3}{5}\right) \\
& + \gamma^2 \cdot \left(24d^2 - 69d\mu^2 + 12d\mu + 36\mu^4 - 18\mu^3 + 90\mu^2\right) + \gamma\left(\frac{864d\mu}{5} - \frac{864\mu^3}{5} + \frac{864\mu^2}{5}\right) + \frac{837\mu^2}{5}.
\end{aligned}
\tag{83}
$$

Using $\gamma \leq \frac{1}{2\sqrt{2}\sqrt{d}}$, we have

$$
\begin{aligned}
p_5^{(3)}(x)/\gamma^2 \geq {} & \gamma^2 \cdot \left(\frac{21d^2}{2} - 69d\mu^2 - 18d\mu + 36\mu^4 - 18\mu^3 + \frac{117\mu^2}{2} - \frac{15\mu^3}{d}\right) \\
& + \gamma\left(\frac{648d\mu}{5} - \frac{918\mu^3}{5} + \frac{432\mu^2}{5} - \frac{108\mu^3}{5d}\right) + \frac{837\mu^2}{5}.
\end{aligned}
\tag{84}
$$

Using $d \geq \mu$, we get

$$p_5^{(3)}(x)/\gamma^2 \geq \gamma^2 \cdot \left( \frac{21d^2}{2} - 69d\mu^2 - 18d\mu + 36\mu^4 - 18\mu^3 + \frac{87\mu^2}{2} \right) + \gamma \left( \frac{648d\mu}{5} - \frac{918\mu^3}{5} + \frac{324\mu^2}{5} \right) + \frac{837\mu^2}{5}. \quad (85)$$

Using $d \geq 1$ and $\mu \leq 0.1$, we get

$$p_5^{(3)}(x)/\gamma^2 \geq \gamma^2 \cdot \left( \frac{801d}{100} + 36\mu^4 + \frac{417\mu^2}{10} \right) + \gamma \left( \frac{1161\mu^2}{25} + \frac{648\mu}{5} \right) + \frac{837\mu^2}{5}. \quad (86)$$

Thus, $p_5^{(3)}(x) > 0$. Next, we consider

$$
\begin{aligned}
p_5^{(2)}(x)/\gamma^3 &= \gamma^7 \left( -64d^4\mu^2 - 32d^3\mu^4 - 256d^3\mu^3 - 64d^2\mu^5 - 320d^2\mu^4 - 32d\mu^6 - 128d\mu^5 \right) \\
&+ \gamma^6 \cdot \left( \frac{864d^4\mu}{5} + \frac{432d^3\mu^3}{5} + \frac{3456d^3\mu^2}{5} + \frac{864d^2\mu^4}{5} + 864d^2\mu^3 + \frac{432d\mu^5}{5} + \frac{1728d\mu^4}{5} \right) + \gamma^5 \\
&\cdot \left( 56d^3\mu^2 - 52d^2\mu^4 + 128d^2\mu^3 + 32d^2\mu^2 - 12d\mu^6 - 56d\mu^5 + 184d\mu^4 + 64d\mu^3 + 12\mu^6 + 80\mu^5 \right) \\
&+ \gamma^4 \left( -\frac{972d^3\mu}{5} + \frac{1242d^2\mu^3}{5} - 432d^2\mu^2 + \frac{324d\mu^5}{5} + \frac{2052d\mu^4}{5} - \frac{2268d\mu^3}{5} + \frac{162\mu^5}{5} - 216\mu^4 \right) \\
&+ \gamma^3 \left( -\frac{6626d^2\mu^2}{25} - \frac{1044d\mu^4}{25} - \frac{13452d\mu^3}{25} - 52d\mu^2 - 8\mu^6 - 12\mu^5 - \frac{5188\mu^4}{25} - 40\mu^3 \right) \\
&+ \gamma^2 \cdot \left( \frac{216d^2\mu}{5} - \frac{621d\mu^3}{5} + \frac{108d\mu^2}{5} + \frac{324\mu^5}{5} - \frac{162\mu^4}{5} + 162\mu^3 \right) \\
&+ \gamma \left( \frac{3213d\mu^2}{25} - \frac{3438\mu^4}{25} + \frac{3438\mu^3}{25} + 18\mu^2 \right) + \frac{1701\mu^3}{25}.
\end{aligned}
\quad (87)
$$

We ignore all positive terms w.r.t. $\gamma^6$ and obtain

$$
\begin{aligned}
p_5^{(2)}(x)/\gamma^3 &\geq \gamma^7 \left( -64d^4\mu^2 - 32d^3\mu^4 - 256d^3\mu^3 - 64d^2\mu^5 - 320d^2\mu^4 - 32d\mu^6 - 128d\mu^5 \right) + \gamma^5 \\
&\cdot \left( 56d^3\mu^2 - 52d^2\mu^4 + 128d^2\mu^3 + 32d^2\mu^2 - 12d\mu^6 - 56d\mu^5 + 184d\mu^4 + 64d\mu^3 + 12\mu^6 + 80\mu^5 \right) \\
&+ \gamma^4 \left( -\frac{972d^3\mu}{5} + \frac{1242d^2\mu^3}{5} - 432d^2\mu^2 + \frac{324d\mu^5}{5} + \frac{2052d\mu^4}{5} - \frac{2268d\mu^3}{5} + \frac{162\mu^5}{5} - 216\mu^4 \right) \\
&+ \gamma^3 \left( -\frac{6626d^2\mu^2}{25} - \frac{1044d\mu^4}{25} - \frac{13452d\mu^3}{25} - 52d\mu^2 - 8\mu^6 - 12\mu^5 - \frac{5188\mu^4}{25} - 40\mu^3 \right) \\
&+ \gamma^2 \cdot \left( \frac{216d^2\mu}{5} - \frac{621d\mu^3}{5} + \frac{108d\mu^2}{5} + \frac{324\mu^5}{5} - \frac{162\mu^4}{5} + 162\mu^3 \right) \\
&+ \gamma \left( \frac{3213d\mu^2}{25} - \frac{3438\mu^4}{25} + \frac{3438\mu^3}{25} + 18\mu^2 \right) + \frac{1701\mu^3}{25}.
\end{aligned}
\quad (88)
$$

Using $\gamma \leq \frac{1}{2\sqrt{2}\sqrt{d}}$, we have

$$
\begin{aligned}
p_5^{(2)}(x)/\gamma^3 &\geq \gamma^5 \cdot \left( 48d^3\mu^2 - 56d^2\mu^4 + 96d^2\mu^3 + 32d^2\mu^2 - 12d\mu^6 - 64d\mu^5 + 144d\mu^4 + 64d\mu^3 + 8\mu^6 + 64\mu^5 \right) \\
&+ \gamma^4 \cdot \left( \frac{1242d^2\mu^3}{5} + \frac{324d\mu^5}{5} + \frac{2052d\mu^4}{5} + \frac{162\mu^5}{5} \right) + \gamma^2 \\
&\cdot \left( \frac{189d^2\mu}{10} - \frac{621d\mu^3}{5} - \frac{162d\mu^2}{5} + \frac{324\mu^5}{5} - \frac{162\mu^4}{5} + \frac{1053\mu^3}{10} - \frac{27\mu^4}{d} \right) \\
&+ \gamma \left( \frac{9539d\mu^2}{100} - \frac{7137\mu^4}{50} + \frac{3513\mu^3}{50} + \frac{23\mu^2}{2} - \frac{\mu^6}{d} - \frac{3\mu^5}{2d} - \frac{1297\mu^4}{50d} - \frac{5\mu^3}{d} \right) + \frac{1701\mu^3}{25}.
\end{aligned}
\quad (89)
$$

Using $d \geq \mu$, we get

$$p_5^{(2)}(x)/\gamma^3 \geq \gamma^5 \cdot \left(48d^3\mu^2 - 56d^2\mu^4 + 96d^2\mu^3 + 32d^2\mu^2 - 12d\mu^6 - 64d\mu^5 + 144d\mu^4 + 64d\mu^3 + 8\mu^6 + 64\mu^5\right)$$
$$+ \gamma^4 \cdot \left(\frac{1242d^2\mu^3}{5} + \frac{324d\mu^5}{5} + \frac{2052d\mu^4}{5} + \frac{162\mu^5}{5}\right) + \gamma^2 \tag{90}$$
$$\cdot \left(\frac{189d^2\mu}{10} - \frac{621d\mu^3}{5} - \frac{162d\mu^2}{5} + \frac{324\mu^5}{5} - \frac{162\mu^4}{5} + \frac{783\mu^3}{10}\right)$$
$$+ \gamma\left(\frac{9539d\mu^2}{100} - \mu^5 - \frac{3606\mu^4}{25} + \frac{1108\mu^3}{25} + \frac{13\mu^2}{2}\right) + \frac{1701\mu^3}{25}.$$

Using $d \geq 1$ and $\mu \leq 0.1$, we get

$$p_5^{(2)}(x)/\gamma^3 \geq \gamma^5 \cdot \left(48d^2\mu^2 + \frac{22597d\mu^3}{250} + 32d\mu^2 + 8\mu^6 + 64\mu^5 + \frac{688\mu^4}{5} + 64\mu^3\right)$$
$$+ \gamma^4 \cdot \left(\frac{1242d\mu^3}{5} + \frac{486\mu^5}{5} + \frac{2052\mu^4}{5}\right) + \gamma^2 \tag{91}$$
$$\cdot \left(\frac{7209d\mu}{500} + \frac{324\mu^5}{5} + \frac{3753\mu^3}{50}\right) + \gamma\left(\frac{14943\mu^3}{500} + \frac{10189\mu^2}{100}\right) + \frac{1701\mu^3}{25}.$$

Thus, we have $p_5^{(2)}(x) > 0$. We continue, and consider

$$p_5^{(1)}(x)/\gamma^4 = \gamma^8 \cdot \left(16d^4\mu^4 + 8d^3\mu^6 + 64d^3\mu^5 + 16d^2\mu^7 + 80d^2\mu^6 + 8d\mu^8 + 32d\mu^7\right)$$
$$+ \gamma^7 \left(-\frac{576d^4\mu^3}{5} - \frac{288d^3\mu^5}{5} - \frac{2304d^3\mu^4}{5} - \frac{576d^2\mu^6}{5} - 576d^2\mu^5 - \frac{288d\mu^7}{5} - \frac{1152d\mu^6}{5}\right) + \gamma^6$$
$$\cdot \left(\frac{2988d^4\mu^2}{25} + \frac{1444d^3\mu^4}{25} + \frac{11952d^3\mu^3}{25} + 15d^2\mu^6 + \frac{2788d^2\mu^5}{25} + \frac{2828d^2\mu^4}{5} + 2d\mu^8 + 6d\mu^7\right.$$
$$\left. + \frac{44d\mu^6}{25} + \frac{4376d\mu^5}{25} - 5\mu^8 - 20\mu^7\right) + \gamma^5 \cdot \left(\frac{504d^3\mu^3}{5} - \frac{468d^2\mu^5}{5} + \frac{1152d^2\mu^4}{5} + \frac{288d^2\mu^3}{5}\right.$$
$$\left. - \frac{108d\mu^7}{5} - \frac{504d\mu^6}{5} + \frac{1656d\mu^5}{5} + \frac{576d\mu^4}{5} + \frac{108\mu^7}{5} + 144\mu^6\right) + \gamma^4\left(-\frac{6723d^3\mu^2}{50} + \frac{18581d^2\mu^4}{100}\right. \tag{92}$$
$$\left. - \frac{1494d^2\mu^3}{5} + \frac{1133d\mu^6}{25} + \frac{16893d\mu^5}{50} - \frac{13087d\mu^4}{50} + \mu^8 + 5\mu^7 + \frac{6541\mu^6}{100} - \frac{547\mu^5}{5}\right)$$
$$+ \gamma^3 \left(-\frac{12978d^2\mu^3}{125} + \frac{2268d\mu^5}{125} - \frac{27756d\mu^4}{125} - \frac{468d\mu^3}{5} - \frac{72\mu^7}{5} - \frac{108\mu^6}{5} - \frac{23364\mu^5}{125} - 72\mu^4\right)$$
$$+ \gamma^2 \cdot \left(\frac{747d^2\mu^2}{25} - \frac{18981d\mu^4}{200} + \frac{747d\mu^3}{50} + \frac{2691\mu^6}{50} - \frac{4041\mu^5}{100} + \frac{1881\mu^4}{20}\right)$$
$$+ \gamma\left(\frac{5589d\mu^3}{125} - \frac{7614\mu^5}{125} + \frac{7614\mu^4}{125} + \frac{162\mu^3}{5}\right) + \frac{27621\mu^4}{2000}.$$

We ignore all positive terms w.r.t. $\gamma^8$, use $\gamma \leq \frac{1}{2\sqrt{2}\sqrt{d}}$ and $d \geq \mu$ to obtain

$$
p_5^{(1)}(x)/\gamma^4 \geq \frac{2268d\gamma^3\mu^5}{125} + \gamma^6 \cdot \left( \frac{2988d^4\mu^2}{25} + \frac{1444d^3\mu^4}{25} + \frac{11952d^3\mu^3}{25} + 15d^2\mu^6 \right.
$$

$$
+ \frac{2788d^2\mu^5}{25} + \frac{2828d^2\mu^4}{5} + 2d\mu^8 + 6d\mu^7 + \frac{44d\mu^6}{25} + \frac{4376d\mu^5}{25} - 5\mu^8 - 20\mu^7 \left. \right) + \gamma^5
$$

$$
\cdot \left( \frac{432d^3\mu^3}{5} - \frac{504d^2\mu^5}{5} + \frac{864d^2\mu^4}{5} + \frac{288d^2\mu^3}{5} - \frac{108d\mu^7}{5} - \frac{576d\mu^6}{5} + \frac{1296d\mu^5}{5} + \frac{576d\mu^4}{5} \right.
$$

$$
+ \frac{72\mu^7}{5} + \frac{576\mu^6}{5} \left. \right) + \gamma^4 \cdot \left( \frac{18581d^2\mu^4}{100} + \frac{1133d\mu^6}{25} + \frac{16893d\mu^5}{50} + \mu^8 + 5\mu^7 + \frac{6541\mu^6}{100} \right)
$$

$$
+ \gamma^2 \cdot \left( \frac{5229d^2\mu^2}{400} - \frac{18981d\mu^4}{200} - \frac{2241d\mu^3}{100} + \frac{2691\mu^6}{50} - \frac{4041\mu^5}{100} + \frac{19063\mu^4}{400} \right)
$$

$$
+ \gamma \left( \frac{15867d\mu^3}{500} - \frac{9\mu^6}{5} - \frac{15903\mu^5}{250} + \frac{1224\mu^4}{125} + \frac{117\mu^3}{10} \right) + \frac{27621\mu^4}{2000}. \tag{93}
$$

Using $d \geq 1$ and $\mu \leq 0.1$, we get

$$
p_5^{(1)}(x)/\gamma^4 \geq \gamma^6
$$

$$
\cdot \left( \frac{2988d^3\mu^2}{25} + \frac{1444d^2\mu^4}{25} + \frac{11952d^2\mu^3}{25} + 15d\mu^6 + \frac{2788d\mu^5}{25} + \frac{2828d\mu^4}{5} + \frac{33\mu^6}{100} + \frac{4376\mu^5}{25} \right) \tag{94}
$$

$$
+ \gamma^5 \cdot \left( \frac{10674d^2\mu^3}{125} + \frac{214533d\mu^4}{1250} + \frac{288d\mu^3}{5} + \frac{72\mu^7}{5} + \frac{576\mu^6}{5} + \frac{1296\mu^5}{5} + \frac{576\mu^4}{5} \right)
$$

$$
+ \gamma^4 \cdot \left( \frac{18581d\mu^4}{100} + \mu^8 + 5\mu^7 + \frac{11073\mu^6}{100} + \frac{16893\mu^5}{50} \right) + \frac{2268\gamma^3\mu^5}{125} + \gamma^2
$$

$$
\cdot \left( \frac{197649d\mu^2}{20000} + \frac{2691\mu^6}{50} + \frac{87233\mu^4}{2000} \right) + \gamma \left( \frac{2133\mu^4}{625} + \frac{21717\mu^3}{500} \right) + \frac{27621\mu^4}{2000}.
$$

Thus, we have $p_5^{(1)}(x) > 0$. Next, we consider

$$
\begin{aligned}
p_5(x)/\gamma^5 = \gamma^8 \cdot & \left( \frac{144d^4\mu^5}{5} + \frac{72d^3\mu^7}{5} + \frac{576d^3\mu^6}{5} + \frac{144d^2\mu^8}{5} + 144d^2\mu^7 + \frac{72d\mu^9}{5} + \frac{288d\mu^8}{5} \right) \\
+ \gamma^7 & \left( -\frac{2592d^4\mu^4}{25} - \frac{1496d^3\mu^6}{25} - \frac{10368d^3\mu^5}{25} - 4d^2\mu^8 - \frac{2992d^2\mu^7}{25} - \frac{2512d^2\mu^6}{5} \right. \\
& \left. - \frac{1096d\mu^8}{25} - \frac{4384d\mu^7}{25} \right) + \gamma^6 \cdot \left( \frac{3564d^4\mu^3}{125} + \frac{1332d^3\mu^5}{125} + \frac{14256d^3\mu^4}{125} + 27d^2\mu^7 \right. \\
& + \frac{1764d^2\mu^6}{125} + \frac{2124d^2\mu^5}{25} + \frac{18d\mu^9}{5} + \frac{54d\mu^8}{5} - \frac{11268d\mu^7}{125} - \frac{7272d\mu^6}{125} - 9\mu^9 - 36\mu^8 \Big) \\
+ \gamma^5 \cdot & \left( \frac{2718d^3\mu^4}{25} - \frac{1556d^2\mu^6}{25} + \frac{6084d^2\mu^5}{25} + \frac{396d^2\mu^4}{25} - \frac{847d\mu^8}{50} - \frac{2018d\mu^7}{25} \right. \\
& + \frac{6352d\mu^6}{25} + \frac{792d\mu^5}{25} + \frac{361\mu^8}{25} + \frac{548\mu^7}{5} \Big) + \gamma^4 \left( -\frac{8019d^3\mu^3}{250} + \frac{33093d^2\mu^5}{500} \right. \\
& - \frac{1782d^2\mu^4}{25} + \frac{1449d\mu^7}{125} + \frac{41229d\mu^6}{250} + \frac{4689d\mu^5}{250} + \frac{9\mu^9}{5} + 9\mu^8 + \frac{41373\mu^7}{500} + \frac{909\mu^6}{25} \Big) \\
+ \gamma^3 & \left( -\frac{96777d^2\mu^4}{2500} + \frac{135999d\mu^6}{5000} - \frac{67977d\mu^5}{1250} - \frac{1287d\mu^4}{50} - \frac{324\mu^8}{25} - \frac{486\mu^7}{25} - \frac{264951\mu^6}{2500} \right. \\
& \left. - \frac{99\mu^5}{5} \right) + \gamma^2 \cdot \left( \frac{891d^2\mu^3}{125} - \frac{36693d\mu^5}{1000} + \frac{891d\mu^4}{250} + \frac{6723\mu^7}{250} - \frac{18873\mu^6}{500} - \frac{567\mu^5}{100} \right) \\
+ \gamma & \left( \frac{41877d\mu^4}{5000} - \frac{8019\mu^6}{625} + \frac{8019\mu^5}{625} + \frac{891\mu^4}{100} \right) + \frac{88209\mu^5}{50000}.
\end{aligned} \tag{95}
$$

We ignore all positive terms w.r.t. $\gamma^8$ and use $\gamma \leq \frac{1}{4\sqrt{d}}$ and $d \geq \mu$ to obtain

$$
\begin{aligned}
p_5(x)/\gamma^5 \geq & \frac{135999d\gamma^3\mu^6}{5000} + \gamma^6 \cdot \left( \frac{3564d^4\mu^3}{125} + \frac{1332d^3\mu^5}{125} + \frac{14256d^3\mu^4}{125} + 27d^2\mu^7 + \frac{1764d^2\mu^6}{125} \right. \\
& + \frac{2124d^2\mu^5}{25} + \frac{18d\mu^9}{5} + \frac{54d\mu^8}{5} - \frac{11268d\mu^7}{125} - \frac{7272d\mu^6}{125} - 9\mu^9 - 36\mu^8 \Big) + \gamma^5 \cdot \left( \frac{2556d^3\mu^4}{25} - \frac{3299d^2\mu^6}{50} \right. \\
& + \frac{5436d^2\mu^5}{25} + \frac{396d^2\mu^4}{25} - \frac{1719d\mu^8}{100} - \frac{441d\mu^7}{5} + \frac{5567d\mu^6}{25} + \frac{792d\mu^5}{25} + \frac{117\mu^8}{10} + \frac{2466\mu^7}{25} \Big) \\
+ \gamma^4 \cdot & \left( \frac{33093d^2\mu^5}{500} + \frac{1449d\mu^7}{125} + \frac{41229d\mu^6}{250} + \frac{4689d\mu^5}{250} + \frac{9\mu^9}{5} + 9\mu^8 + \frac{41373\mu^7}{500} + \frac{909\mu^6}{25} \right) \\
+ \gamma^2 \cdot & \left( \frac{20493d^2\mu^3}{4000} - \frac{36693d\mu^5}{1000} - \frac{891d\mu^4}{1000} + \frac{6723\mu^7}{250} - \frac{18873\mu^6}{500} - \frac{567\mu^5}{100} \right) \\
+ \gamma & \left( \frac{238239d\mu^4}{40000} - \frac{81\mu^7}{100} - \frac{70227\mu^6}{5000} + \frac{112311\mu^5}{40000} + \frac{4851\mu^4}{800} \right) + \frac{88209\mu^5}{50000}.
\end{aligned} \tag{96}
$$

Using $d \geq 1$ and $\mu \leq 0.1$, we get

$$
\begin{aligned}
p_5(x)/\gamma^5 \geq \gamma^6 \cdot & \left( \frac{4527621\mu^5}{50000} + \frac{14256\mu^4}{125} + \frac{3564\mu^3}{125} \right) + \gamma^5 \\
\cdot & \left( \frac{117\mu^8}{10} + \frac{2466\mu^7}{25} + \frac{5567\mu^6}{25} + \frac{124119\mu^5}{500} + \frac{117418481\mu^4}{1000000} \right) + \gamma^4 \\
\cdot & \left( \frac{9\mu^9}{5} + 9\mu^8 + \frac{47169\mu^7}{500} + \frac{50319\mu^6}{250} + \frac{42471\mu^5}{500} \right) + \frac{135999\gamma^3\mu^6}{5000} + \gamma^2 \\
\cdot & \left( \frac{6723\mu^7}{250} + \frac{2286387\mu^3}{500000} \right) + \gamma \left( \frac{279027\mu^5}{200000} + \frac{480789\mu^4}{40000} \right) + \frac{88209\mu^5}{50000}.
\end{aligned} \tag{97}
$$

Thus, we have $p_5(x) > 0$. It means that $x = \frac{9\gamma\mu}{5} + 1$ is an upper bound to all solutions of (56). Thus, we get $\max_{x \in \mathbb{R}^5, \|x\|=1} \left\{ x^\top (\mathbf{I} + \mathbf{C}) x \right\} \leq \frac{9\gamma\mu}{5} + 1$. $\qquad \square$

## D. Auxiliary Results

For all $x, y, x_1, \ldots, x_n \in \mathbb{R}^d$, $s > 0$ and $\alpha \in (0, 1]$, we have:

$$\|x + y\|^2 \leq (1 + s) \|x\|^2 + (1 + s^{-1}) \|y\|^2, \tag{98}$$

$$\|x + y\|^2 \leq 2 \|x\|^2 + 2 \|y\|^2, \tag{99}$$

$$\langle x, y \rangle \leq \frac{\|x\|^2}{2s} + \frac{s \|y\|^2}{2}. \tag{100}$$

**Lemma D.1.** *The inequality $\bar{y} > b + a \log(\bar{y})$ holds for $\bar{y} = 8 \max\{a, b\} \log(\max\{a, b\})$ and for all $a, b \geq 4$.*

*Proof.* Using simple algebra,

$$
\begin{aligned}
& a \log(\bar{y}) + b \\
&= a \log(8 \max\{a, b\} \log(\max\{a, b\})) + b \\
&= \log(8)a + a \log(\max\{a, b\}) + a \log(\log(\max\{a, b\})) + b \\
&\leq \log(8)a + 2a \log(\max\{a, b\}) + b \\
&\leq \log(8)a + 2a \log(a) + 2a \log(b) + b \\
&< 5a \log(a) + 2a \log(b) + b \leq 8 \max\{a, b\} \log(\max\{a, b\}) = \bar{y},
\end{aligned}
$$

where we use $\max\{x, y\} \leq x + y$ and $\log(\log(x)) \leq \log x$ for all $x \geq 4$. $\qquad \square$

**Lemma D.2.**

$$\sum_{s=1}^{\infty} \frac{(1+x)^{s/2}}{(1+x)^s - 1} \leq \frac{20}{x} \log\left(\frac{1}{x}\right)$$

*for all $0 < x \leq \frac{11}{25}$.*

*Proof.*

$$
\sum_{s=1}^{\infty} \frac{(1+x)^{s/2}}{(1+x)^s - 1} = \left( \sum_{s=1}^{\left\lceil \frac{\log(2)}{\log(1+x)} \right\rceil} \frac{(1+x)^{s/2}}{(1+x)^s - 1} + \sum_{s=\left\lceil \frac{\log(2)}{\log(1+x)} \right\rceil + 1}^{\infty} \frac{(1+x)^{s/2}}{(1+x)^s - 1} \right)
$$

$$
\leq \left( \sum_{s=1}^{\left\lceil \frac{\log(2)}{\log(1+x)} \right\rceil} \frac{4}{(1+x)^s - 1} + 2 \sum_{s=\left\lceil \frac{\log(2)}{\log(1+x)} \right\rceil + 1}^{\infty} (1+x)^{-s/2} \right)
$$

since $(1+x)^p = 2$ for $p = \frac{\log(2)}{\log(1+x)}$. Next,

$$
\sum_{s=1}^{\infty} \frac{(1+x)^{s/2}}{(1+x)^s - 1} \leq \left( \sum_{s=1}^{\left\lceil \frac{\log(2)}{\log(1+x)} \right\rceil} \frac{4}{sx} + \frac{2}{\sqrt{1+x} - 1} \right) \leq \frac{4}{x} \left( \log\left( \left\lceil \frac{\log(2)}{\log(1+x)} \right\rceil \right) + 1 \right) + \frac{8}{x}
$$

$$
\leq \frac{20}{x} \log\left(\frac{1}{x}\right).
$$

where we used the standard inequalities $\frac{x}{4} \leq \log(1+x) \leq x$ for all $0 < x \leq 1$, and $x \leq \frac{11}{25}$. $\qquad \square$

**Lemma D.3.** *Let us consider the problem*

$$\max_{x \in \mathbb{R}^{d+p}, \|x\| \leq 1} \left\{ x^\top \mathbf{B} x \right\} \tag{101}$$

*with*

$$\mathbf{B} = \begin{bmatrix} \mathbf{A} & ae^\top \\ ea^\top & bee^\top \end{bmatrix} \in \mathbb{R}^{(d+p) \times (d+p)},$$

$\mathbf{A} \in \mathbb{R}^{d \times d}$, $a \in \mathbb{R}^d$, $b \in \mathbb{R}$, *and* $e := [1, \ldots, 1]^\top \in \mathbb{R}^p$. *Then*

$$\max_{x \in \mathbb{R}^{d+p}, \|x\| \leq 1} \left\{ x^\top \mathbf{B} x \right\} = \max_{x \in \mathbb{R}^{d+1}, \|x\| \leq 1} \left\{ x^\top \mathbf{C} x \right\}, \tag{102}$$

*where*

$$\mathbf{C} = \begin{bmatrix} \mathbf{A} & \sqrt{p}a \\ \sqrt{p}a^\top & pb \end{bmatrix} \in \mathbb{R}^{(d+1) \times (d+1)}.$$

*Proof.* We partition the vector $x$ in two blocks: $x = [y^\top, z^\top]^\top$ and get

$$x^\top \mathbf{B} x = y^\top \mathbf{A} y + b \langle e, z \rangle^2 + 2 \langle a, y \rangle \langle e, z \rangle.$$

The problem from (101) is equivalent to

$$y^\top \mathbf{A} y + b \langle e, z \rangle^2 + 2 \langle a, y \rangle \langle e, z \rangle \to \max_{\substack{y \in \mathbb{R}^d, z \in \mathbb{R}^p, \\ \|y\|^2 + \|z\|^2 \leq 1}}.$$

Let us define $t := \langle e, z \rangle$, then the problem is equivalent to

$$y^\top \mathbf{A} y + bt^2 + 2 \langle a, y \rangle t \to \max_{\substack{y \in \mathbb{R}^d, z \in \mathbb{R}^p, t \in \mathbb{R}, \\ \|y\|^2 + \|z\|^2 \leq 1, t = \langle e, z \rangle}}.$$

We can get a relaxed maximization problem using the inequality $\|z\|_1 \leq \sqrt{p} \|z\|$ :

$$y^\top \mathbf{A} y + bt^2 + 2 \langle a, y \rangle t \to \max_{\substack{y \in \mathbb{R}^d, z \in \mathbb{R}^p, t \in \mathbb{R}, \\ \|y\|^2 + \frac{1}{p} \|z\|_1^2 \leq 1, t = \langle e, z \rangle}}.$$

Since $|t| = |\langle e, z \rangle| \leq \|z\|_1$ , we can relax our problem even further and obtain

$$y^\top \mathbf{A} y + bt^2 + 2 \langle a, y \rangle t \to \max_{\substack{y \in \mathbb{R}^d, t \in \mathbb{R}, \\ \|y\|^2 + \frac{1}{p} t^2 \leq 1}}. \tag{103}$$

Let us take $u = \frac{1}{\sqrt{p}} t$ and get

$$y^\top \mathbf{A} y + pbu^2 + 2\sqrt{p} \langle a, y \rangle u \to \max_{\substack{y \in \mathbb{R}^d, u \in \mathbb{R}, \\ \|y\|^2 + u^2 \leq 1}}.$$

This problem is equivalent to the r.h.s. of (102). Thus,

$$\max_{x \in \mathbb{R}^{d+p}, \|x\| \leq 1} \left\{ x^\top \mathbf{B} x \right\} \leq \max_{x \in \mathbb{R}^{d+1}, \|x\| \leq 1} \left\{ x^\top \mathbf{C} x \right\}.$$

At the same time, we know that the r.h.s. of (102) is equivalent to (103). Let us define $z_i := \frac{t}{p}$ for all $i \in [p]$. Then $t = \langle e, z \rangle$ and $\|z\|^2 = \frac{t^2}{p}$ and the r.h.s. of (102) and (103) are equivalent to

$$y^\top \mathbf{A} y + b \langle e, z \rangle^2 + 2 \langle a, y \rangle \langle e, z \rangle \to \max_{\substack{y \in \mathbb{R}^d, t \in \mathbb{R}, z \in \mathbb{R}^p \\ \|y\|^2 + \|z\|^2 \leq 1, z_i = \frac{t}{p} \forall i \in [p]}}.$$

Let us ignore the equality in the constraint and get a relaxed maximization problem

$$y^\top \mathbf{A} y + b \langle e, z \rangle^2 + 2 \langle a, y \rangle \langle e, z \rangle \to \max_{\substack{y \in \mathbb{R}^d, z \in \mathbb{R}^p \\ \|y\|^2 + \|z\|^2 \leq 1}},$$

which is equivalent to (101). Thus,

$$\max_{x \in \mathbb{R}^{d+p}, \|x\| \leq 1} \left\{ x^\top \mathbf{B} x \right\} \geq \max_{x \in \mathbb{R}^{d+1}, \|x\| \leq 1} \left\{ x^\top \mathbf{C} x \right\}.$$

$\square$

## E. Lower Bound for Perceptron Algorithm on Minimalist Example

**Theorem 5.2** (Lower Bound for Perceptron Algorithm). *The batch perceptron algorithm (Perceptron Algorithm) requires at least*

$$\Omega\left(\tfrac{d}{\mu}\right)$$

*steps to find a separator of the dataset (5) for any $\varepsilon \leq \frac{\mu}{8 \max\{\|a_1\|, \|a_2\|\}}$, $\mu \leq 1/2$, step size $\phi > 0$, and for any starting point $z_0 \in B(0, \phi\varepsilon)$.*

*Proof.* Without loss of generality, we assume that $\phi = 1$, since the method is invariant to the chosen step size $\phi$ if we scale the starting point by $\phi$. In the first step of the algorithm,

$$z_1 = z_0 + \frac{1}{4}(a_1 + a_2) = z_0 + \frac{1}{4} \begin{bmatrix} -\mu & 0 & \dots & 0 \end{bmatrix} = \begin{bmatrix} [z_0]_1 - \frac{1}{4}\mu & [z_0]_2 & \dots & [z_0]_d \end{bmatrix}.$$

Notice that $a_1^\top z_1 = a_1^\top z_0 - \frac{\mu}{4} \leq \varepsilon \|a_1\| - \frac{\mu}{4} \leq 0$ and $a_2^\top z_1 = a_2^\top z_0 + \frac{\mu}{4}(1 + \mu) \geq -\|a_2\|\varepsilon + \frac{\mu}{4} > 0$, where we use that $\varepsilon \leq \frac{\mu}{8 \max\{\|a_1\|, \|a_2\|\}}$. Thus, only the first sample is not correctly classified, leading to the step

$$z_2 = z_1 + \frac{1}{2} \begin{bmatrix} 1 & 1 & \dots & 1 \end{bmatrix} = \begin{bmatrix} \frac{1}{2} + [z_0]_1 - \frac{1}{4}\mu & \frac{1}{2} + [z_0]_2 & \dots & \frac{1}{2} + [z_0]_d \end{bmatrix}.$$

Next,

$$a_1^\top z_2 = a_1^\top z_0 + \frac{d}{2} - \frac{\mu}{4} \geq -\varepsilon \|a_1\| + \frac{d}{2} - \frac{\mu}{4} \geq \frac{d}{2} - \frac{\mu}{2} > 0$$

and

$$a_2^\top z_2 = a_2^\top z_0 - (1 + \mu)\left(\frac{1}{2} - \frac{1}{4}\mu\right) - \frac{d-1}{2}$$
$$= a_2^\top z_0 - \frac{1}{4}\mu + \frac{1}{4}\mu^2 - d \leq \varepsilon \|a_2\| - \frac{d}{2} \leq 0,$$

where we use that $\varepsilon \leq \frac{\mu}{8 \max\{\|a_1\|, \|a_2\|\}}$ and $d \geq \frac{\mu}{2}$. Thus, only the second sample is not correctly classified, leading to the step

$$z_3 = z_2 + \frac{1}{2} \begin{bmatrix} -(1 + \mu) & -1 & \dots & -1 \end{bmatrix} = \begin{bmatrix} [z_0]_1 - (\frac{1}{4} + \frac{1}{2})\mu & [z_0]_2 & \dots & [z_0]_d \end{bmatrix}.$$

Next, $a_1^\top z_3 = a_1^\top z_0 - \frac{3\mu}{4} < 0$ and $a_2^\top z_1 = a_2^\top z_0 + \left(\frac{\mu}{4} + \frac{\mu}{2}\right)(1 + \mu) > 0$. Therefore, only the first sample is not correctly classified, leading to the step

$$z_4 = z_3 + \frac{1}{2} \begin{bmatrix} 1 & 1 & \dots & 1 \end{bmatrix} = \begin{bmatrix} \frac{1}{2} + [z_0]_1 - (\frac{1}{4} + \frac{1}{2})\mu & \frac{1}{2} + [z_0]_2 & \dots & \frac{1}{2} + [z_0]_d \end{bmatrix},$$

Using the same reasoning as for $z_1$ and $z_3$, notice that $a_1^\top z_{2k+1}$ decreases and $a_2^\top z_{2k+1}$ increases. Thus, these periodic steps will repeat until $2k^{\text{th}}$ step, where

$$z_{2k} = z_{2k-1} + \frac{1}{2}\begin{bmatrix}1 & 1 & \cdots & 1\end{bmatrix} = \begin{bmatrix}\frac{1}{2} + [z_0]_1 - (\frac{1}{4} + \frac{k-1}{2})\mu & \frac{1}{2} + [z_0]_2 & \cdots & \frac{1}{2} + [z_0]_d\end{bmatrix},$$

and when, necessarily, either
(Opt. 1):

$$a_2^\top z_{2k} = a_2^\top z_0 - (1+\mu)\left(\frac{1}{2} - \left(\frac{1}{4} + \frac{k-1}{2}\right)\mu\right) - \frac{d-1}{2} \geq 0,$$

meaning that

$$0 < a_2^\top z_0 - (1+\mu)\left(\frac{1}{2} - \left(\frac{1}{4} + \frac{k-1}{2}\right)\mu\right) - \frac{d-1}{2}$$

$$\leq \varepsilon\|a_2\| + \mu\left(\frac{1}{2} + k - 1\right) - \frac{d}{2} \leq \mu + \mu(k-1) - \frac{d}{2}$$

where we use that $\mu \leq 1$. Therefore, $k \geq \frac{d}{2\mu}$. Otherwise, it is necessary that
(Opt. 2):

$$a_1^\top z_{2k} = a_1^\top z_0 + \frac{d}{2} - \left(\frac{1}{4} + \frac{k-1}{2}\right)\mu \leq 0.$$

Thus,

$$0 \geq a_1^\top z_0 + \frac{d}{2} - \left(\frac{1}{4} + \frac{k-1}{2}\right)\mu$$

$$\geq -\varepsilon\|a_1\| + \frac{d}{2} - \left(\frac{1}{4} + \frac{k-1}{2}\right)\mu \geq -\mu + \frac{d}{2} - (k-1)\mu = \frac{d}{2} - k\mu,$$

leading to $k \geq \frac{d}{2\mu}$. In both cases, the total required number of iterations is $\Omega\left(\frac{d}{\mu}\right)$. $\qquad\square$

## F. Multi-Layer Models

We can generalize our result from Sections B.1 and B.2 to the case, when our model is

$$m(b_i; \mathbf{C}_1, \ldots, \mathbf{C}_\ell, v) := (\mathbf{C}_1 \cdots \mathbf{C}_\ell b_i)^\top v, \qquad \mathbf{C}_\ell \in \mathbb{R}^{f_\ell \times d}, \ldots, \mathbf{C}_1 \in \mathbb{R}^{f_1 \times f_2}, v \in \mathbb{R}^{f_1}.$$

In this case, there exists a *symmetric $(\ell+1)$–multilinear map*

$$\mathbf{A}_i : \underbrace{\mathbb{R}^{f_\ell d + \cdots + f_1 f_2 + f_1} \times \ldots \times \mathbb{R}^{f_\ell d + \cdots + f_1 f_2 + f_1}}_{\ell+1 \text{ times}} \to \mathbb{R}$$

such that

$$m(b_i; \mathbf{C}_1, \ldots, \mathbf{C}_\ell, v) = \frac{1}{\ell+1}\mathbf{A}_i[w, \ldots, w]_{\ell+1},$$

where $w = [\text{vec}(\mathbf{C}_1)^\top, \ldots, \text{vec}(\mathbf{C}_\ell)^\top, v^\top]^\top$.

### F.1. One step of GD with multi-layer model

Unlike the quadratic case, we consider GD with adaptive step sizes:

$$w_{t+1} = w_t - \gamma_t \nabla f(w_t),$$

which might be essential to get a reduction to a generalized perceptron algorithm. Similarly to (10),

$$w_{t+1} = w_t + \frac{\gamma_t}{n} \sum_{i=1}^{n} \frac{1}{1 + \exp(\frac{1}{\ell+1} \mathbf{A}_i[w_t, \ldots, w_t]_{\ell+1})} \nabla_{w_t} \left( \frac{1}{\ell+1} \mathbf{A}_i[w_t, \ldots, w_t]_{\ell+1} \right).$$

There exists the $\ell$–multilinear map

$$\mathbf{G}_i : \underbrace{\mathbb{R}^{f_\ell d + \cdots + f_1 f_2 + f_1} \times \cdots \times \mathbb{R}^{f_\ell d + \cdots + f_1 f_2 + f_1}}_{\ell \text{ times}} \to \mathbb{R}^{f_\ell d + \cdots + f_1 f_2 + f_1}$$

such that

$$\mathbf{G}_i[\underbrace{w_t, \ldots, w_t}_{\ell \text{ times}}]_\ell = \nabla_{w_t} \left( \frac{1}{\ell+1} \mathbf{A}_i[\underbrace{w_t, \ldots, w_t}_{(\ell+1) \text{ times}}]_{\ell+1} \right).$$

Therefore,

$$w_{t+1} = w_t + \frac{\gamma_t}{n} \sum_{i=1}^{n} \frac{1}{1 + \exp(\frac{1}{\ell+1} \mathbf{A}_i[w_t, \ldots, w_t]_{\ell+1})} \mathbf{G}_i[w_t, \ldots, w_t]_\ell. \tag{104}$$

## F.2. Reduction to generalized perceptron algorithm

**Theorem F.1.** *Consider the steps*

$$w_{t+1} = w_t + \frac{\gamma_t}{n} \sum_{i=1}^{n} \frac{1}{1 + \exp(\frac{1}{\ell+1} \mathbf{A}_i[w_t, \ldots, w_t]_{\ell+1})} \mathbf{G}_i[w_t, \ldots, w_t]_\ell$$

*with $\gamma_t = \gamma/\|w_t\|^{\ell-1}$ and almost all choices[5] of $\gamma < 1/\max_{i \in [n]} \max_{\|\theta\|=1} \|\mathbf{G}_i[\theta, \ldots, \theta]_\ell\|$. Assume that the direction $\theta_0 := \frac{w_0}{\|w_0\|}$ of the starting point $w_0$ is fixed, $\mathbf{A}_i[\theta_0, \ldots, \theta_0] \neq 0$ for all $i \in [n]$, and $\|w_0\| \to \infty$. For $\|w_0\| \to \infty$, $\frac{w_{t+1}}{\|w_{t+1}\|}$ is well-defined and equals $\frac{\theta_{t+1}}{\|\theta_{t+1}\|}$, where*

$$\theta_{t+1} := \theta_t + \frac{\bar{\gamma}_t}{n} \sum_{i \in S_t} \mathbf{G}_i[\theta_t, \ldots, \theta_t]_\ell, \quad S_t := \left\{ i \in [n] : \frac{1}{\ell+1} \mathbf{A}_i[\theta_t, \ldots, \theta_t]_{\ell+1} \leq 0 \right\}, \tag{105}$$

*and $\bar{\gamma}_t := \gamma/\|\theta_t\|^{\ell-1}$.*

*Proof.* Using mathematical induction, we will prove that $\frac{w_t}{\|w_t\|} \overset{\|w_0\| \to \infty}{=} \frac{\theta_t}{\|\theta_t\|}$, $\|w_t\| \overset{\|w_0\| \to \infty}{\to} \infty$, and $\mathbf{A}_i[\theta_t, \ldots, \theta_t]_{\ell+1} \neq 0$ for all $i \in [n]$ and $t \geq 0$. For $t = 0$, it holds by the assumption. Assume that it holds for $t \geq 0$, then

$$\frac{w_{t+1}}{\|w_{t+1}\|} = \frac{w_t + \frac{\gamma_t}{n} \sum_{i=1}^{n} \frac{1}{1 + \exp(\frac{1}{\ell+1} \mathbf{A}_i[w_t, \ldots, w_t]_{\ell+1})} \mathbf{G}_i[w_t, \ldots, w_t]_\ell}{\left\| w_t + \frac{\gamma_t}{n} \sum_{i=1}^{n} \frac{1}{1 + \exp(\frac{1}{\ell+1} \mathbf{A}_i[w_t, \ldots, w_t]_{\ell+1})} \mathbf{G}_i[w_t, \ldots, w_t]_\ell \right\|}$$

$$= \frac{\frac{w_t}{\|w_t\|} + \frac{\gamma}{n} \sum_{i=1}^{n} \frac{1}{1 + \exp(\frac{\|w_t\|^{\ell+1}}{\ell+1} \mathbf{A}_i[\frac{w_t}{\|w_t\|}, \ldots, \frac{w_t}{\|w_t\|}]_{\ell+1})} \mathbf{G}_i[\frac{w_t}{\|w_t\|}, \ldots, \frac{w_t}{\|w_t\|}]_\ell}{\left\| \frac{w_t}{\|w_t\|} + \frac{\gamma}{n} \sum_{i=1}^{n} \frac{1}{1 + \exp(\frac{\|w_t\|^{\ell+1}}{\ell+1} \mathbf{A}_i[\frac{w_t}{\|w_t\|}, \ldots, \frac{w_t}{\|w_t\|}]_{\ell+1})} \mathbf{G}_i[\frac{w_t}{\|w_t\|}, \ldots, \frac{w_t}{\|w_t\|}]_\ell \right\|},$$

where we use the choice of $\gamma_t$ and the properties of multilineal functions. Using $\frac{w_t}{\|w_t\|} \overset{\|w_0\| \to \infty}{=} \frac{\theta_t}{\|\theta_t\|}$, $\mathbf{A}_i[\theta_t, \ldots, \theta_t]_{\ell+1} \neq 0$ for all $i \in [n]$, and $\|w_t\| \overset{\|w_0\| \to \infty}{\to} \infty$, we have

$$\frac{1}{1 + \exp(\frac{\|w_t\|^{\ell+1}}{\ell+1} \mathbf{A}_i[\frac{w_t}{\|w_t\|}, \ldots, \frac{w_t}{\|w_t\|}]_{\ell+1})} \overset{\|w_0\| \to \infty}{=} \mathbb{1}\left[\mathbf{A}_i[\theta_t, \ldots, \theta_t]_{\ell+1} < 0\right] = \mathbb{1}\left[\mathbf{A}_i[\theta_t, \ldots, \theta_t]_{\ell+1} \leq 0\right].$$

---

[5] There is a set of measure zero that depends on $w_0$ where we can not guarantee the statement of the theorem.

Thus, we get

$$\frac{w_{t+1}}{\|w_{t+1}\|} \stackrel{\|w_0\|\to\infty}{=} \frac{\frac{\theta_t}{\|\theta_t\|} + \frac{\gamma}{n}\sum_{i\in S_t} \mathbf{G}_i[\frac{\theta_t}{\|\theta_t\|},\ldots,\frac{\theta_t}{\|\theta_t\|}]_\ell}{\left\|\frac{\theta_t}{\|\theta_t\|} + \frac{\gamma}{n}\sum_{i\in S_t} \mathbf{G}_i[\frac{\theta_t}{\|\theta_t\|},\ldots,\frac{\theta_t}{\|\theta_t\|}]_\ell\right\|} = \frac{\theta_t + \frac{\gamma}{n\|\theta_t\|^{\ell-1}}\sum_{i\in S_t} \mathbf{G}_i[\theta_t,\ldots,\theta_t]_\ell}{\left\|\theta_t + \frac{\gamma}{n\|\theta_t\|^{\ell-1}}\sum_{i\in S_t} \mathbf{G}_i[\theta_t,\ldots,\theta_t]_\ell\right\|},$$

where the last norm is positive due to the condition on $\gamma$. Using the definition of $\theta_{t+1}$,

$$\frac{w_{t+1}}{\|w_{t+1}\|} \stackrel{\|w_0\|\to\infty}{=} \frac{\theta_{t+1}}{\|\theta_{t+1}\|}.$$

The norm

$$\|w_{t+1}\| = \|w_t\| \left\|\frac{w_t}{\|w_t\|} + \frac{\gamma}{n}\sum_{i=1}^n \frac{1}{1 + \exp(\frac{\|w_t\|^{\ell+1}}{\ell+1}\mathbf{A}_i[\frac{w_t}{\|w_t\|},\ldots,\frac{w_t}{\|w_t\|}]_{\ell+1})}\mathbf{G}_i[\frac{w_t}{\|w_t\|},\ldots,\frac{w_t}{\|w_t\|}]_\ell\right\| \stackrel{\|w_0\|\to\infty}{=} \infty$$

because $\|w_t\| \to \infty$ and the second term converges to $\frac{\|\theta_{t+1}\|}{\|\theta_t\|} > 0$.

It left to show that $\mathbf{A}_i[\theta_{t+1},\ldots,\theta_{t+1}]_{\ell+1} \neq 0$ for almost all choices of $\gamma$. Clearly, $\mathbf{A}_i[\theta_{t+1},\ldots,\theta_{t+1}]_{\ell+1}$ is the polynomial function of $\gamma$. When $\gamma = 0$, $\mathbf{A}_i[\theta_{t+1},\ldots,\theta_{t+1}]_{\ell+1} = \mathbf{A}_i[\theta_t,\ldots,\theta_t]_{\ell+1} \neq 0$ for all $i \in [n]$. Thus, for almost all choices of $\gamma$, $\mathbf{A}_i[\theta_{t+1},\ldots,\theta_{t+1}]_{\ell+1} \neq 0$ for all $i \in [n]$. We have proved the next step of the mathematical induction. $\square$

# G. Extra Experiments

The code was implemented in Python 3 using PyTorch (Paszke et al., 2019) and executed on a machine with 52 CPUs (Intel(R) Xeon(R) Gold 6278C @ 2.60,GHz) and one GPU (Tesla V100-SXM3-32GB). In all experiments, we either use all available samples or uniformly select $n/2$ samples from each class. We run gradient descent (GD) with different model architectures and step sizes. In all experiments, the bias weights are turned off, and the models are initialized with the default PyTorch initialization: the weights of linear and convolution layers are sampled from $\mathcal{U}(-\sqrt{1/\text{input\_features}}, \sqrt{1/\text{input\_features}})$ and $\mathcal{U}(-\sqrt{1/(\text{input\_channels} \times \text{kernel\_size})}, \sqrt{1/(\text{input\_channels} \times \text{kernel\_size})})$, respectively.

### G.1. Experiments with $m_{\text{lin}}$ and $m_{\text{cv}}$ with kernel size $k = 2$

In this section, we verify our experimental conclusions from Section 1.1 and extend the results of Figure 1 to other numbers of samples, other classes, and datasets.

G.1.1. CIFAR-10 WITH WITH CLASSES 0 AND 1 AND $n = 5000$ SAMPLES

Here, we repeat the experiment from Figure 1, adding plots of the losses. The conclusions are the same as Figure 1: the nonlinear model converges much faster. Notice that the losses are also non-stable and chaotic even though our setup is fully deterministic.

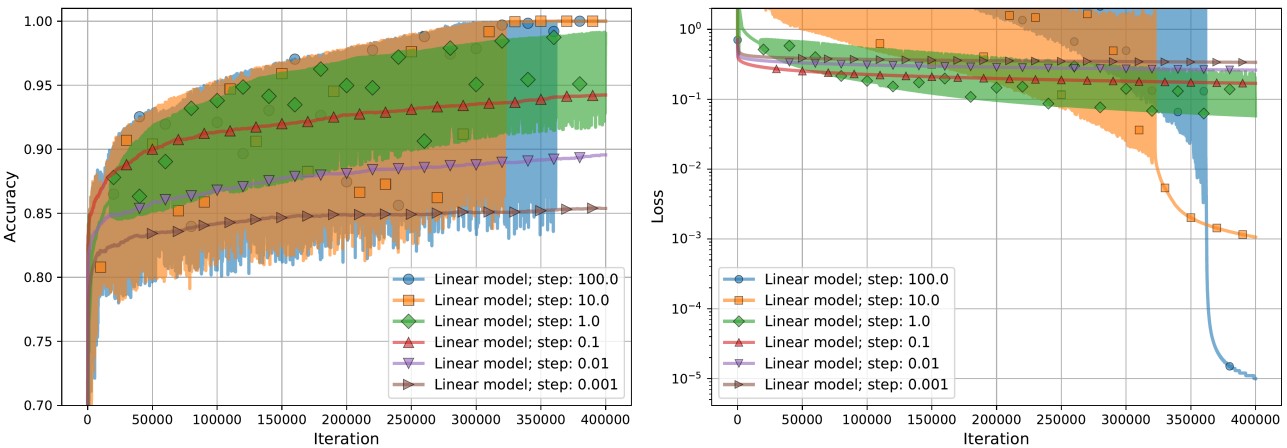

*Figure 4.* Linear model $m_{\text{lin}}$ with classes 0 and 1 and $n = 5000$ samples on CIFAR-10

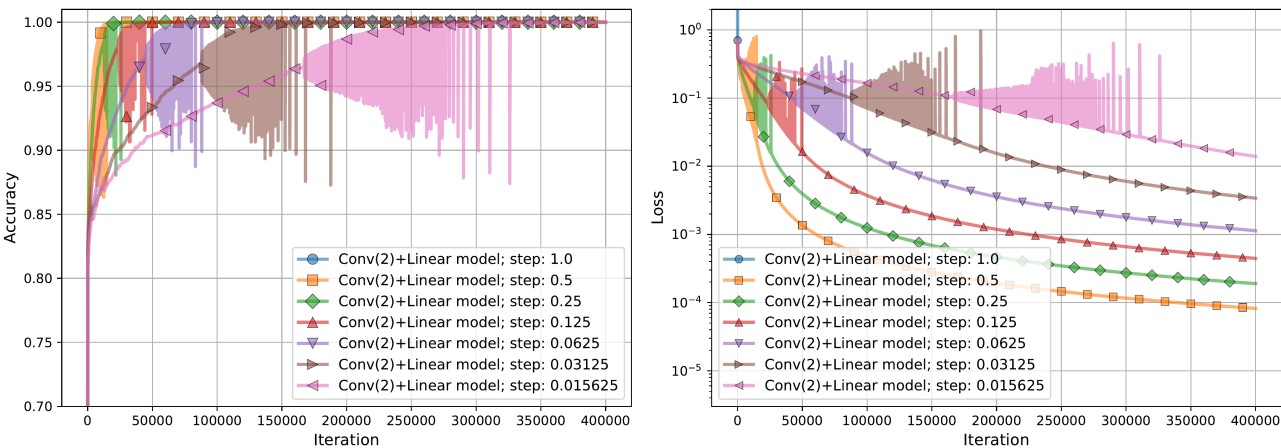

*Figure 5.* Non-linear model $m_{\text{cv}}$ with kernel size $k = 2$, classes 0 and 1, and $n = 5000$ samples on CIFAR-10

G.1.2. CIFAR-10 WITH WITH CLASSES 3 AND 7 AND $n = 5000$ SAMPLES

The setup in this experiment is the same as in Section G.1.1, except that we consider classes 3 and 7 (the pair is chosen randomly). We can see that after 400K iterations, $m_{\text{lin}}$ does not find a separator (the best accuracy is below 1.0), whereas $m_{\text{cv}}$ requires at most 100K iterations.

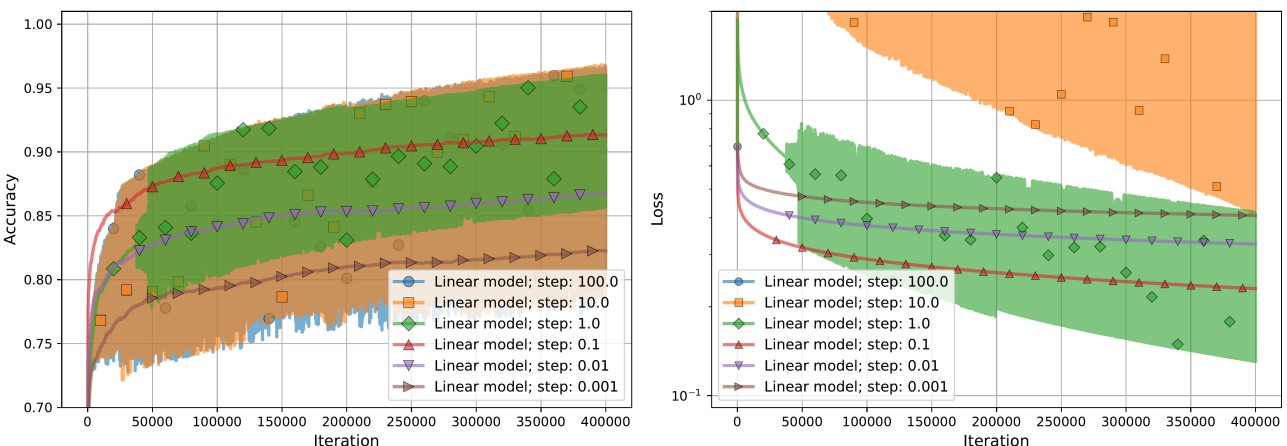

*Figure 6.* Linear model $m_{\text{lin}}$ with classes 3 and 7 and $n = 5000$ samples on CIFAR-10

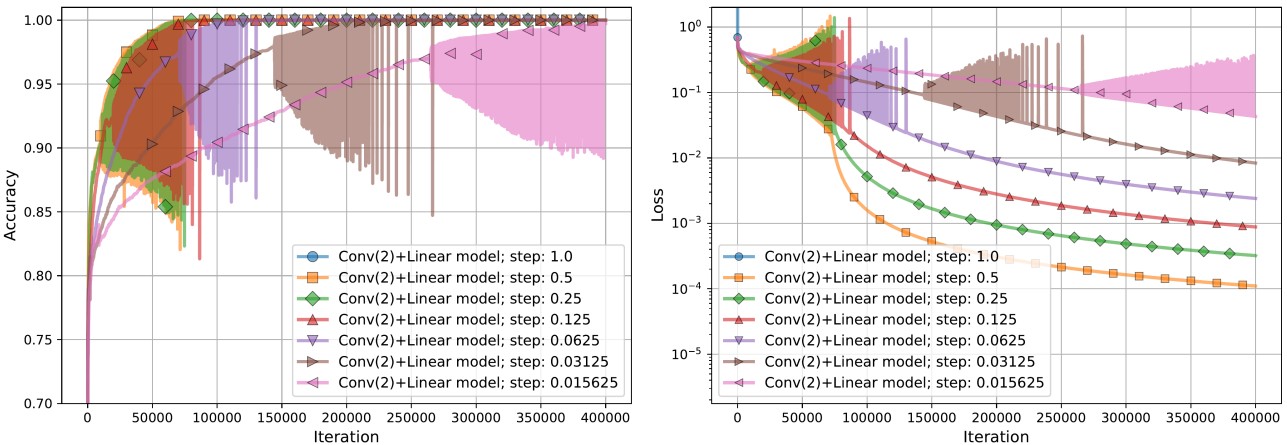

*Figure 7.* Non-linear model $m_{\text{cv}}$ with kernel size $k = 2$, classes 3 and 7, and $n = 5000$ samples on CIFAR-10

### G.1.3. CIFAR-10 WITH WITH CLASSES 0 AND 1 AND ALL SAMPLES

Unlike Section G.1.1, we take all samples from both classes. In this setting, both models do not find a separator after 400K iterations, but the nonlinear model converges to a higher accuracy.

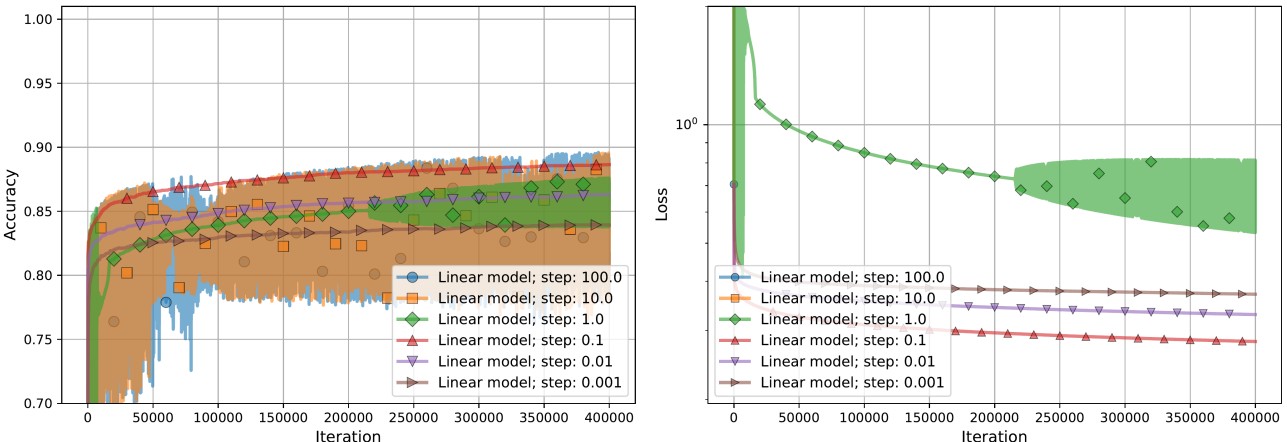

*Figure 8.* Linear model $m_{\text{lin}}$ with classes 0 and 1 and all samples on CIFAR-10

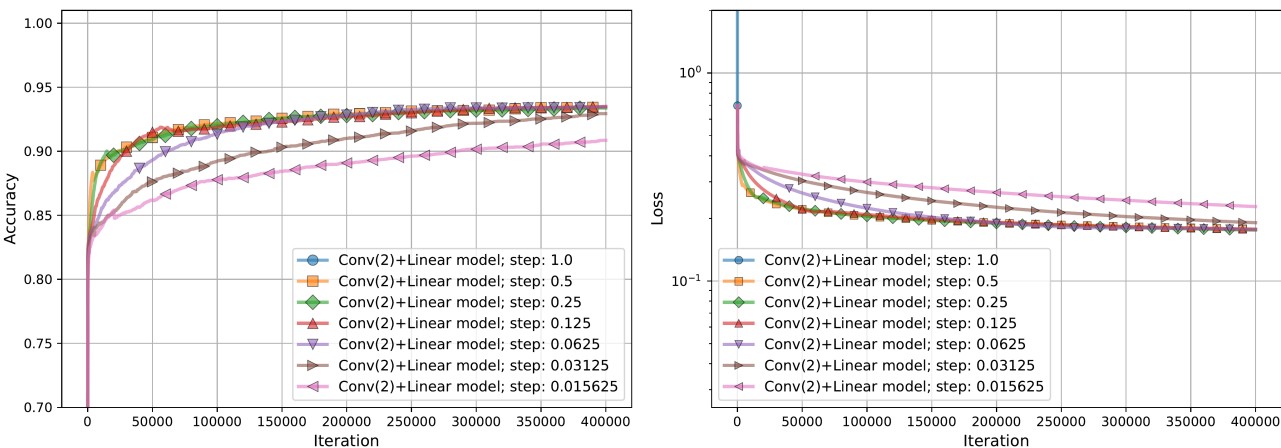

*Figure 9.* Non-linear model $m_{\text{cv}}$ with kernel size $k = 2$, classes 0 and 1, and all samples on CIFAR-10

### G.1.4. EUROSAT WITH WITH CLASSES 3 AND 7 AND $n = 5000$ SAMPLES

We now turn to other datasets: EuroSAT (Helber et al., 2019), FashionMNIST (Xiao et al., 2017), Food101 (Bossard et al., 2014), MNIST (LeCun et al., 2010), and Gisette (Guyon et al., 2004). Across these datasets, for various choices of classes and sample sizes, we observe that the nonlinear model generally finds separating solutions much faster. The only exception occurs on MNIST (see Section G.1.11), where the linear model converges to a separator more quickly than the nonlinear one. The reason for this behavior is not entirely clear; we hypothesize that MNIST is comparatively simpler than the other datasets, with larger inter-class margins, making it easier for linear models to find a separator.

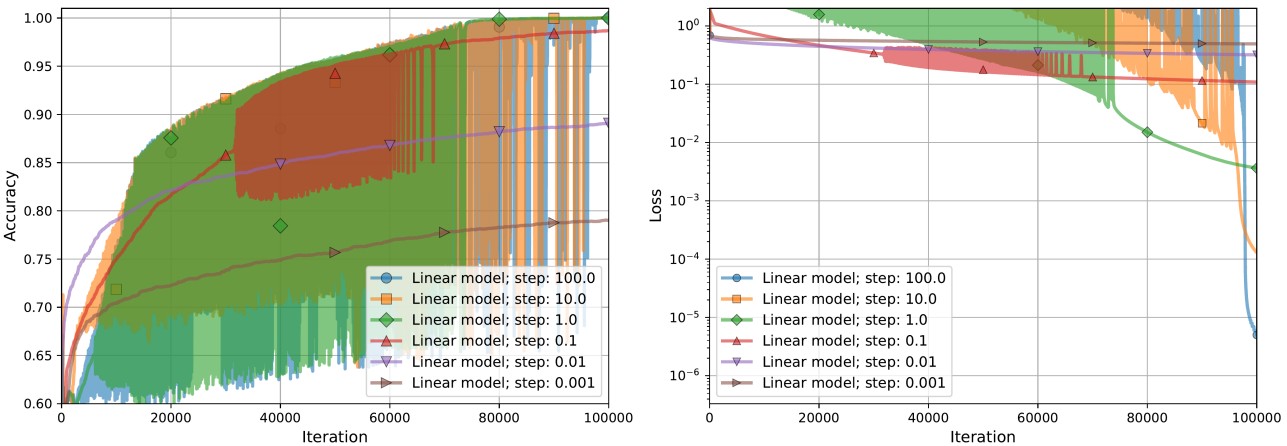

*Figure 10.* Linear model $m_{\text{lin}}$ with classes 3 and 7 and $n = 5000$ samples on EuroSAT

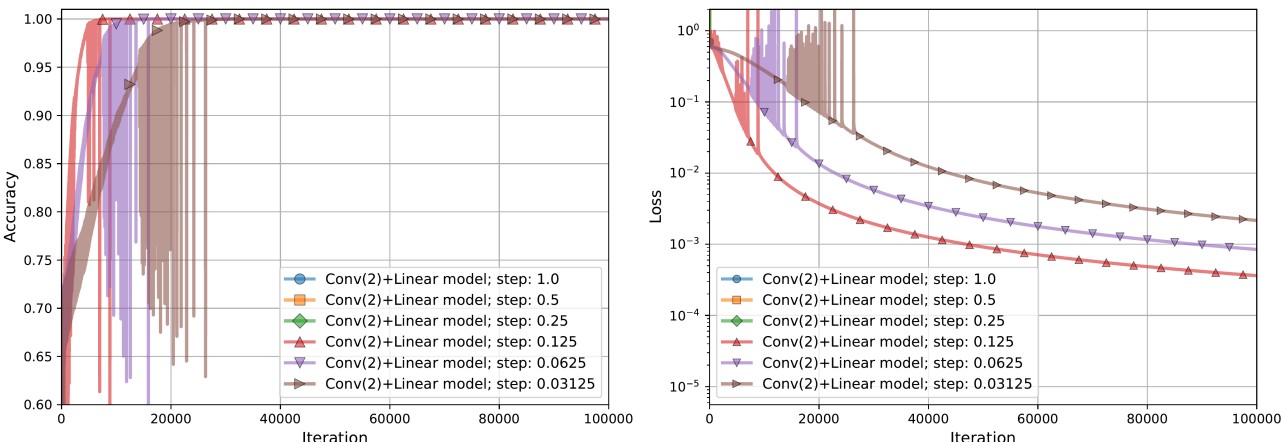

*Figure 11.* Non-linear model $m_{\text{cv}}$ with kernel size $k = 2$, classes 3 and 7, and $n = 5000$ samples on EuroSAT

### G.1.5. EuroSAT with with classes 0 and 1 and $n = 5000$ samples

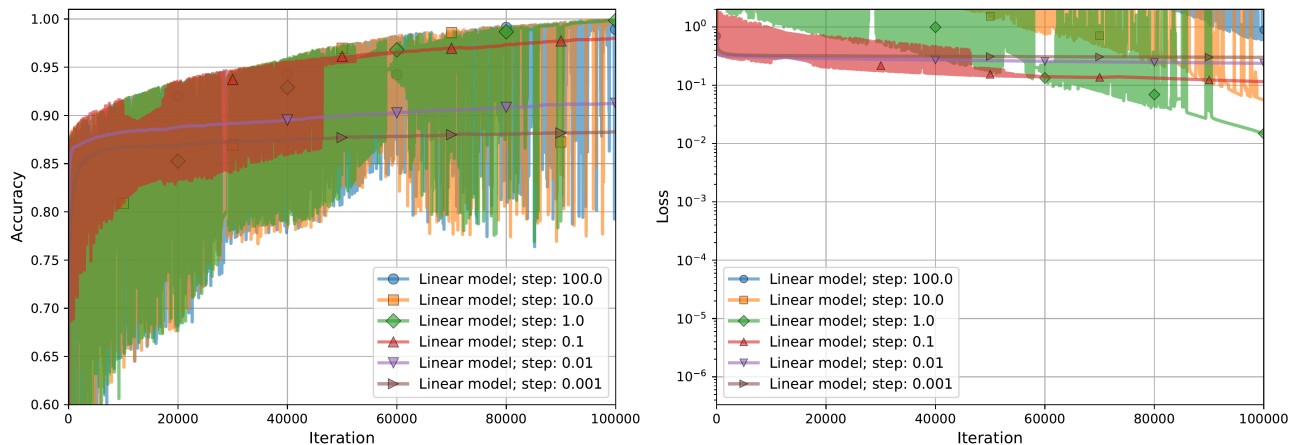

*Figure 12.* Linear model $m_{\text{lin}}$ with classes 0 and 1 and $n = 5000$ samples on EuroSAT

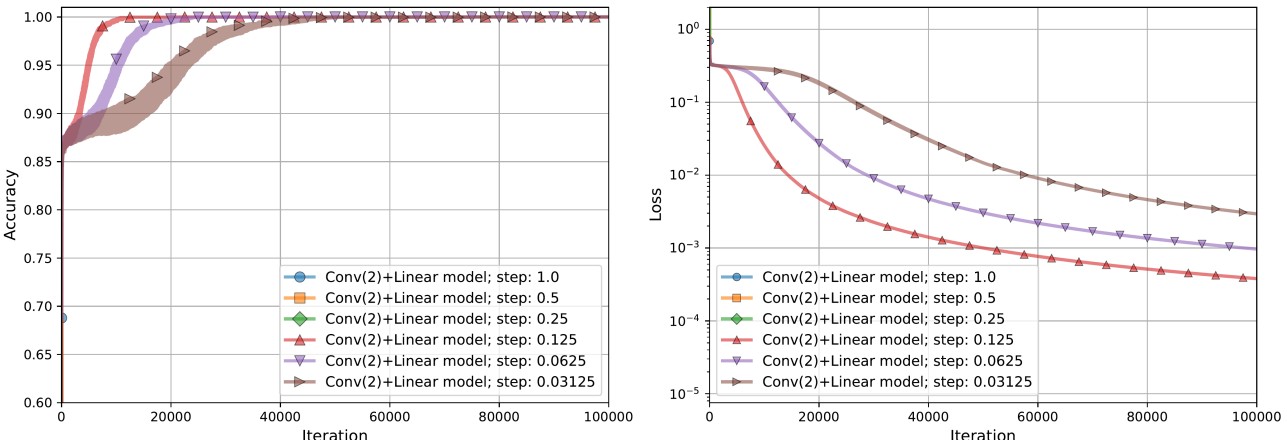

*Figure 13.* Non-linear model $m_{\text{cv}}$ with kernel size $k = 2$, classes 0 and 1, and $n = 5000$ samples on EuroSAT

### G.1.6. FASHIONMNIST WITH WITH CLASSES 5 AND 7 AND $n = 5000$ SAMPLES

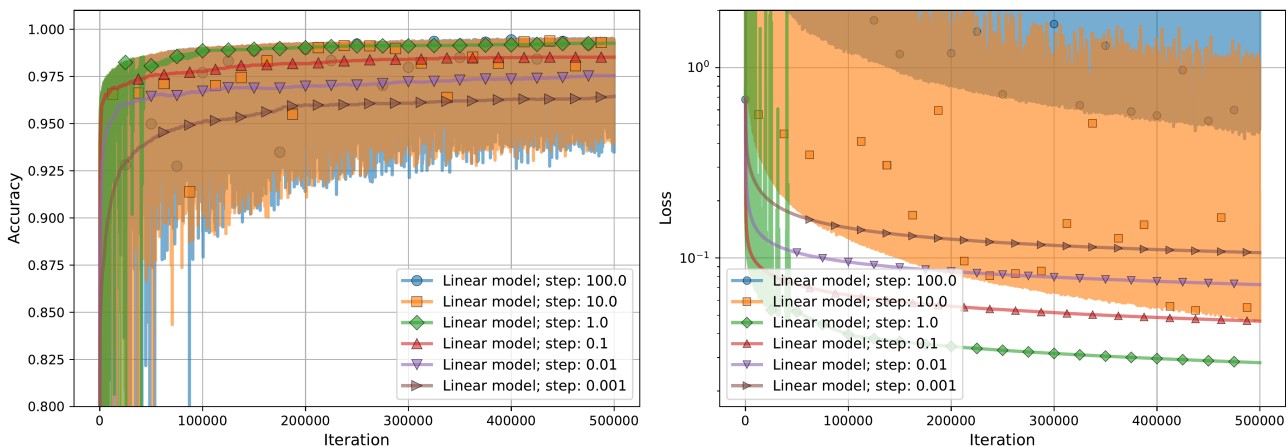

*Figure 14.* Linear model $m_{\text{lin}}$ with classes 5 and 7 and $n = 5000$ samples on FashionMNIST

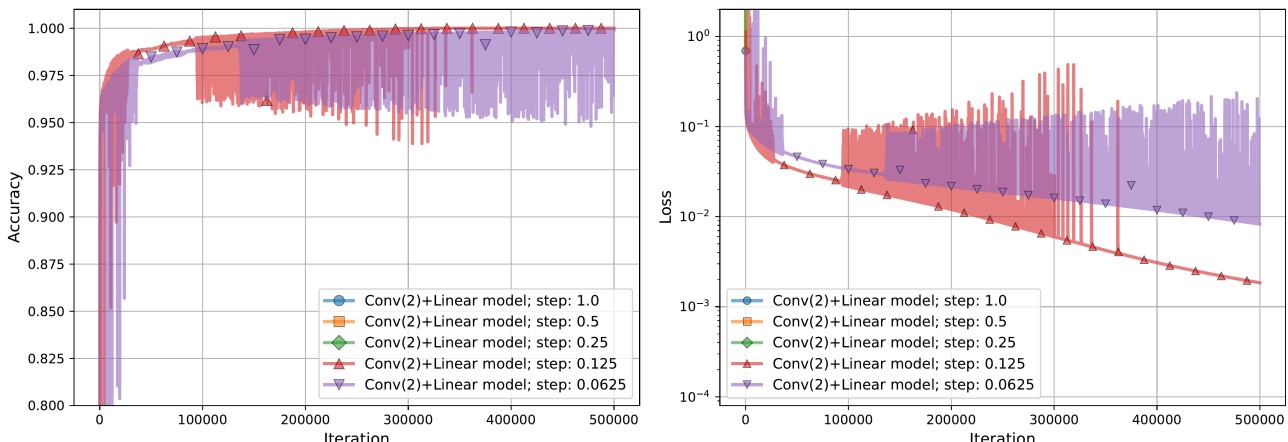

*Figure 15.* Non-linear model $m_{\text{cv}}$ with kernel size $k = 2$, classes 5 and 7, and $n = 5000$ samples on FashionMNIST

G.1.7. FASHIONMNIST WITH WITH CLASSES 7 AND 9 AND $n = 5000$ SAMPLES

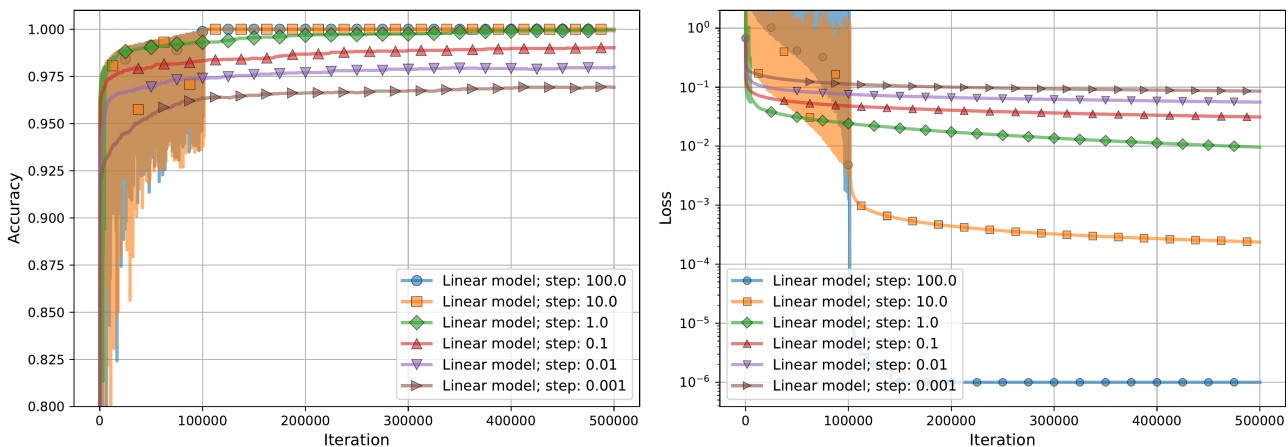

*Figure 16.* Linear model $m_{\text{lin}}$ with classes 7 and 9 and $n = 5000$ samples on FashionMNIST

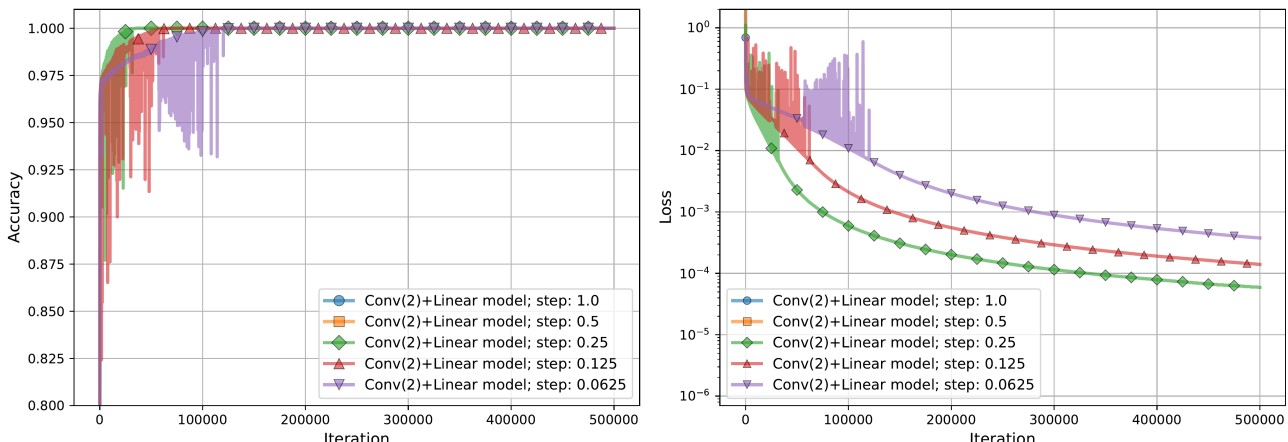

*Figure 17.* Non-linear model $m_{\text{cv}}$ with kernel size $k = 2$, classes 7 and 9, and $n = 5000$ samples on FashionMNIST

### G.1.8. FOOD101 WITH WITH CLASSES 0 AND 1 AND $n = 10000$ SAMPLES

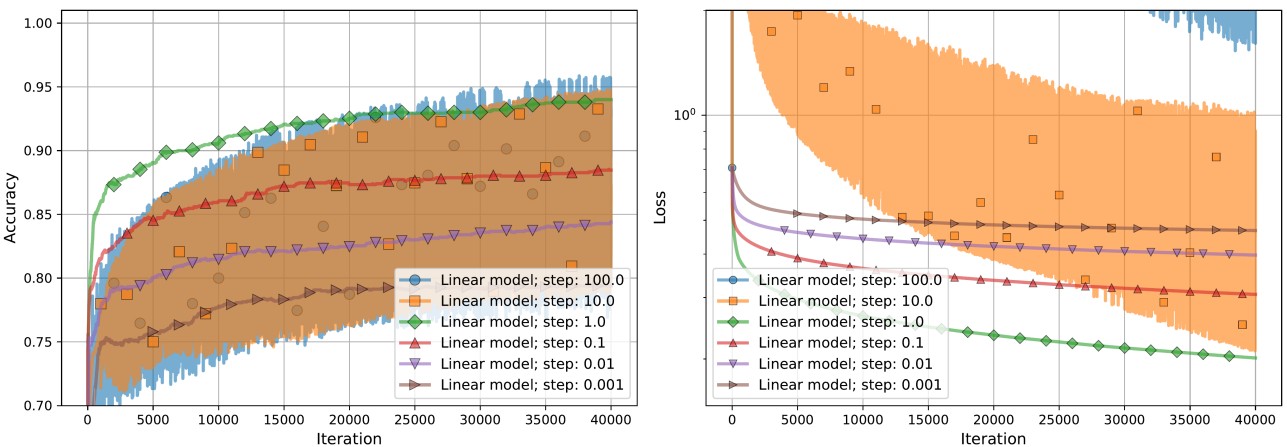

*Figure 18.* Linear model $m_{\text{lin}}$ with classes 0 and 1 and $n = 10000$ samples on Food101

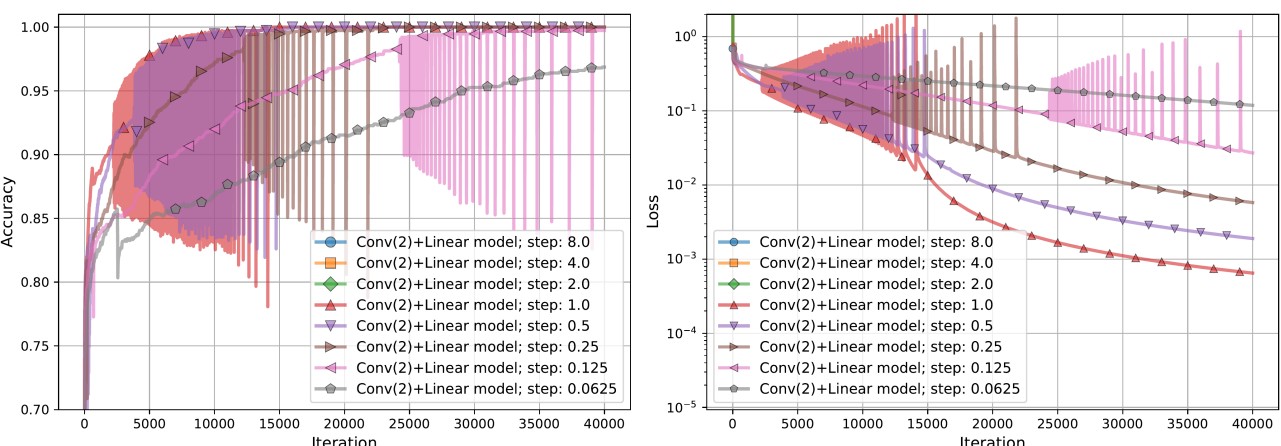

*Figure 19.* Non-linear model $m_{\text{cv}}$ with kernel size $k = 2$, classes 0 and 1, and $n = 10000$ samples on Food101

G.1.9. FOOD101 WITH WITH CLASSES 3 AND 7 AND $n = 10000$ SAMPLES

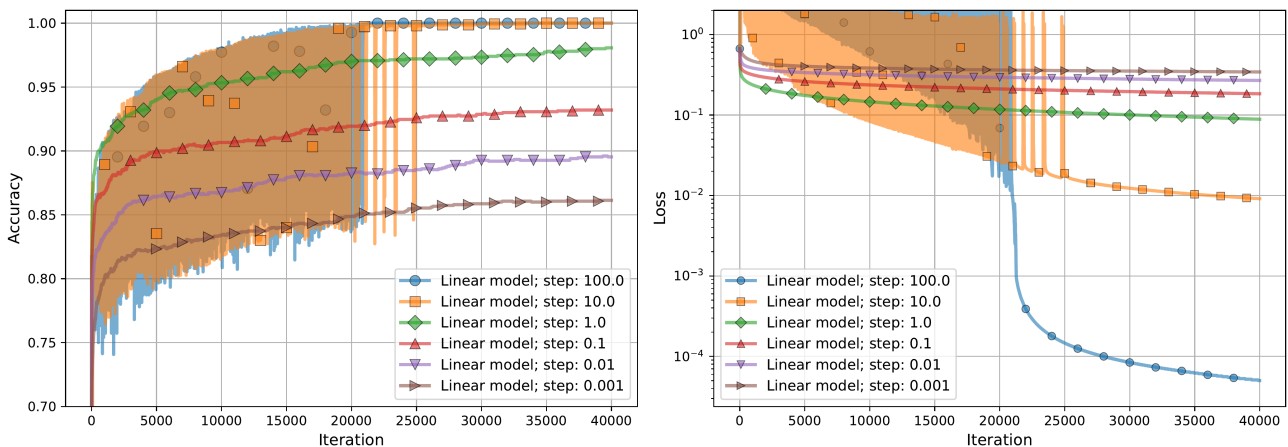

*Figure 20.* Linear model $m_{\text{lin}}$ with classes 3 and 7 and $n = 10000$ samples on Food101

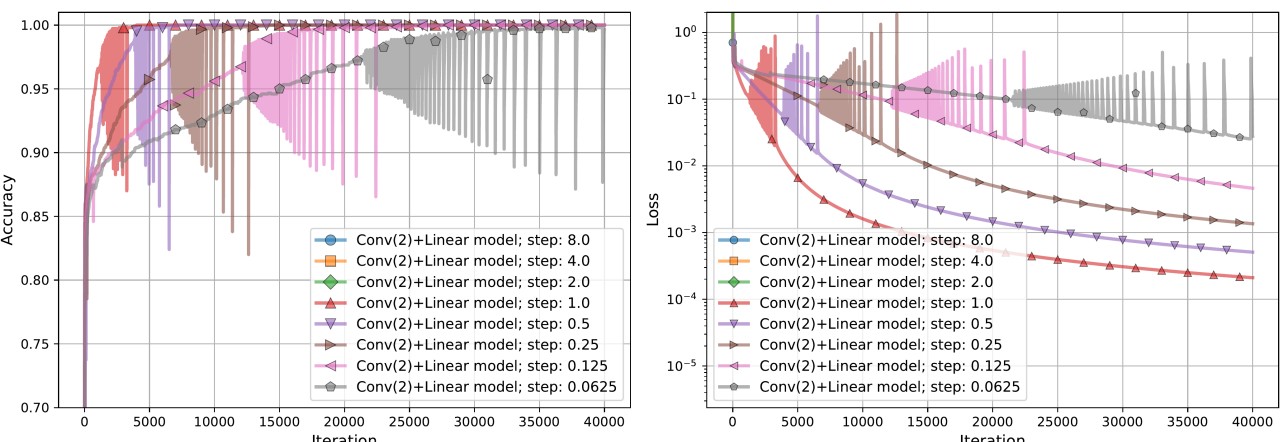

*Figure 21.* Non-linear model $m_{\text{cv}}$ with kernel size $k = 2$, classes 3 and 7, and $n = 10000$ samples on Food101

G.1.10. MNIST WITH WITH CLASSES 0 AND 1 AND ALL SAMPLES

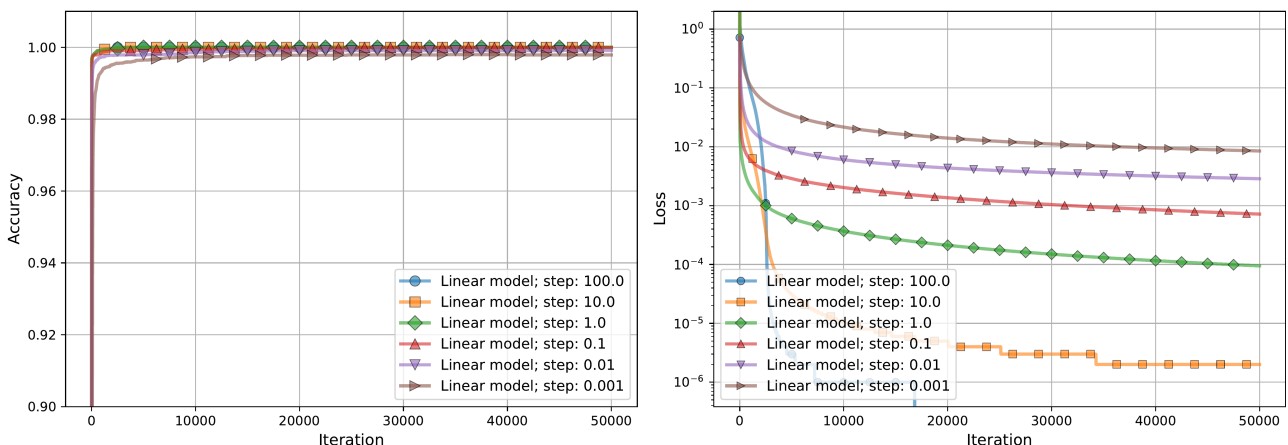

*Figure 22.* Linear model $m_{\text{lin}}$ with classes 0 and 1 and all samples on MNIST

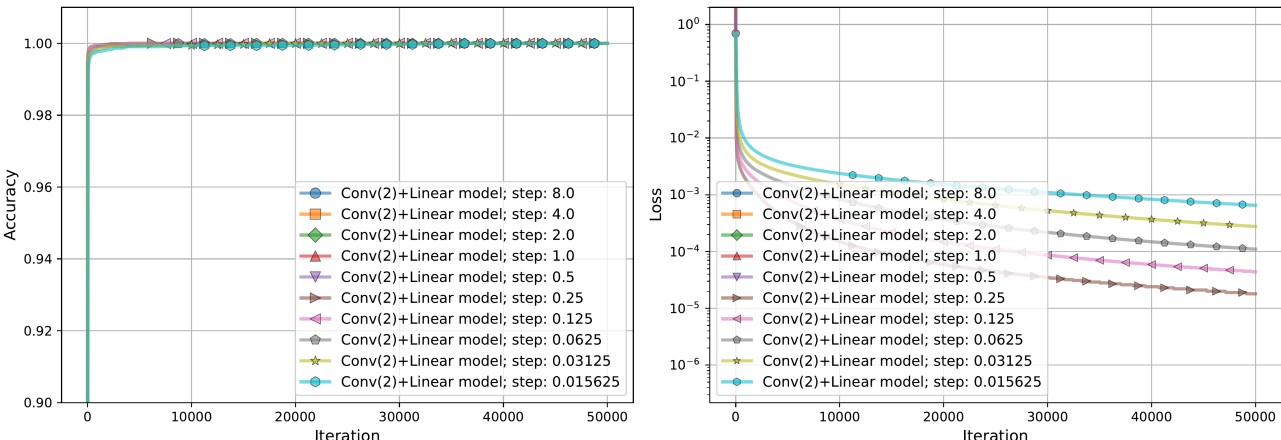

*Figure 23.* Non-linear model $m_{\text{cv}}$ with kernel size $k = 2$, classes 0 and 1, and all samples on MNIST

G.1.11. MNIST WITH WITH CLASSES 7 AND 8 AND ALL SAMPLES

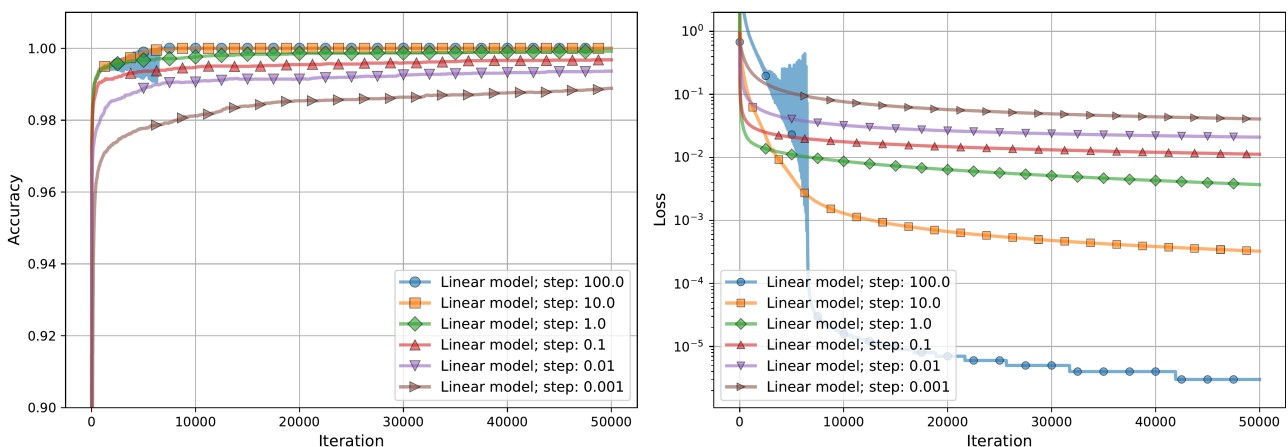

*Figure 24.* Linear model $m_{\text{lin}}$ with classes 7 and 8 and all samples on MNIST

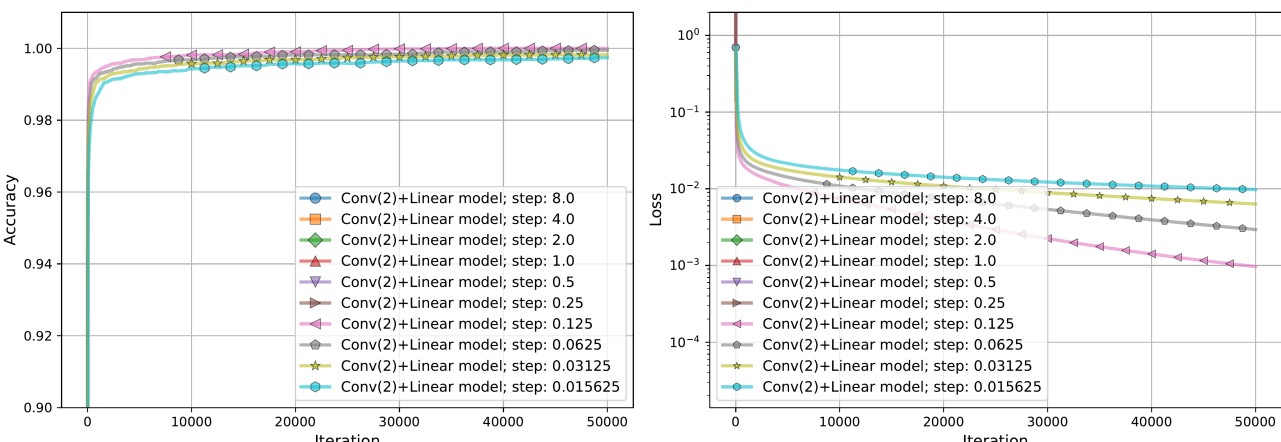

*Figure 25.* Non-linear model $m_{\text{cv}}$ with kernel size $k = 2$, classes 7 and 8, and all samples on MNIST

### G.1.12. GISETTE WITH ALL SAMPLES

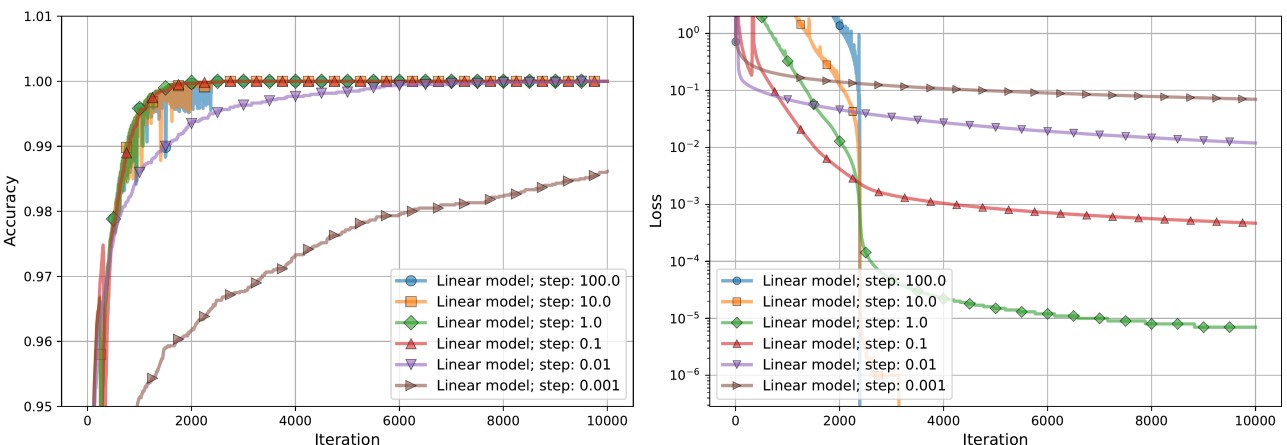

*Figure 26.* Linear model $m_{\text{lin}}$ with all samples on Gisette

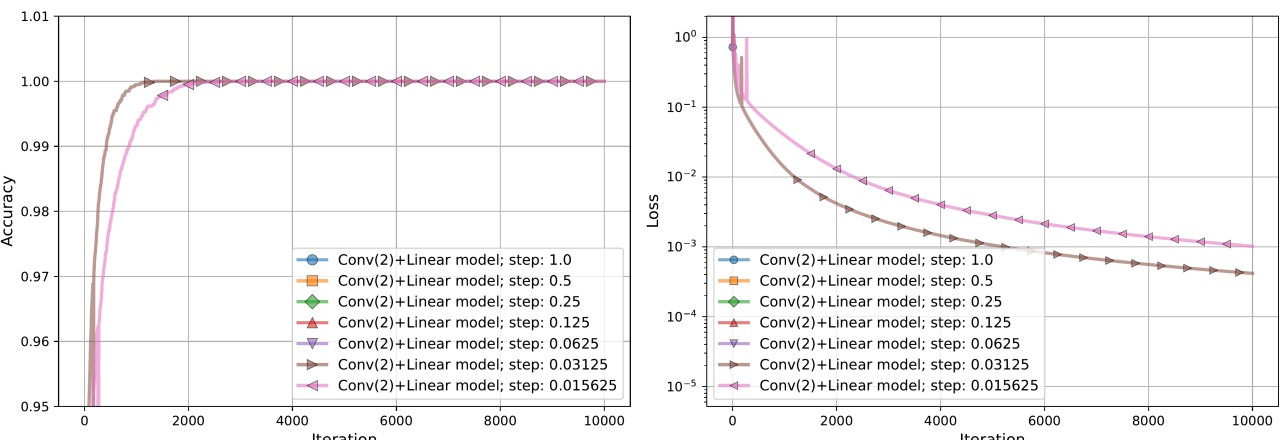

*Figure 27.* Non-linear model $m_{\text{cv}}$ with kernel size $k = 2$, and all samples on Gisette

### G.2. The largest step size vs. kernel size on dataset (5)

In Table 2, we show how the choice of the largest step sizes changes as the kernel size of the nonlinear model $m_{\mathrm{cv}}$ increases on dataset (5). Similar to the experiments with CIFAR-10 in Figure 3, we observe that the step size decreases with an increasing kernel size.

*Table 2.* The largest step size vs. kernel size on dataset (5) with nonlinear model $m_{\mathrm{cv}}$.

| Kernel size | Best step size |
| :---: | :---: |
| 10 | 0.0158 |
| 100 | 0.0079 |
| 1000 | 0.0020 |

### G.3. The monotonically increasing nature of the iterates' norms

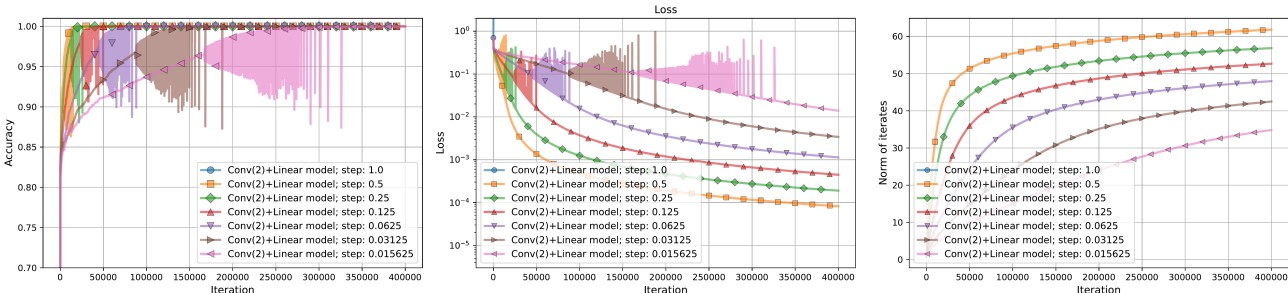

*Figure 28.* Non-linear model $m_{cv}$ with kernel size $k = 2$, classes 0 and 1, and $n = 5000$ samples on CIFAR-10

In this section, we present the same experiment as in Figure 1b and Figure 5. However, we additionally show how the norms of the iterates change with the number of iterations. The main observation in Figure 28 is that the iterates' norms increase.

### G.4. Increasing the initial norm of the model

In this section, we study the robustness of the implicit acceleration as we increase the norm of the initial iterate. In Figures 29 and 30, we present the same experiments as in Figures 1b and 5, with the only difference being that the initial point is scaled by 10 or 100, and we observe that the nonlinear model still converges faster than the linear model in Figure 1a.

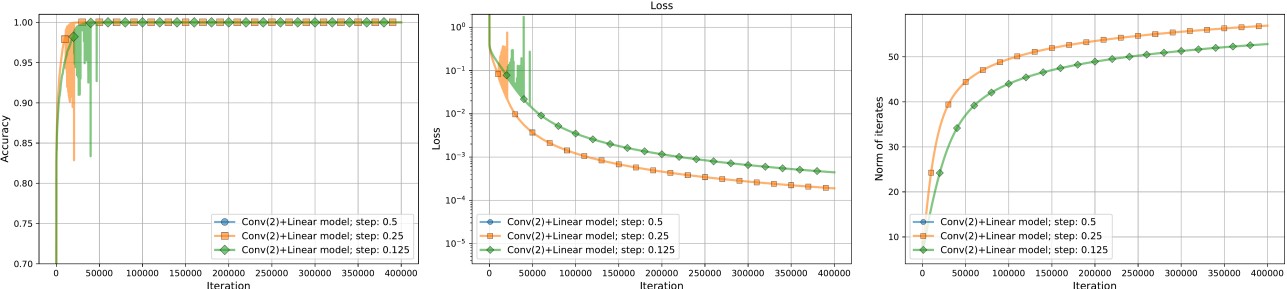

*Figure 29.* Non-linear model $m_{cv}$ with kernel size $k = 2$, classes 0 and 1, $n = 5000$ samples on CIFAR-10, and scaled the initial iterate by 10

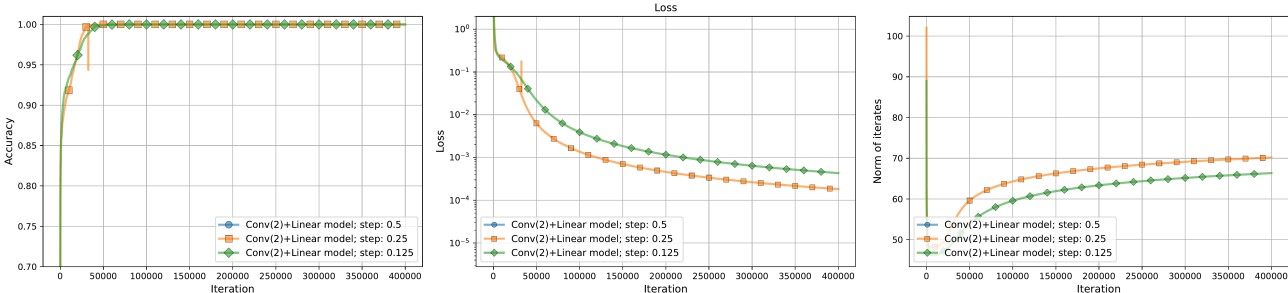

*Figure 30.* Non-linear model $m_{cv}$ with kernel size $k = 2$, classes 0 and 1, $n = 5000$ samples on CIFAR-10, and scaled the initial iterate by 100

### G.5. ResNet and CIFAR-10

We consider the image classification problem on CIFAR-10 (Krizhevsky et al., 2009) using ResNet18 (He et al., 2016) without batch normalization (BN) (Ioffe & Szegedy, 2015) and with the ELU activation (Clevert et al., 2015). Since we consider the binary optimization problems, we only take classes 0 and 1 in CIFAR-10. All methods start from the same

starting point randomly generated by PyTorch. In Figure 31, we observe behavior similar to that in Figure 1: for small $\gamma$, the plots are smooth and the losses decrease monotonically, but starting from $\gamma = 0.01$, the convergence becomes chaotic, even though GD with the logistic loss is deterministic.

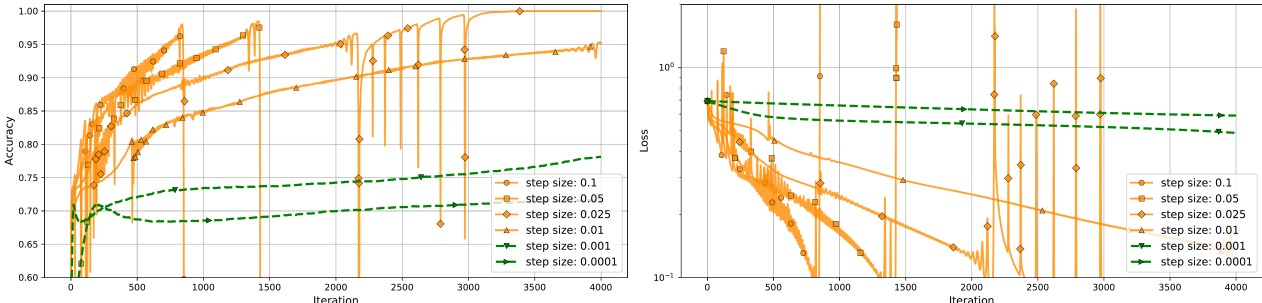

*Figure 31.* Training ResNet18 on CIFAR-10 with the logistic loss and the vanilla GD method.

