# OpenReview forum: "Gradient Descent as a Perceptron Algorithm: Understanding Dynamics and Implicit Acceleration"
_ICML.cc/2026/Conference — ICML 2026 regular_

### Official Review · Reviewer_4ybP · 2026-03-07

**Soundness:** 2
**Presentation:** 3
**Significance:** 1
**Originality:** 2
**Overall Recommendation:** 2
**Confidence:** 5

**Summary:**

This paper investigates the optimization dynamics of gradient descent (GD) applied to non-linear models, with a particular focus on the observed phenomena of implicit acceleration and non-monotonic convergence. The authors link gradient descent to variants of the classic perceptron algorithm. The study demonstrates that under the assumption of large initial weight norms, the GD updates with logistic loss can be equivalently reduced to a quadratic perceptron algorithm. This reduction enables the authors to analyze the convergence properties of the algorithm leveraging standard linear algebra tools. Furthermore, the paper explores extending this analytical framework to multi-layer models. Overall, the work attempts to offer a theoretical perspective for understanding implicit acceleration and the dynamics of gradient-based optimization in neural network training.

**Compliance With Llm Reviewing Policy:**

Affirmed.

**Final Justification:**

I maintain my initial rating, as the key concerns regarding the theoretical assumptions, the justification of the large norm, and the transition from practical initialization remain largely unresolved. The authors’ rebuttal does not sufficiently address these issues, so my rating remains unchanged.

**Key Questions For Authors:**

(1) Much of the theoretical analysis relies on the large initialization norm assumption \mid\mid w_0\mid\mid\rightarrow\infty, under which the GD dynamics can be approximated by the quadratic perceptron algorithm. However, in practical training neural networks are typically initialized with standard Gaussian initialization where \|w_0\| \approx O(1). Could the authors provide a quantitative criterion or threshold on \mid\mid w_t\mid\mid under which the dynamics of GD can be guaranteed to approximate the theoretical regime within some tolerance (e.g., direction deviation <\epsilon)? If such a threshold cannot be derived, could the authors provide any transition analysis showing how GD starting from standard initialization eventually enters the regime assumed in Theorem 2.1?

(2) The convergence result in Theorem 5.1 is established for a highly simplified dataset containing only two samples (n=2), and the analysis relies on the special spectral structure arising from the alternating action of two matrices A_1 and A_2. If the dataset contains n>2 linearly separable samples, the update dynamics would involve multiple non-commuting matrices. In such a scenario, does the exponential growth (or ping-pong effect) described in the paper still hold theoretically? Could the authors provide either theoretical arguments or empirical evidence supporting that the claimed convergence behavior extends beyond the n=2 setting?

(3) In the MNIST experiments (Fig. 24–25), the results show that the linear model converges faster than the nonlinear model, which appears to contradict the central claim that nonlinear parameterizations can accelerate optimization. Could the authors provide a more quantitative explanation using their matrix analysis framework? For instance, is it possible to analyze the spectral structure of the matrices A_i induced by the MNIST dataset to explain why the quadratic perceptron dynamics do not lead to faster convergence in this case?

(4) The loss function considered in the paper is limited to the logistic loss. More complex loss functions commonly used in machine learning or deep learning are not discussed. It would strengthen the paper to investigate whether the proposed method can also be applied to or perform well under more general loss functions.

(5) Although the theoretical analysis is quite complex, the practical significance of the results is not very clear.

**Limitations:**

Yes. The authors briefly acknowledge some limitations of their work in the conclusion (Section 8), noting that their analysis “only scratches the surface” and that extending the theoretical results remains an open problem. While this acknowledgment is appreciated, the discussion of limitations remains relatively brief and could be further elaborated. In particular, the paper does not sufficiently discuss the practical implications of several key theoretical assumptions, such as the requirement of the large initialization norm regime\|w_0\| -->\infty. It would be helpful for the authors to clarify how realistic this assumption is in practical training settings and what its limitations imply for the applicability of the proposed theory. A clearer articulation of these limitations would help readers better understand the contexts in which the proposed insights are expected to apply.

**Strengths And Weaknesses:**

Strengths

a)	This paper rigorously reduces and equates the gradient descent process of nonlinear models with logistic loss to a novel Quadratic Perceptron Algorithm, providing a mathematically perspective on optimization dynamics.

b)	The theoretical analysis bounding the iteration complexity for the proposed Quadratic Perceptron Algorithm on the toy dataset is technically highly complex.

Weaknesses

a)	The paper claims to explain optimization dynamics and implicit acceleration in neural networks, yet the theoretical analysis is limited to a highly simplified two-layer linear model without nonlinear activations. While useful for analysis, it remains unclear whether the insights obtained in this setting generalize to modern deep neural networks with highly non-convex optimization landscapes.

b)	Theorem 2.1 heavily relies on the extreme assumption that the initial weight norm approaches infinity. Although the authors observe in the appendix experiments that the norm gradually increases, the paper provides absolutely no theoretical justification to prove how the model stably and inevitably warms up from a standard, small random initialization and converges to this extreme perceptron state. This deprives Theorem 2.1 of a realistic physical foundation for training.

c)	The main theoretical guarantee (Theorem 5.1) is established for a very minimal binary classification dataset consisting of only two samples. Although such toy settings can provide intuition, it is not clear whether the convergence behavior observed in this extremely small-scale scenario extends to more realistic data distributions with larger sample sizes. Additional theoretical discussion or empirical validation on more general settings would strengthen the claims.

---

> ### Author Rebuttal · Authors · 2026-03-26
>
> Thank you for the review!
>
> **(W.a)**
>
> In the paper, we only claim that our theoretical foundation helps to explain the phenomenon observed in Section 1.1. We are extra cautious with the wording and never try to overclaim our results. Nevertheless, our work provides new and, to the best of our knowledge, the first theoretical insights into the implicit acceleration of a non-linear model observed in Section 1.1. We strongly believe that this is an important starting point toward understanding the dynamics of optimization methods in training.
>
> **(W.b)**
>
> The assumption that the initial weight norm approaches infinity is not extreme at all, since it can always be enforced in practice. First, we observe in Section G.3 that the iterates automatically tend toward large norms. Second, we also have additional experiments (see **[Extra Experiments](https://www.dropbox.com/scl/fi/qzru23yso830q3dli9rf3/plots.png?rlkey=jmqrubrg2h9q1b6txe1wo9loj&st=bh3zbsh5&dl=0)**) for the case where the initial iterate is scaled by 10 and 100 (two last rows in the new plot), and we observe that the nonlinear model still converges faster than the linear one (first row).
>
> We do not aim to solve all possible problems in this field or to fully explain why the norms increase. Nevertheless, beyond our observations, there is theoretical evidence predicting that the norm of the model increases (e.g., [1,2]).
>
> **(W.c)**
>
> Empirically, this result extends to many datasets and classification problems. In Section G, we run many more experiments on additional datasets and observe that on almost all datasets, including EuroSAT, FashionMNIST, Food101, and Gisette, not only CIFAR10.
>
> **(Q.1)**
>
> In its current form, Theorem 2.1 and the subsequent results rely heavily on the assumption that the initial point has a large norm. Note that even in this regime, our results are new and insightful. Extending them to the regime where the initial point has a small norm is important but significantly more technical; we rely on the fact that all operations are linear (see “How can the reduction help us?” in Section 2). Also, consider the discussion “Numerical observation of the norms” in Section 2, where we explain that the norm of the iterates increases. Thus, the assumption that the initial norm is large is practical after the first “warm-up” steps.
>
> **(Q.2)**
>
> Even with $n = 2,$ the update dynamics involve multiple *non-commuting* matrices, making the analysis for $n = 2$ non-trivial. Another problem is that the index (1 or 2) of the matrix at the current step depends on the previous steps, which does not allow us to use classical linear algebra convergence results.
>
> > In such a scenario, does the exponential growth (or ping-pong effect) described in the paper still hold theoretically? Could the authors provide either theoretical arguments or empirical evidence supporting that the claimed convergence behavior extends beyond the n=2 setting?
>
> Theoretically, we do not know. Empirically, we observe it on many datasets. Note that we added additional experiments where we scale the initial iterate by 10 and 100 and observe the acceleration (see **[Extra Experiments](https://www.dropbox.com/scl/fi/qzru23yso830q3dli9rf3/plots.png?rlkey=jmqrubrg2h9q1b6txe1wo9loj&st=bh3zbsh5&dl=0)**).
>
> **(Q.3)**
>
> Yes, we tried to debug the reason and even attempted to go deeper, as the reviewer suggested. However, it is neither feasible nor clear how to obtain a meaningful answer to this question by analyzing 10K samples and matrices.
>
> **(Q.4)**
>
> The logistic loss is one of the most popular losses in practice. Focusing on it and its structure is an important endeavor, and many works are devoted entirely to the logistic loss (see [2–6]).
>
> **(Q.5)**
>
> Our work provides new and, to the best of our knowledge, the first theoretical insights into the implicit acceleration of a nonlinear model observed in Section 1.1. We also explain it with a minimalistic example and, for the first time in the literature, derive explicit theorems. We strongly believe that this is an important starting point toward understanding the dynamics of optimization methods in training, which is important for practitioners because it might help to understand how to accelerate the current methods.
>
> ---
>
> [1]: K. Lyu et al., Gradient Descent Maximizes the Margin of Homogeneous Neural Networks.
>
> [2]: D. Soudry, et al., The Implicit Bias of Gradient Descent on Separable Data.
>
> [3]: J. Wu, et al., Large Step-Size Gradient Descent for Logistic Loss: Non-Monotonicity of the Loss Improves Optimization Efficiency.
>
> [4]: J. Wu, et al., Implicit Bias of Gradient Descent for Logistic Regression at the Edge of Stability.
>
> [5]: R. Zhang, et al., Gradient Descent Converges Arbitrarily Fast for Logistic Regression via Large and Adaptive Stepsizes.
>
> [6]: R. Zhang, et al., Minimax Optimal Convergence of Gradient Descent in Logistic Regression via Large and Adaptive Stepsizes.

---

> > ### Author Rebuttal · Reviewer_4ybP · 2026-04-02
> >
> > I appreciate the effort made by the authors, but I will maintain my current rating. The main theoretical limitations remain unresolved. Several key questions raised by the reviewer, such as the justification of the large norm, the transition from practical initialization, and the extension beyond highly simplified settings, are either left unanswered or acknowledged as open problems. As a result, the current contribution falls short of providing a sufficiently complete or general theoretical understanding for acceptance.

---

> > > ### Author Response · Authors · 2026-04-04
> > >
> > > Thank you for the rebuttal acknowledgment.
> > >
> > > Our main theoretical claims and analysis are derived in the regime where the norm of the iterates is large. We emphasize that this regime is practically relevant, as gradient descent naturally increases the norms over time (see “Numerical Observation of the Norms” in Section 2). Therefore, our theory applies after the initial warm-up phase. Moreover, it is always possible to initialize with a large norm to operate in the regime supported by our theory.
> > >
> > > To the best of our knowledge, this is the first work to observe and provide strong theoretical justification that GD with nonlinear models and logistic loss exhibits implicit acceleration, a phenomenon not previously observed. Analyzing and proving this is highly challenging, not only for us but for the community as a whole. If it were a simple task, implicit acceleration in this setting with logistic losses would have been discovered long ago. For the first time, we prove that nonlinear models can converge $\sqrt{d}$ times faster than linear ones, which we believe is an interesting result for the community.

---

### Official Review · Reviewer_xQXr · 2026-03-12

**Soundness:** 3
**Presentation:** 3
**Significance:** 2
**Originality:** 3
**Overall Recommendation:** 4
**Confidence:** 3

**Summary:**

This paper analyzes gradient descent (GD) on logistic loss for a nonlinear reparameterization of a linear classifier, specifically the convolutional linear model $m_{cv}(b;c,v) = (c * b)^\top v$ with kernel size $k = 2$. Previous work has shown that GD in this setting is equivalent to the Perceptron algorithm. The authors further show that, in the large norm regime, GD can be reduced to a generalized perceptron-style method called the Quadratic Perceptron Algorithm, which provides a simpler view of the optimization dynamics.

The main theoretical result proves that when the initialization direction is fixed and its norm tends to infinity, the normalized gradient descent iterates follow the same directions as the quadratic perceptron updates and produce identical training predictions. The authors study a two-sample synthetic dataset and prove that the noisy quadratic perceptron finds a separator in $\tilde{\Theta}\left(\frac{\sqrt{d}}{\mu}\right)$ steps, while the classical perceptron requires at least $\Omega\left(\frac{d}{\mu}\right)$ steps. Experiments on this construction, as well as on datasets such as CIFAR-10, provide empirical support for the theory.

The paper also sketches extensions to larger kernels, two-layer linear models, and multilinear networks by reducing their dynamics to generalized perceptron-style updates.

**Compliance With Llm Reviewing Policy:**

Affirmed.

**Final Justification:**

I appreciate the authors’ response. Based on their response and the other reviews, I will maintain my overall positive score.

**Key Questions For Authors:**

1. For the algorithm that is actually analyzed, namely the Quadratic Perceptron Algorithm with noise, the update rule appears somewhat different from the earlier version beyond simply adding noise. Are these procedures equivalent if the noise term is ignored? If they are the same, it would be better to present them in a consistent form. Otherwise, the differences should be clearly stated.

2. What step size is used for the Perceptron Algorithm in Theorem 5.2? This does not seem to be explicitly specified. It would be helpful to clarify this, especially to understand whether the comparison is fair (for example, whether the step size is set to the largest value allowed by the analysis).

3. What are the main technical challenges in extending the results beyond the two sample example studied in the paper? The current proof appears highly specialized to the specific data construction and setup, making it difficult to assess whether the analysis could generalize to more complex settings.

**Limitations:**

yes

**Strengths And Weaknesses:**

**Strengths**

1. Understanding the dynamics of gradient descent with large step sizes in certain nonlinear models is an interesting research question.

2. This paper approaches the problem by connecting gradient descent to variants of perceptron-style algorithms in the logistic regression setting under nonlinear parameterizations, which appears to be a novel and interesting perspective.

3. The paper is generally easy to follow.

**Weaknesses**

1. The theoretical separation between the Quadratic Perceptron Algorithm and the classical Perceptron Algorithm is demonstrated only on a two-sample example. This setting appears rather simple and may limit the broader significance of the result.

2. Since the ultimate goal is to characterize the behavior of gradient descent, the current analysis relies heavily on its connection to perceptron-type algorithms. It would be better to present a more complete result directly for gradient descent. At present, the argument depends on several reductions and simplifying assumptions (e.g., large-norm initialization, adding noise, etc.), and it is not entirely clear whether all of these conditions can be satisfied simultaneously.

   Moreover, given the simplicity of the two-sample setting, it may be possible to provide more concrete guarantees. For example, the paper could specify how large the initialization norm must be, since the results still depend on these quantities and they are currently only assumed to hold in order to establish the connection between gradient descent and the perceptron dynamics.

---

> ### Author Rebuttal · Authors · 2026-03-26
>
> Thank you for the review!
>
> > The theoretical separation between the Quadratic Perceptron Algorithm and the classical Perceptron Algorithm is demonstrated only on a two-sample example. This setting appears rather simple and may limit the broader significance of the result.
>
> This is true. However, note that even for such a simple setup, we provide the first theoretical guarantees. Extending these results is important and may be very technical, since analyzing the behavior of GD in non-linear models is mathematically challenging. At the very least, we have resolved this question in a minimalistic example.
>
> > Since the ultimate goal is to characterize the behavior of gradient descent, the current analysis relies heavily on its connection to perceptron-type algorithms. It would be better to present a more complete result directly for gradient descent. At present, the argument depends on several reductions and simplifying assumptions, and it is not entirely clear whether all of these conditions can be satisfied simultaneously.
>
> The added noise assumption is simply a formalization of the fact that all operations on computers and GPUs are performed imprecisely. Indeed, the large-norm initialization is not always practical. However, notice the discussion "Numerical observation of the norms" in Sec. 2, where we explain that the norm of the iterates increases. Thus, the assumption that $|w_0|$ is large is practical up to the first ``warm-up'' steps. Note that even in this regime, our results are new and insightful. Extending them to the regime where the initial point has a small norm is important but significantly more technical; we rely on the fact that all operations are linear (see “How can the reduction help us?” in Sec. 2).
>
> > Moreover, given the simplicity of the two-sample setting, it may be possible to provide more concrete guarantees. For example, the paper could specify how large the initialization norm must be, since the results still depend on these quantities and they are currently only assumed to hold in order to establish the connection between gradient descent and the perceptron dynamics.
>
> Even in the regime where $|w_0| \to \infty$, analyzing the two-sample setting is a very difficult task. It is also not entirely clear how to extend the analysis to the case $| w_0 | < \infty$, since we rely on the fact that all operations are linear (see “How can the reduction help us?” in Section 2). Without $| w_0| \to \infty$, the discussion there does not hold.
>
> > For the algorithm that is actually analyzed, namely the Quadratic Perceptron Algorithm with noise, the update rule appears somewhat different from the earlier version beyond simply adding noise. Are these procedures equivalent if the noise term is ignored? If they are the same, it would be better to present them in a consistent form. Otherwise, the differences should be clearly stated.
>
> They are not fully equivalent.  To simplify the analysis, compared to Quadratic Perceptron Algorithm with Noise, we do not consider the case where both samples are misclassified. In other words, if both samples are misclassified, then we update the iterate using only the first sample. We have clarified this in the PDF.
>
> > What step size is used for the Perceptron Algorithm in Theorem 5.2? This does not seem to be explicitly specified. It would be helpful to clarify this, especially to understand whether the comparison is fair (for example, whether the step size is set to the largest value allowed by the analysis).
>
> In the Perceptron Algorithm, the step size is fixed to $1.$ Notice that the comparison is fair because the predictions of the Perceptron Algorithm are invariant to the choice of step size in the sense that the iterates generated by $z_{t+1} = z_{t} + \frac{1}{n}\sum_{i \in S_t} y_i  b_i$ and $\bar z_{t+1} = \bar z_{t} + \frac{\gamma}{n}\sum_{i \in \bar S_t} y_i  b_i$ have the same predictions ($S_t$ and $\bar S_t$ are equal) if $\bar z_{0} = \gamma z_{0}.$ Thus, Theorem 5.2 works with any step size $\gamma$ and for any starting point $z_0 \in B(0,\gamma \varepsilon).$ We have already clarified this in the PDF.
>
> > What are the main technical challenges in extending the results beyond the two sample example studied in the paper? The current proof appears highly specialized to the specific data construction and setup, making it difficult to assess whether the analysis could generalize to more complex settings.
>
> There are many challenges, which we describe in Section 5.1. Intuitively, with two samples, the ping-pong behavior can occur only between two “players,” and it is sufficient to analyze the sequence of steps $0,1,1,0,0,\ldots,0,1,1$, where $1$ corresponds to updating with the first sample and $2$ with the second. In the proof, we manually analyze the two-step subsequences $[0,0]$, $[0,1]$, $[1,0]$, and $[1,1]$. Extending this to the multisample scenario is much more combinatorially challenging, since the number of scenarios increases exponentially.

---

> > ### Author Rebuttal · Reviewer_xQXr · 2026-04-02
> >
> > Thank you for the response. In my assessment, this is a clear and well-written paper that focuses on a relatively simple two-data-point example. While this is helpful for providing a clean proof, it also limits the paper’s potential impact in more general or broader settings.

---

### Official Review · Reviewer_zKjP · 2026-03-13

**Soundness:** 3
**Presentation:** 4
**Significance:** 2
**Originality:** 3
**Overall Recommendation:** 4
**Confidence:** 4

**Summary:**

The paper studies gradient descent (GD) with logistic loss on a simple two-layer linear model that applies a short convolution (kernel size k=2) to inputs before a linear readout, and observes a striking empirical speedup over standard linear logistic regression despite the models having the same function class. To explain this “implicit acceleration,” the authors prove that GD dynamics in a large-norm regime reduce to a generalized perceptron, specifically a “Quadratic Perceptron Algorithm”. The paper provides theoretical proof as well as empirical evidence on CiFAR10 dataset by comparing the convergence and oscillation of the training curve linear model and two-layer nonlinear model.

**Compliance With Llm Reviewing Policy:**

Affirmed.

**Final Justification:**

This paper has a rigorous claim and proof of arguments, and explains an often omitted phenomenon. I tend to accept this paper.

**Key Questions For Authors:**

How are the parameters initialized in the figures?

**Limitations:**

Yes

**Strengths And Weaknesses:**

Strengths:
- Soundness: starting from a clear empirical observation, the (convolution+linear) model converges substantially faster than a linear model on a binary CIFAR-10 task, and then proposes different alternative explanations to this phenomenon and falsifies them, finally providing a theoretical proof to the proposition, making the paper very sound.
- Presentation: The figures are of high-quality. The major claims are clearly articulated. The core theorems are properly highlighted. The presentation is, in general, supreme.
- Originality: The paper does not propose a new algorithm, but deepens the understanding of the GD algorithm. The authors provide a formal proof of implicit acceleration due to nonlinearity and prove that gradient descent on logistic loss can be reformulated as a generalized perceptron algorithm. Its originality is proven.
- Reproducibility: I believe it is reproducible as a comprehensive appendix and supplementary materials are provided.

Weaknesses:
- The significance of this paper is in doubt in terms of generalizability and lacks real large-scale experiment evidence.
- The theorem explicitly assumes |w_0| → ∞, it could be concerning, as real training never starts with infinite norm, and it does not quantify how large is large enough in practice. Also, in Figure 1, how w_0 is initialized is not explicitly specified, which makes the plot less meaningful. There's no empirical comparison between training from large and small w_0.
- The paper claims the perceptron reduction explains acceleration, but the algorithm itself lacks general guarantees.
- The paper doesn't have enough empirical evidence proving that the training curve of the 2-layer model actually aligns with the quadratic perceptron algorithm.
- The paper advances the theoretical ML area, but may not have a big impact, as it builds on a very narrow setting:
(1) Although it briefly mentions the multi-layer model setting, there's no real empirical evidence on real large-scale models, or more complex architectures, for example, models with all kinds of normalization layers and fancy activation functions.
(2) The experiments are all on Cifar-10, which may introduce unnoticeable bias.
(3) There are strong assumptions, such as logistic regression, large w_0.
However, I acknowledge it is hard to include all things in one research paper.

---

> ### Author Rebuttal · Authors · 2026-03-26
>
> Thank you for your review! We now respond to the weaknesses.
>
> > The significance of this paper is in doubt in terms of generalizability and lacks real large-scale experiment evidence.
>
> Notice that we mainly investigate the two-linear model with kernel size $= 2$. In Section G, we run many more experiments on additional datasets, including EuroSAT, FashionMNIST, Food101, MNIST, and Gisette, not only CIFAR10. We believe that noticing acceleration in the two-layer model and explaining it in some regimes is an important endeavor, as it has never been observed before.
>
> > The theorem explicitly assumes |w_0| → ∞, it could be concerning, as real training never starts with infinite norm, and it does not quantify how large is large enough in practice.
>
> Please notice the discussion "Numerical observation of the norms" in Section 2, where we explain that the norm of the iterates increases. Thus, the assumption that $\|w_0\|$ is large is practical up to the first ``warm-up'' steps.
>
> > Also, in Figure 1, how w_0 is initialized is not explicitly specified, which makes the plot less meaningful. There's no empirical comparison between training from large and small w_0.
>
> In all experiments, the bias weights are turned off, and the models are initialized with the default PyTorch initialization: the weights of linear and convolution layers are sampled from $U(-\sqrt{1 / \textnormal{IF}}, \sqrt{1 / \textnormal{IF}})$ and $U(-\sqrt{1 / (\textnormal{IC $\times$ KS})}, \sqrt{1 / (\textnormal{IC $\times$ KS})}),$ respectively, where IF = the number of input features, IC = the number of input channels, KS = kernel size.
>
> During our preparation, we repeated the same experiment as in Figure 1, but with the weights scaled by factors of 10 and 100. We arrive at the same conclusion: the non-linear model converges faster. We have already added the new plots to the PDF (see the plot in **[Extra Experiments](https://www.dropbox.com/scl/fi/qzru23yso830q3dli9rf3/plots.png?rlkey=jmqrubrg2h9q1b6txe1wo9loj&st=bh3zbsh5&dl=0) [(url)](https://www.dropbox.com/scl/fi/qzru23yso830q3dli9rf3/plots.png?rlkey=jmqrubrg2h9q1b6txe1wo9loj&st=bh3zbsh5&dl=0)**: the first row corresponds to the linear mode, the first row corresponds to the nonlinear mode from the paper, the second and third are the new experiments where we scale the initial point by 10 or 100).
>
> > The paper claims the perceptron reduction explains acceleration, but the algorithm itself lacks general guarantees.
>
> In the paper, we only claim that our theoretical framework helps explain the phenomenon observed in Section 1.1. We are especially cautious with the wording and never attempt to overclaim our results.
>
> > The paper doesn't have enough empirical evidence proving that the training curve of the 2-layer model actually aligns with the quadratic perceptron algorithm.
>
> The fact that they are identical in the high-norm regime is supported by Theorem 2.1. In practice, when $|w_0| < \infty$, this is, of course, not true, and we have never claimed it.
>
> > The paper advances the theoretical ML area, but may not have a big impact, as it builds on a very narrow setting: (1) Although it briefly mentions the multi-layer model setting, there's no real empirical evidence on real large-scale models, or more complex architectures, for example, models with all kinds of normalization layers and fancy activation functions. (2) The experiments are all on Cifar-10, which may introduce unnoticeable bias. (3) There are strong assumptions, such as logistic regression, large w_0. However, I acknowledge it is hard to include all things in one research paper.
>
> We agree that (1) is an important future work that would require enormous computational effort. In this work, we focus on small-scale neural networks, and even in this setup, both empirical and theoretical results are new and might serve as a starting point for large-scale experiments and further theoretical work. Regarding (2), we run our experiments on EuroSAT, FashionMNIST, Food101, MNIST, and Gisette in Section G. (3) Assuming that the loss is the logistic loss is not restrictive at all since it is one of the most widely used losses in practice, and there are many research directions fully focused on this setup [2-6].
>
> ---
>
> [2]: D. Soudry, E. Hoffer, M. S. Nacson, S. Gunasekar, and N. Srebro, The Implicit Bias of Gradient Descent on Separable Data.
>
> [3]: J. Wu, P. L. Bartlett, M. Telgarsky, and B. Yu, Large Step-Size Gradient Descent for Logistic Loss: Non-Monotonicity of the Loss Improves Optimization Efficiency.
>
> [4]: J. Wu, V. Braverman, and J. D. Lee, Implicit Bias of Gradient Descent for Logistic Regression at the Edge of Stability.
>
> [5]: R. Zhang, J. Wu, and P. Bartlett, Gradient Descent Converges Arbitrarily Fast for Logistic Regression via Large and Adaptive Stepsizes.
>
> [6]: R. Zhang, J. Wu, L. Lin, and P. L. Bartlett, Minimax Optimal Convergence of Gradient Descent in Logistic Regression via Large and Adaptive Stepsizes.

---

> > ### Author Rebuttal · Reviewer_zKjP · 2026-04-04
> >
> > Thank you for the kind responses. It has convinced me of some details of the manuscript. I would like to raise my confidence.

---

### Official Review · Reviewer_rYfT · 2026-03-13

**Soundness:** 4
**Presentation:** 3
**Significance:** 3
**Originality:** 3
**Overall Recommendation:** 5
**Confidence:** 2

**Summary:**

This paper connects dynamics of some specific MLP models with classic systems that have been extensively analyzed. In particular, connections between convergence of "continuous" regression systems and the convergence of "perceptron" systems, which allows the faster convergence results to be applied to these more modern (albeit still rather simple) networks.

**Compliance With Llm Reviewing Policy:**

Affirmed.

**Final Justification:**

I have raised my rating in light of the discussions. A theoretical result showing a speedup, as opposed to a slowdown, from the nonlinearity, is of general interest. It might take more than one paper to really figure out what's going on, but nobody will pursue it if they don't know about it.

**Key Questions For Authors:**

-

**Limitations:**

yes

**Strengths And Weaknesses:**

This is not really my area and I am not up to speed on recent results. I don't trust myself to spot errors in the theorems or proofs.

The work seems sound. The presentation is reasonably clear, although I would find low-dimensional examples and diagrams giving a geometric intuition helpful. The does seem significant, albeit not highly, and original.

In its current form the work does seems a bit incomplete, so although it is interesting and publishable in its current form, a bit more development seems likely to make it more coherent and powerful and of more general interest.

TECHNICAL POINTS

- There are connections drawn (e.g., in the Implicit Acceleration paragraph in §7) that could use better support and a deeper exposition.

- There is are transitions to chaos in Fig. 1 and in the supplementary material, a phenomenon explored decades ago where the transition was shown to result from a narrowing ravine. It seems likely that a similar phenomenon is occurring here.

MINOR THINGS
- Eq (1), this notation seems odd to me.
- gr, "equals to" should read "is equal to" or just "equals"
- The "Duda" reference has a stray "G." It is R. O. Duda, and P. E. Hart, and D. Stork"

---

> ### Author Rebuttal · Authors · 2026-03-26
>
> Thank you very much for the review and appreciating our work. In our revision, we have already fixed the spotted typos.
>
> > There are connections drawn (e.g., in the Implicit Acceleration paragraph in §7) that could use better support and a deeper exposition.
>
> The closest work is [1], which investigates an entirely different direction and analyzes $\ell_p$ losses. It is not fully clear how to draw a connection between the setups since their and our work fully rely on the structure of the considered losses.
>
> > There is are transitions to chaos in Fig. 1 and in the supplementary material, a phenomenon explored decades ago where the transition was shown to result from a narrowing ravine. It seems likely that a similar phenomenon is occurring here.
>
> In practice, losses are always observed to be jittery and chaotic. Of course, this is not new. However, we are not aware of any work that observes implicit acceleration in such a small non-linear model as in Figure 1.
>
> ---
>
> [1]: S. Arora, N. Cohen, and E. Hazan, On the Optimization of Deep Networks: Implicit Acceleration by Overparameterization.

---

> > ### Author Rebuttal · Reviewer_rYfT · 2026-04-03
> >
> > My only actual concern was that the work seems incomplete, which isn't something that one could reasonably expect to be addressed in a response to a review.
> >
> > The responses to the other reviews were also very informative.

---

### Decision · Program_Chairs · 2026-04-30

**Decision:**

Accept (regular)

**Comment:**

**Summary:** The paper studies gradient descent (GD) with logistic loss on a simple two-layer linear model that applies a short convolution (kernel size k=2) to inputs before a linear readout, and observes empirical speedup over standard linear logistic regression. To explain this “implicit acceleration,” the authors prove that GD dynamics in a large-norm regime reduce to a generalized perceptron, specifically a “Quadratic Perceptron Algorithm”. The paper provides theoretical proof as well as empirical evidence on CiFAR10 dataset by comparing the convergence and oscillation of the training curve of the linear model and the two-layer nonlinear model.

**Strengths:**
- Understanding the dynamics of optimization methods, such as GD, in the context of non-linear models, is an important direction
- Well-written paper starting from an empirical observation, and providing a theoretical explanation for the implicit acceleration phenomenon
- Novel analysis technique connects the GD dynamics to a quadratic perceptron and shows the theoretical benefit of non-linearity

**Weaknesses:**
- Simplified setup analyzing $n = 2$ sample dataset with the logistic loss and convolutional kernel size = 2 limits the widespread applicability of the theoretical result.

**Decision and Suggested Changes:** This paper was on the borderline. After carefully reading the rebuttal/discussion, I went through the paper myself. I think the paper's contributions merit acceptance. Addressing the following concerns will help strengthen the current version of the paper:
- Low-dimensional examples and diagrams giving a geometric intuition would be a useful addition (Rev. rYfT)
- Clarifying/conjecturing whether the large initialization condition is necessary would be helpful. Explaining the difficulty of analysis for small initialization norm is important. Perhaps obtaining a slower rate until the norm becomes large enough, and then following the quadratic perceptron proof, might be a reasonable compromise (Rev. zKjP, Rev. xQXr, Rev. 4ybP)
- Clarifying or empirically testing whether the implicit acceleration holds for losses without an exponential tail (such as the squared loss)